# ConTSG-Bench: A Unified Benchmark for Conditional Time Series Generation

Shaocheng Lan [1]   Shuqi Gu [1]   Zhangzhi Xiong [1]   Kan Ren [1]

## Abstract

Conditional time series generation plays a critical role in addressing data scarcity and enabling causal analysis in real-world applications. Despite its increasing importance, the field lacks a standardized and systematic benchmarking framework for evaluating generative models across diverse conditions. To address this gap, we introduce the **Con**ditional **T**ime **S**eries **G**eneration **Bench**mark (ConTSG-Bench). ConTSG-Bench comprises a suite of large-scale, well-aligned datasets spanning diverse conditioning modalities and levels of semantic abstraction, enabling systematic evaluation of representative generation methods across these dimensions with a comprehensive suite of metrics for generation fidelity and condition adherence. Both the quantitative benchmarking and in-depth analyses of conditional generation behaviors have revealed the traits and limitations of the current approaches, highlighting critical challenges and promising research directions, particularly with respect to precise structural controllability and downstream task utility under complex conditions. We have open-sourced ConTSG-Bench at `https://github.com/seqml/ConTSG-Bench`.

## 1. Introduction

Conditional time series generation (ConTSG) has emerged as a transformative capability for scientific and industrial advancement. Its application spans from realistic data simulation for healthcare and climate applications (Lai et al., 2025; Lu et al., 2024; Narasimhan et al., 2024), causal inference (Xia et al., 2025), to privacy-preserving data synthesis (Liu et al., 2025). While unconditional generation has seen significant progress (Pei et al., 2021; Desai et al., 2021; Jeon et al., 2022) with established benchmarks for

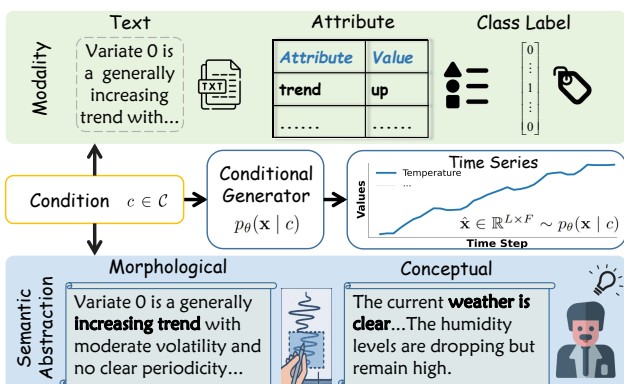

*Figure 1.* Conditional time series generation with varying conditioning modalities (text, attribute, class label) and semantic abstraction levels (morphological vs. conceptual).

fidelity and diversity (Ang et al., 2023), the research frontier has shifted toward *controllable* synthesis: the ability to generate high-fidelity time series data that strictly adheres to user-defined, multimodal conditions.

However, the landscape of ConTSG remains highly fragmented, hindering algorithmic innovation and practically effective model development. Current methodologies are isolated by their specific conditioning modalities: some rely on discrete class labels (Lee et al., 2023), others on structured attributes (Narasimhan et al., 2024), and recent works have begun exploring natural language descriptions (Gu et al., 2025). These models are typically evaluated on incompatible datasets with different condition modalities, making it infeasible to systematically compare conditional generation effectiveness or relative model performance.

Furthermore, evaluations in prior works overlook critical capabilities required for robust real-world deployment of ConTSG models. The primary dimension involves the *semantic abstraction of conditions* (Figure 1): while some methods specify target morphology directly (Dohi et al., 2025) (e.g., specify volatility and periodicity), others describe high-level concepts (Wagner et al., 2020) (e.g., weather conditions). The latter requires models to autonomously infer corresponding temporal patterns from abstract semantics, which is significantly more challenging. Beyond semantic abstraction level, practical simulation demands *fine-grained controllability* to execute precise local constraints, which is often obscured by aggregate global metrics (Williams et al.,

[1]School of Information Science and Technology, ShanghaiTech University, Shanghai, China. Correspondence to: Kan Ren <renkan@shanghaitech.edu.cn>.

2025). Finally, true robustness necessitates *compositional generalization*, ensuring models can synthesize time series underlying novel conditions, e.g., attribute combinations that are absent from the training distribution (Jing et al., 2024). As current evaluations typically isolate these factors under single-modality settings, the resulting performance landscape of ConTSG models remains incomplete.

To resolve these critical gaps, we introduce **ConTSG-Bench**, a unified evaluation benchmark for conditional time series generation. Our benchmark is the first to systematically disentangle condition types along two axes (Figure 1): modality (*class label*, *attribute*, *text*) and semantic abstraction level (*morphology*, *concept*). We curate a series of large-scale datasets featuring aligned conditions across all three modalities, including dual-level annotations for selected subsets to enable controlled cross-abstraction comparisons. Furthermore, ConTSG-Bench provides a unified evaluation suite that jointly assesses fidelity, condition adherence, fine-grained control, compositional generalization and downstream utility, allowing model behaviors to be characterized along multiple practically relevant dimensions.

Leveraging this framework, we conduct a rigorous evaluation of representative models, yielding several pivotal insights. For instance, we observe that text-conditioned models achieve the highest performance ceiling yet exhibit significant variance across architectures. Notably, the evaluated generators still struggle with precise fine-grained control and compositional generalization, especially under segment-level and out-of-distribution conditions, suggesting that existing methods may lack the structural inductive biases or algorithmic innovation necessary for complex real-world synthesis.

In summary, our contributions are summarized as follows:

- *A Unified Benchmarking Framework*: We establish the first systematic evaluation protocol for conditional time series generation, covering diverse condition types and multi-faceted metrics for fidelity and condition adherence.

- *Multimodal Aligned Datasets*: We construct large-scale datasets with aligned multimodal conditions and varied semantic abstraction levels, specifically designed to address the scarcity of aligned data and enable rigorous cross-modality benchmarking.

- *Systematic Evaluation and Analysis*: We provide an in-depth characterization of state-of-the-art models, uncovering critical bottlenecks that can inform future research in conditional time series generation. To facilitate reproducibility and future research, we publicly release all code, datasets, and evaluation pipelines.

## 2. Related Works

**Time Series Generation**    Early work on time series generation mainly focuses on unconditional synthesis, where the goal is to model the marginal distribution of a sequence without explicit control signals. Representative approaches include Generative Adversarial Network (GAN) (Goodfellow et al., 2020) and Variational Autoencoders (VAE) (Kingma & Welling, 2019) based models such as TimeGAN (Pei et al., 2021), TimeVAE (Desai et al., 2021), and GT-GAN (Jeon et al., 2022), used for tasks like data augmentation and privacy preservation. Diffusion-TS (Yuan & Qiao, 2024) further introduces an interpretable diffusion architecture for general time-series generation with trend-season decomposition. These methods establish the basic toolkit for time series synthesis but lack mechanisms for fine-grained control over the generated trajectories.

More recently, the field has shifted towards conditional time series generation, where models are guided by auxiliary information. *Label-based* approaches use discrete class labels within conditional Generative Adversarial Networks (GAN) (Goodfellow et al., 2020) or Variational Auto-Encoder (VAE) (Kingma & Welling, 2019) frameworks. TTS-CGAN (Li et al., 2022) employs a Transformer-based conditional GAN with auxiliary classification, while TimeVQVAE (Lee et al., 2023) learns discrete latent codes with label-aware priors to improve fidelity.

Beyond labels, *attribute-conditioned* models condition on low-dimensional metadata, including categorical and continuous covariates. TimeWeaver (Narasimhan et al., 2024) leverages attention-based diffusion with heterogeneous metadata; WaveStitch (Shankar et al., 2025) employs state-space-based diffusion for tabular series with hierarchical attributes; TEdit (Jing et al., 2024) uses multi-scale patch diffusion for attribute-guided editing.

Recently, a complementary line of work explores *text-conditioned* time series generation, using natural language as the conditioning modality. Several concurrent methods study general text-to-time-series generation with diffusion-based (Ho et al., 2020) or Transformer-based (Vaswani et al., 2017) architectures. BRIDGE (Li et al., 2025) and VerbalTS (Gu et al., 2025) propose domain-agnostic diffusion-based generators that couple text encoders with time-series backbones, with VerbalTS further introducing multi-view noise estimation and multi-focal text processing to enhance semantic alignment. T2S (Ge et al., 2025) combines flow matching with a diffusion transformer (DiT) backbone for prompt-based series generation, while Text2Motion (Guo et al., 2022) employs a latent-space autoregressive VAE originally designed for motion synthesis. DiffuSETS (Lai et al., 2025) instead targets the medical domain, conditioning 12-lead ECG synthesis on clinical reports and patient information. Overall, conditional time series generation is

moving from low-dimensional structured conditions toward flexible natural language prompts, but each method is evaluated on its own datasets, condition formats, and metrics.

**Time Series Benchmark** Standardized benchmarking serves as a critical foundation for advancing time series research. Within this landscape, evaluation frameworks for forecasting are comparatively mature. TSLib (Wang et al., 2024), ProbTS (Zhang et al., 2024b) and GIFT-Eval (Aksu et al., 2024) provide unified codebases, large collections of datasets, and standardized pipelines for evaluating deep and foundation models across diverse forecasting settings. Recent work further studies counterfactual forecasting with textual future conditions (Gu et al., 2026), highlighting the value of natural language for specifying hypothetical future scenarios. These forecasting-oriented efforts, however, primarily target predictive accuracy, uncertainty quantification, or future-condition consistency rather than generative modeling under rich conditioning modalities.

Regarding time series generation, TSGBench (Ang et al., 2023) is the pioneering benchmark that standardizes the evaluation of unconditional models. While it incorporates limited experiments on weakly conditioned settings (e.g., discrete class labels), it does not provide a systematic evaluation framework for controllable generation. Crucially, it fails to cover heterogeneous conditioning modalities, such as structured attributes and natural language text, and lacks metrics designed to verify the condition adherence between complex conditions and generated time series.

Our work is complementary to these benchmarks and focuses specifically on conditional time series generation. By establishing aligned conditions across modalities and semantic levels, ConTSG-Bench enables systematic cross-method comparison and reveals failure modes that remain invisible under existing protocols. Table 1 summarizes the key differences across conditioning modalities and evaluation dimensions.

## 3. ConTSG-Bench Framework

This section formalizes the conditional generation task, outlines our research questions, and describes the dataset construction and evaluation pipeline.

### 3.1. Task: Conditional Time Series Generation

We study conditional time series generation, where the goal is to synthesize realistic time series that both match the real-data distribution and adhere to a user-specified condition. Let $\mathbf{x} \in \mathbb{R}^{L \times F}$ denote a time series of length $L$ with $F$ variables. Condition is denoted by $c \in \mathcal{C}$, where $\mathcal{C}$ may take different modalities, including (i) discrete class label $c^{\text{label}}$, (ii) structured attribute vector $c^{\text{attr}}$ with heterogeneous

categorical fields, and (iii) natural-language description $c^{\text{text}}$.

Given a dataset of aligned pairs $\mathcal{D} = \{(\mathbf{x}_i, c_i)\}_{i=1}^{N}$ sampled from an unknown real joint distribution $p_r(\mathbf{x}, c)$, a conditional generator aims to learn a distribution $p_\theta(\mathbf{x} \mid c)$ such that samples $\hat{\mathbf{x}} \sim p_\theta(\mathbf{x} \mid c)$ are (1) realistic with respect to the marginal data distribution and (2) faithful to the condition. Importantly, these two objectives are orthogonal: a model may produce plausible outputs that ignore the condition, or faithfully follow the condition while generating implausible patterns. Our evaluation therefore assesses fidelity and adherence separately.

Beyond modality, conditions also vary in *semantic abstraction* (Figure 1). We distinguish two levels: *morphological* conditions that directly specify temporal structures (e.g., trends, peaks, and their placement), and *conceptual* conditions that describe high-level semantics (e.g., a clinical diagnosis) and require the model to infer the corresponding temporal patterns. ConTSG-Bench systematically covers both dimensions under a unified task formulation.

### 3.2. Evaluation Dimensions

ConTSG-Bench is designed not only to rank models, but also to stress-test the key capabilities that conditional time series generators are expected to have in practice: producing realistic data, following conditions, handling different kinds of conditions, and being useful for downstream tasks. To make these desiderata explicit, we organize our study around five research questions.

Generating realistic time series and following conditions are complementary but distinct capabilities: a model may produce plausible outputs that ignore the condition, or faithfully adhere to the condition while generating implausible patterns. To disentangle these two dimensions, we first ask: **RQ1 (Overall benchmarking).** How do representative conditional time series generation models compare in terms of *generation fidelity* and *condition adherence* across diverse datasets and conditioning modalities?

Beyond overall performance, models may behave very differently depending on the *semantic abstraction* of conditions. For instance, in ECG generation, a condition can describe observable waveform morphology ("irregular R-R intervals and absent P-waves") or a high-level clinical concept ("atrial fibrillation"). Both refer to the same underlying pattern, yet the latter requires the model to infer temporal structures from abstract domain semantics. Since conceptual conditions require expert annotation while morphological descriptions are domain-agnostic and lower-cost, understanding model sensitivity to this distinction has practical value. This motivates **RQ2 (Semantic abstraction):** How sensitive are models to the semantic type of conditions, specifically morphological versus conceptual descriptions,

*Table 1.* Comparison of conditional time series generation methods and benchmarks along three dimensions: (1) supported condition modalities, (2) semantic abstraction levels, and (3) evaluation dimensions beyond fidelity, which is commonly assessed. **Abbreviations:** Attr = Attribute; Morph = Morphological; Adh. = Condition Adherence; Fine-gr. = Fine-grained control; Comp. Gen. = Compositional generalization; Down. Util. = Downstream utility.

| Method | Condition Modality | | | Condition Semantic | | Evaluation Dimensions | | | |
|---|---|---|---|---|---|---|---|---|---|
| | Text | Attr | Label | Morph | Concept | Adh. | Fine-gr. | Comp. Gen. | Down. Util. |
| *Existing Benchmark* | | | | | | | | | |
| TSGBench (Ang et al., 2023) | ✗ | ✗ | ✓ | ✗ | ✗ | ✗ | ✗ | ✗ | ✓ |
| *Label-conditioned* | | | | | | | | | |
| TimeVQVAE (Lee et al., 2023) | ✗ | ✗ | ✓ | ✗ | ✗ | ✗ | ✗ | ✗ | ✓ |
| TTS-CGAN (Li et al., 2022) | ✗ | ✗ | ✓ | ✗ | ✗ | ✗ | ✗ | ✗ | ✓ |
| Diffusion-TS (Yuan & Qiao, 2024) | ✗ | ✗ | ✓ | ✗ | ✓ | ✗ | ✗ | ✗ | ✓ |
| *Attribute-conditioned* | | | | | | | | | |
| TimeWeaver (Narasimhan et al., 2024) | ✗ | ✓ | ✗ | ✗ | ✗ | ✓ | ✗ | ✗ | ✓ |
| TEdit (Jing et al., 2024) | ✗ | ✓ | ✗ | ✗ | ✗ | ✓ | ✗ | ✓ | ✗ |
| WaveStitch (Shankar et al., 2025) | ✗ | ✓ | ✗ | ✗ | ✗ | ✗ | ✗ | ✗ | ✗ |
| *Text-conditioned* | | | | | | | | | |
| Text2Motion (Guo et al., 2022) | ✓ | ✗ | ✗ | ✓ | ✗ | ✓ | ✗ | ✗ | ✗ |
| DiffuSETS (Lai et al., 2025) | ✓ | ✗ | ✗ | ✗ | ✓ | ✗ | ✗ | ✗ | ✗ |
| BRIDGE (Li et al., 2025) | ✓ | ✗ | ✗ | ✗ | ✓ | ✓ | ✗ | ✗ | ✓ |
| T2S (Ge et al., 2025) | ✓ | ✗ | ✗ | ✓ | ✗ | ✓ | ✗ | ✗ | ✗ |
| VerbalTS (Gu et al., 2025) | ✓ | ✓ | ✓ | ✓ | ✗ | ✓ | ✗ | ✗ | ✗ |
| *Unified Benchmark* | | | | | | | | | |
| **ConTSG-Bench (Ours)** | ✓ | ✓ | ✓ | ✓ | ✓ | ✓ | ✓ | ✓ | ✓ |

when the underlying time series is fixed?

Practical applications often require precise control over *local* temporal patterns. For instance, in network monitoring, a user may specify "signal drops in the middle segment, then recovers in the final quarter". **RQ3 (Fine-grained control)** probes: To what extent can models follow such fine-grained local specifications, and what are the dominant failure modes?

In practice, test-time conditions may be out-of-distribution, involving novel attribute combinations unseen during training. For example, a model may encounter "high volatility + downward trend + multiple level shifts", a combination absent in the training set. Robust models should compositionally understand each attribute rather than memorize training combinations. This raises **RQ4 (Compositional generalization)**: Can models generalize to novel attribute combinations where multiple attribute values differ from those observed during training?

Ultimately, generation quality is meaningful only if it translates to practical value. In this benchmark, downstream classification serves as an initial standardized utility test for synthetic data substitutability, while RQ1–RQ4 evaluate more fundamental generation capabilities: fidelity, condition adherence, semantic abstraction, fine-grained control, and compositional generalization. A key use case is data scarcity: when real labeled data is limited, can generated samples substitute for real data in training downstream classifiers? This leads to **RQ5 (Practical utility)**: How well can generated data substitute for real data in downstream classification tasks?

Sections 4.1–4.5 present experimental results and findings for each research question.

### 3.3. Datasets

ConTSG-Bench comprises eight datasets spanning diverse domains including healthcare, meteorology, energy, traffic, and network telemetry, covering both synthetic benchmarks with known ground-truth and real-world data with authentic temporal dynamics. Full statistics are provided in Appendix A.

As discussed in Section 2, existing conditional generators operate under heterogeneous conditioning modalities: label-conditioned methods use discrete class labels, attribute-conditioned methods condition on structured metadata, and text-conditioned methods leverage natural language prompts. A key contribution of ConTSG-Bench is providing *aligned conditions across all three modalities* for each time series: a class label $c^{\text{label}}$, a structured attribute vector $c^{\text{attr}}$, and a textual description $c^{\text{text}}$. Since these conditions are derived from the same underlying semantics, our benchmark enables controlled cross-modality comparison that is otherwise infeasible with existing datasets.

To align these modalities, we design an LLM-based pipeline with three stages. First, we prompt an LLM to generate morphological captions $c^{\text{text}}$ that describe observable

temporal patterns (e.g., trend direction, periodicity, local anomalies) from time series (Appendix A.2.1). Second, we apply an iterative *attribute-schema discovery* procedure: the LLM proposes candidate attributes from sampled captions, merges redundant categories, and finalizes a compact schema; attribute values are then extracted from each caption to form $c^{\text{attr}}$ (Appendix A.2.3). Finally, class labels $c^{\text{label}}$ are obtained by indexing unique attribute combinations (Appendix A.2.4). This pipeline ensures consistency across modalities while requiring minimal manual effort. As an independent audit of the LLM-generated captions, we conduct a 4-way annotation matching validation on a reported subset of LLM-annotated datasets, obtaining 69.45% overall accuracy versus a 25% chance baseline (Appendix A.2.2).

Beyond modality, we observe that existing text-conditioned datasets conflate two distinct levels of *semantic abstraction*. Some datasets provide *morphological* conditions that directly describe observable temporal structures, such as trend shapes and local patterns (Ge et al., 2025); others provide *conceptual* conditions that describe high-level domain semantics without revealing the waveform (Li et al., 2025); still others mix both types without explicit distinction (Feng et al., 2025). ConTSG-Bench explicitly disentangles these two levels: for PTB-XL and Weather datasets, we provide *paired* morphological and conceptual conditions for the same time series, enabling direct comparison of how models handle different abstraction levels (Appendix A.3).

This two-dimensional design, systematically covering condition modality and semantic abstraction, distinguishes ConTSG-Bench from prior work and enables more fine-grained analysis of conditional generation capabilities.

### 3.4. Evaluated Methods

To comprehensively assess conditional time series generation, we benchmark eleven representative models spanning all three conditioning modalities supported by ConTSG-Bench (Table 1). We include *label-conditioned* models TimeVQVAE (Lee et al., 2023), TTS-CGAN (Li et al., 2022), and adapted Diffusion-TS (Yuan & Qiao, 2024), which condition on discrete class labels and represent early VQ/GAN-based approaches as well as a diffusion-based classifier-guided baseline. For *attribute-conditioned* generation, we evaluate TimeWeaver (Narasimhan et al., 2024), TEdit (Jing et al., 2024), and WaveStitch (Shankar et al., 2025), which condition on structured attribute vectors containing heterogeneous categorical and continuous fields, and are designed for controllable synthesis and counterfactual analysis. We also benchmark five recent *text-conditioned* generators: BRIDGE (Li et al., 2025), VerbalTS (Gu et al., 2025), T2S (Ge et al., 2025), DiffuSETS (Lai et al., 2025), and Text2Motion (Guo et al., 2022), which generate time series from natural-language descriptions using various gen-

erative approaches and text encoding strategies. As shown in Table 1, text-conditioned models exhibit the richest diversity in design choices, yet their coverage of semantic abstraction levels and evaluation dimensions remains limited prior to our benchmark. All models are trained on the same training splits of ConTSG-Bench with validation-based early stopping, ensuring fair comparison. Detailed model implementations and training configurations are provided in Appendix B.

### 3.5. Evaluation Protocol

Since conditional generation is inherently one-to-many, each model produces $K$ samples $\{\hat{\mathbf{x}}^{(k)}\}_{k=1}^{K} \sim p_\theta(\mathbf{x} \mid c)$ per condition. Depending on the evaluation goal, metrics either aggregate statistics over all $K$ samples or adopt a best-of-$K$ strategy that selects the sample closest to a reference time series.

We organize our evaluation along two complementary axes: *generation fidelity*, which assesses whether generated series are statistically realistic regardless of the specific condition; and *condition adherence*, which measures alignment between the generated output and the specified condition. Within each axis, we employ both *embedding-based* and *statistical* metrics. Embedding-based metrics require a shared representation space where time series and textual conditions can be directly compared. To this end, we train a Contrastive Text–Time Series Pretraining (CTTP) model (Gu et al., 2025) per dataset, which learns aligned representations by maximizing similarity between matched (time series, text) pairs. The resulting time-series encoder $\phi_{\text{ts}}$ and text encoder $\phi_{\text{text}}$ are frozen and reused for all embedding-based evaluations (see Appendix C.2 for training details). We audit CTTP score faithfulness with an LLM-based pairwise preference validation in Appendix C.2.1.

The detailed evaluation protocols, including specific metrics, formulas, and experimental settings for each research question, are presented alongside their corresponding experimental results in Section 4.

## 4. Experimental Results

### 4.1. Overall Benchmarking

**Protocol.** We assess both generation fidelity and condition adherence using the following metrics. *Generation fidelity* assesses whether generated series are statistically realistic regardless of specific conditions. We report embedding-based metrics including Fréchet Inception Distance (FID) (Heusel et al., 2017) between CTTP embeddings of real and generated series, and Precision/Recall (Kynkäänniemi et al., 2019) in the embedding space, as well as statistical metrics such as marginal distribution difference, autocorrelation difference, skewness, and kurtosis differences. *Condition*

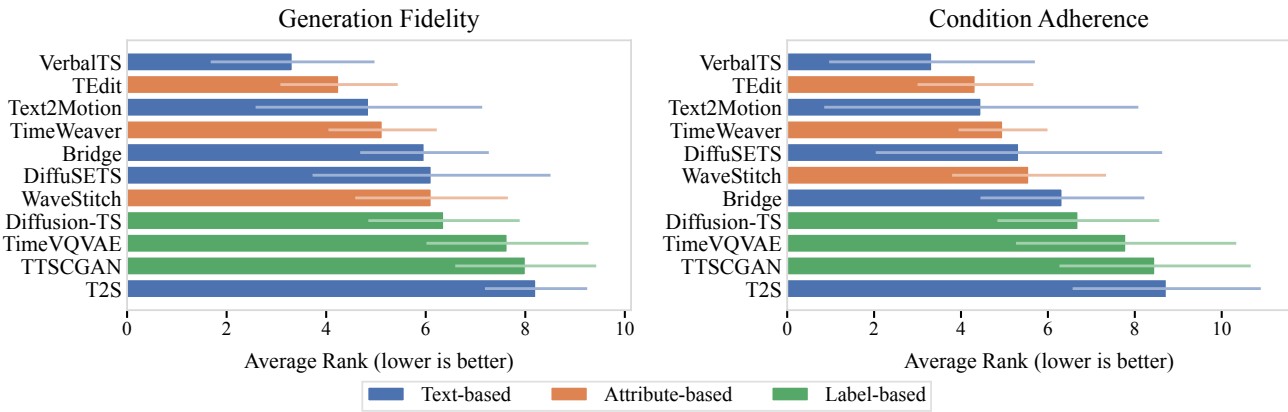

*Figure 2.* Model ranking under two metric groups: (left) generation fidelity that evaluates marginal distribution of generated time series; (right) condition adherence that evaluates joint/conditional alignment between time series and conditions.

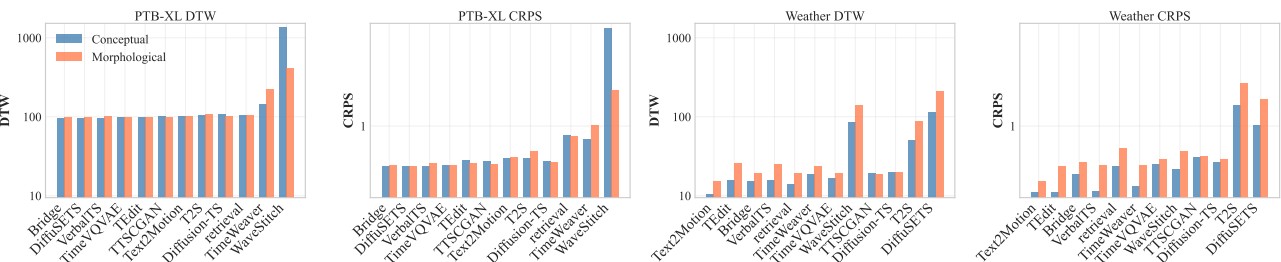

*Figure 3.* **Morphological vs. conceptual conditioning: absolute performance.** DTW and CRPS on PTB-XL and Weather under the two condition types.

*adherence* measures alignment between generated outputs and specified conditions. We report CTTP Score (the dot product between the CTTP time-series and text embeddings for a generated time series and its conditioning text), Joint Fréchet Time Series Distance (J-FTSD) (Narasimhan et al., 2024), and joint Precision/Recall, where each sample is represented by the concatenation of its time series embedding and condition embedding. Formal definitions are provided in Appendix C.1. For each model, dataset, and metric, we obtain a scalar score averaged over three random seeds. We normalize metric directions so that higher values indicate better performance, then convert scores to ranks per metric and dataset. To summarize overall performance, we first average ranks across all metrics within each metric group (fidelity or adherence), then average these group-level ranks across datasets; error bars reflect cross-dataset variability.

**Results.** Figure 2 presents overall rankings across both metric groups, with the left panel reflecting generation fidelity and the right panel reflecting condition adherence. Our results for RQ1 reveal three high-level patterns. First, *good generation fidelity does not guarantee condition adherence.* While some models (e.g., VerbalTS) perform consistently well on both dimensions, others (e.g., DiffuSETS) show significant rank improvements only under conditional evaluation, confirming the need to evaluate these two aspects

separately. Second, *text conditioning offers the highest performance ceiling but also the largest variance.* Text-conditioned models span both the best and worst ranks, whereas attribute-conditioned methods generally cluster in the upper-middle tier and label-conditioned baselines tend to fall into the lower tier. This suggests that while natural language provides richer expressiveness, current architectures vary widely in their ability to leverage it. Third, *cross-dataset robustness remains a major challenge.* The large error bars indicate that no model dominates across all datasets, and rankings can shift substantially depending on data characteristics. This motivates future work on domain-agnostic architectures and training strategies that generalize across heterogeneous time series domains. Detailed per-dataset metric scores are reported in Appendix D.2.

### 4.2. Morphological vs. Conceptual Conditions

**Protocol.** To compare how models handle conditions at different semantic abstraction levels, we need metrics that capture generation quality when the underlying time series is fixed but the condition form varies. Embedding-based metrics such as CTTP Score are sensitive to the textual form of conditions: morphological and conceptual descriptions have different text representations even when describing the same time series, making cross-type comparisons un-

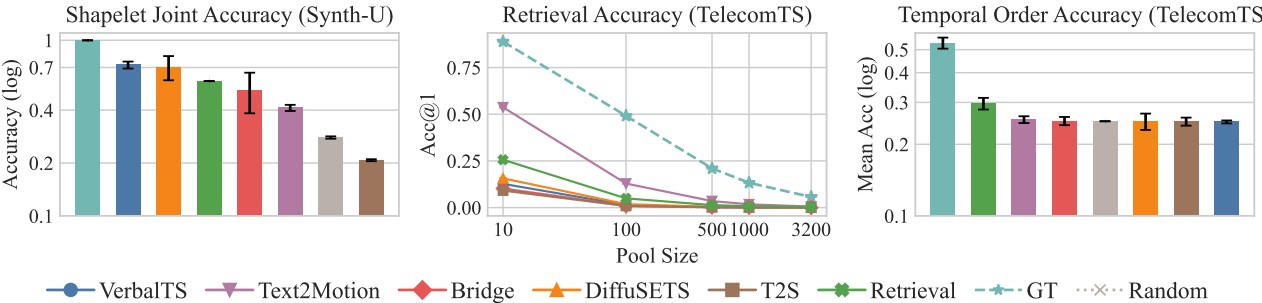

*Figure 4.* **Fine-grained control evaluation.** *Left:* Joint shapelet classification accuracy on Synth-U, where all three segment-level local patterns must be correctly generated. *Middle:* Segment retrieval accuracy (Acc@1) as a function of candidate pool size on TelecomTS-Segment. *Right:* Segment–text temporal order accuracy on TelecomTS-Segment. The retrieval curve denotes a non-parametric nearest-neighbor baseline.

fair. We therefore adopt reference-based metrics that use the source time series as a fixed anchor. Specifically, for each condition, we generate $K$ samples and compute Dynamic Time Warping (DTW) (Sakoe & Chiba, 1978) and Continuous Ranked Probability Score (CRPS) relative to the source time series from which the condition was derived. We report minimum DTW (best-of-$K$) and mean CRPS (over all $K$ samples) (Appendix C.1).

**Results.** Figure 3 reveals that condition semantics affect generation difficulty in a dataset-dependent manner. On PTB-XL, morphological and conceptual conditions lead to similar DTW/CRPS for most models, whereas on Weather the gap is substantial, with conceptual conditions often yielding lower error. This suggests that the relative difficulty of morphological versus conceptual conditioning depends on the intrinsic regularity of the underlying signals: highly structured domains (e.g., ECG) may be equally accessible from either condition type, while complex natural phenomena benefit more from expert-level conceptual descriptions. We provide additional analysis of model ranking stability across condition types in Appendix D.3.

### 4.3. Fine-Grained Control

**Protocol.** To evaluate whether models can follow fine-grained local specifications, we employ three complementary approaches depending on dataset characteristics. *(i) Classifier-based evaluation.* On synthetic data where the local pattern of each segment (e.g., peak, sag) is determined by the generation script, we train a segment-level 1D-CNN classifier to verify whether generated segments contain the specified patterns. We report joint classification accuracy, which requires all segment-level patterns to be correctly generated. *(ii) Retrieval-based evaluation.* For each segment of a generated sample, we construct a candidate pool containing its true segment-level description plus $n - 1$ distractors sampled from the test set, retrieve the closest description

using CTTP embeddings, and report top-1 retrieval accuracy. To reduce variance from pool composition, we repeat the construction $m$ times and average results. We compare against a naive retrieval baseline that retrieves the nearest training segment based on text embeddings. *(iii) Temporal order evaluation.* On real-world data with segment-level captions, we test whether each generated segment can correctly retrieve its corresponding positional description (e.g., segment 1 → description 1). Retrieval accuracy and confusion matrices reveal whether models preserve the intended temporal order. We instantiate these protocols on two datasets: Synth-U (three segments with controllable peaks and sags) and TelecomTS-Segment, where each sequence is partitioned into four segments with independent captions. Implementation details are provided in Appendix D.4.

**Results.** On Synth-U (Figure 4, left), most text-conditioned generators exceed the random baseline, indicating they can capture coarse local patterns when the underlying signal family is simple. However, only VerbalTS and DiffuSETS consistently outperform a naive retrieval baseline, indicating that simple retrieval is already highly competitive. Per-segment classifier accuracies in Appendix D.4.2 show that this ranking is stable across the beginning, middle, and end segments. On TelecomTS-Segment (Figure 4, middle and right), results differ markedly: as the candidate pool grows, most generators rapidly approach the random baseline, implying insufficient discriminability for segment-level retrieval. For temporal order, mean accuracy is near chance for all generators; detailed confusion matrices in Appendix D.4.3 reveal that failure patterns vary across models. Together, these results reveal that fine-grained controllability does not reliably transfer from simple synthetic data to real-world dynamics, and most models fail to achieve segment-level semantic alignment comparable to simple retrieval baselines. This motivates future work on segment-aware objectives and architectures with explicit positional control.

## 4.4. Compositional Generalization

**Protocol.** To assess generalization to novel attribute combinations, we adopt the same retrieval-based protocol as RQ3 and measure compositional distance from the training distribution. To quantify how far a test condition lies from training examples, we define the Hamming distance between two attribute vectors as $\text{HD}(c_1^{\text{attr}}, c_2^{\text{attr}}) = \sum_{j=1}^{M} \mathbf{1}[c_{1,j}^{\text{attr}} \neq c_{2,j}^{\text{attr}}]$, where $M$ is the number of attributes. For each test condition with attribute vector $c_{\text{test}}^{\text{attr}}$, we compute the average Hamming distance to its $k$ nearest neighbors in the training set:

$$d_{\text{knn}}(c_{\text{test}}^{\text{attr}}) = \frac{1}{k} \sum_{c^{\text{attr}} \in \text{KNN}_k(c_{\text{test}}^{\text{attr}})} \text{HD}(c_{\text{test}}^{\text{attr}}, c^{\text{attr}}). \quad (1)$$

Since CTTP encoders themselves may exhibit limited compositional generalization, we normalize retrieval accuracy as $\text{Acc}_{\text{norm}} = \text{Acc}_{\text{gen}}/\text{Acc}_{\text{ref}}$, where $\text{Acc}_{\text{gen}}$ and $\text{Acc}_{\text{ref}}$ denote accuracy using generated samples and reference time series respectively. We partition test samples into the lowest 20% and highest 20% by $d_{\text{knn}}$, corresponding to conditions closest to and farthest from the training distribution, and compare their $\text{Acc}_{\text{norm}}$ to quantify robustness to novel attribute combinations.

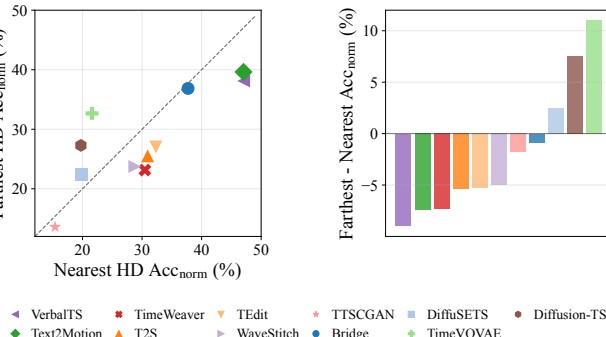

*Figure 5.* **Compositional generalization analysis.** *Left:* normalized retrieval accuracy for head (lowest 20% by $d_{\text{knn}}$, closest to the training distribution) vs. tail (highest 20% by $d_{\text{knn}}$, farthest from the training distribution) test samples; points below the diagonal indicate performance degradation on out-of-distribution combinations. *Right:* accuracy gap (tail − head) for each model, where negative values reflect sensitivity to novel attribute combinations.

Figure 5 compares normalized retrieval accuracy between the lowest 20% and highest 20% test samples by $d_{\text{knn}}$, corresponding to conditions closest to and farthest from the training distribution.

**Results.** Three patterns emerge from the results. First, most models exhibit performance degradation from head to tail, confirming that novel attribute combinations pose a broad challenge across current generators. Second, stronger models (e.g., VerbalTS, which also achieves better performance in Section 4.1) show larger drops yet their tail accuracy

still exceeds the head accuracy of weaker models, suggesting that better condition adherence provides an absolute advantage even under distribution shift. Third, models with minimal or reversed degradation (e.g., TimeVQVAE) tend to have low absolute accuracy, indicating that their apparent robustness stems from weak responsiveness to conditions rather than true compositional understanding. Together, these findings suggest that models which faithfully adhere to conditions are more sensitive to novel combinations, highlighting the need for architectures that can generalize individual attribute semantics beyond memorized training patterns.

## 4.5. Practical Utility

**Protocol.** To measure practical utility, we evaluate whether generated data can substitute for real data in training downstream classifiers. We train a multi-head classifier where each head predicts the value of a corresponding attribute, and compare two training settings: using fully real data versus fully generated data. We report macro-averaged accuracy across attribute classes and quantify utility loss using the *drop rate*:

$$\text{Drop Rate} = 1 - \frac{\text{acc}_{\text{gen}} - \text{acc}_{\text{rand}}}{\text{acc}_{\text{real}} - \text{acc}_{\text{rand}}}, \quad (2)$$

where $\text{acc}_{\text{real}}$ and $\text{acc}_{\text{gen}}$ denote classifier accuracy when trained on real and generated data respectively, and $\text{acc}_{\text{rand}}$ is the random-guessing baseline. This formulation normalizes the utility gap by the maximum achievable improvement over random guessing, making the metric comparable across datasets with varying task difficulty. A lower drop rate indicates better substitutability.

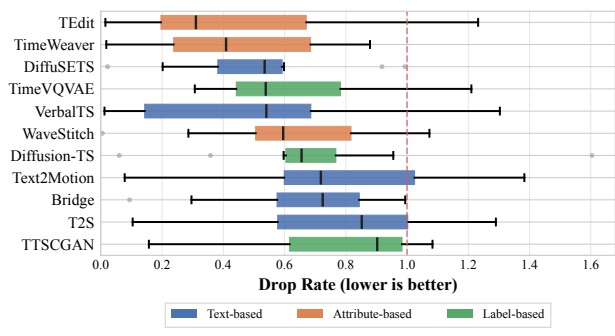

*Figure 6.* Drop Rate distribution across dataset configurations and models. Lower Drop Rate indicates better substitutability of generated data. Models are ordered by median Drop Rate across dataset configurations.

**Results.** Figure 6 visualizes the Drop Rate across all models and dataset configurations. In aggregate, most models achieve mean Drop Rates below 1.0, indicating that generated data often retains useful discriminative information.

However, several model–dataset pairs exceed 1.0 on complex datasets, suggesting that synthetic data can sometimes harm downstream training. Moreover, we observe substantial variance in model rankings across datasets, indicating that no single model consistently dominates. This suggests that the utility of generated data is highly dataset-dependent and cannot be reliably predicted from generation fidelity metrics alone. Appendix D.5 reports the sensitivity of RQ5 to the downstream classifier architecture across ten dataset configurations.

## 5. Conclusion and Future Work

We introduced ConTSG-Bench, the first comprehensive benchmark for conditional time series generation that spans multiple conditioning modalities and semantic abstraction levels. Our large-scale evaluation of representative models reveals several key insights. First, generation fidelity and condition adherence are complementary capabilities that require separate evaluation, and text-based conditioning offers the highest performance ceiling but also the widest variance. Second, the evaluated methods still show substantial limitations in fine-grained local control and compositional generalization: most models fail to surpass simple retrieval baselines on segment-level tasks, and models with stronger condition adherence often exhibit larger head-to-tail drops under novel attribute combinations while still maintaining stronger absolute performance than weaker models. Third, the downstream utility of generated data varies substantially across datasets and cannot be reliably predicted from fidelity metrics alone. These findings motivate future work on architectures with compositional inductive biases, segment-aware objectives, and domain-agnostic generalization strategies.

## Acknowledgment

This research was mainly supported by the National Natural Science Foundation of China (Grant No. 62406193). We are also supported by the ShanghaiTech AI Initiative (Grant No. AI2026B08).

The authors gratefully acknowledge further assistance provided by the Shanghai Frontiers Science Center of Human-centered Artificial Intelligence, the MoE Key Lab of Intelligent Perception and Human-Machine Collaboration, the ShanghaiTech GenAI Platform, and the HPC Platform of ShanghaiTech University.

## Impact Statement

This paper introduces a benchmark for conditional time series generation, aiming to facilitate standardized evaluation and reproducible research in this area. Time series generation has broad applications in domains such as health-care, finance, and climate science, where synthetic data can help address data scarcity and enable safer experimentation. While we do not foresee immediate negative societal impacts from our benchmarking framework itself, we acknowledge that generative models, if misused, could potentially produce misleading synthetic data. We encourage practitioners to apply appropriate validation when using generated time series in safety-critical applications.

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

# A. Dataset Construction Details

## A.1. Synthetic Datasets

We utilize synthetic datasets constructed in VerbalTS (Gu et al., 2025), including univariate dataset (**Synth-U**) and multivariate dataset (**Synth-M**). With a human-defined attribute set, both datasets are generated using an established pipeline which first synthesizes time series data based on sampled attributes from the set and then synthesizes corresponding textual description through substituting attribute values into the text templates. The **Synth-U** and **Synth-M** statistic details of the number of tokens in the text data are given in Table 2 and Table 3 respectively. Note that we utilize tokenizer from Long-clip(Zhang et al., 2024a) for all the datasets in our experiments.

| Set | Average Tokens | Median Tokens | Max Tokens | Std. Dev. |
|---|---|---|---|---|
| Training | 41.09 | 42.0 | 60 | 8.90 |
| Validation | 41.21 | 42.0 | 60 | 8.94 |
| Test | 41.21 | 42.0 | 60 | 8.92 |

*Table 2.* Summary of token number statistics for Synth-U dataset.

| Set | Average Tokens | Median Tokens | Max Tokens | Std. Dev. |
|---|---|---|---|---|
| Training | 62.23 | 63.0 | 83 | 8.97 |
| Validation | 62.27 | 63.0 | 83 | 9.10 |
| Test | 62.39 | 63.0 | 82 | 9.17 |

*Table 3.* Summary of token number statistics for Synth-M dataset.

### A.1.1. ATTRIBUTE SET

| Attribute Category | Value Options |
|---|---|
| Trend Types[1] | [Linear, Quadratic, Exponential, Logistic] |
| Trend Directions[1] | [Up, Down] |
| Season Cycles[1] | [0, 1, 2, 4] |
| Local Shapelets[2] | [None, Single Peak, Sag, Double Peaks] |
| High Freq. Components[2] | [0, 16, 32, 64] |
| Multivariable* | [X/Y-axis Flip, Shift Forward/Backward] |

\* Only applicable to Synth-M. [1] Primary Attribute. [2] Secondary Attribute.

*Table 4.* Attribute Set

| Trend Type | Function |
|---|---|
| Linear | $\mathbf{x}_{\text{trend}} = \mathbf{t}$ |
| Quadratic | $\mathbf{x}_{\text{trend}} = \mathbf{t}^2$ |
| Exponential | $\mathbf{x}_{\text{trend}} = \frac{2^{\mathbf{t}'}}{1024}$ |
| Logistic | $\mathbf{x}_{\text{trend}} = \frac{1}{1+\exp(-\mathbf{t}')}$ |

$t_i \in [0, 1]$ and $t' \in [-10, 10]$.

*Table 5.* Trend Type Functions

VerbalTS defines 6 types of attribute as summarized in Table 4, including **Trend Types, Trend Directions, Season cycles, Shapelets, High Frequency Components and Multivariables**. Note that only the construction for Synth-M dataset will be assigned with a sampled **Multivariable** attribute. Elaborations on these attributes are as follows.

- **Trend Types and Trend Directions:** Trend component $\mathbf{x}_{\text{trend}}$ of the time series is jointly composed by trend types and trend directions. The trend trajectory types are characterized by 4 functions: linear, quadratic, exponential, and logistic. For each trend trajectory, direction can be either up or down. Complete details of the corresponding functions and value ranges of the trend type are listed in Table 5. For trend directions, up trend indicates $\mathbf{x}_{\text{trend}} = \mathbf{x}_{\text{trend}}$ and down trend indicates $\mathbf{x}_{\text{trend}} = -\mathbf{x}_{\text{trend}}$.

- **Season Cycles:** To simulate season component $\mathbf{x}_{\text{season}}$, synthetic time series data incorporate a set of sinusoidal waves. The periodicity of waves is controlled by parameter $n_{\text{cycle}}$, which takes values from the set $\{0, 1, 2, 4\}$ to represent

different cycles. Season component $\mathbf{x}_{\text{season}}$ can be mathematically formulated as:

$$\mathbf{x}_{\text{season}} = a\sin(2\pi t + \phi), \quad \text{where } t \in [0, n_{\text{cycle}}], n_{\text{cycle}} \in [0, 2^0, 2^1, 2^2], a \sim \mathcal{U}(0.4, 0.6), \phi \sim \mathcal{U}(0, 2\pi) \quad (3)$$

- **Local Shapelets:** Three distinct local shapelets—single peak, sag, and double peaks—are defined to simulate local details in real-world time series which is denoted as $\mathbf{x}_{\text{local}}$. Further details of local shapelets, including morphological definition and stochastic injection, can be referred to Appendix A.1.3.

- **High Frequency Components:** To simulate high-frequency signals in real-world data, synthetic time series incorporate high-frequency components , denoted as $\mathbf{x}_{\text{hf}}$, which are constructed using the same equation 3 as introduced in the Season Cycles except for $n_{\text{cycle}} \in [0, 16, 32, 64], a \sim \mathcal{U}(0.1, 0.3)$.

- **Multivariable:** The multi-variable transfer rules comprise X-axis flip, Y-axis flip, and temporal shifts (forward and backward). Flipping operations flip the time series of the first variable along X-axis or Y-axis to generate time series for the second variable, and the shifting operations translate the time series along the temporal dimension by a shift distance $d_{\text{shift}} \in [20, 40]$. Multivariable attribute is adopted only when generating Synth-M dataset.

### A.1.2. SYNTH-U AND SYNTH-M

As aforementioned, with defined attribute set, time series data can be synthesized accordingly. VerbalTS categorizes attributes into primary and secondary as shown in Table 4. Primary attributes, including Trend Types, Trend Directions and Season Cycles, are shared by all data in the dataset, while secondary attributes, including Local Shapelets and High Frequency Components, are sample-specific in the dataset. Meanwhile noises are common in real-world time series, so noise is added to the time series to increase randomness. The injection of noise is sample-specific and noise is sampled from a Gaussian distribution $\mathbf{x}_{\text{noise}} \sim \mathcal{N}(\mu, \sigma^2), \sigma \in [0.04, 0.06]$.

Utilizing the attribute components predefined above, the synthesis formula for generating Synth-U dataset can be formulated as:

$$\mathbf{x} = \mathbf{x}_{\text{trend}} + \mathbf{x}_{\text{season}} + \mathbf{x}_{\text{local}} + \mathbf{x}_{\text{hf}} + \mathbf{x}_{\text{noise}} \quad (4)$$

Synth-M shares the similar generation formula but in a multivariate setting with the extra attribute Multivariable. Following VerbalTS (Gu et al., 2025), each of Synth-U and Synth-M dataset includes 32000 instances through sampling 1000 samples for 32 (4 Trend Types×2 Trend Directions×4 Season Cycles) combinations of primary attribiutes. Instances are split into training set, validation set and test set in the ratio of 6:1:1.

### A.1.3. LABELING SEGMENTS

Leveraging *local shapelets* attributes, we define four corresponding morphological shapelet labels, including single peak, sag, double peaks and nothing, to simulate real-world time series fine-grained details. Textual description generation utilizes these labels and corresponding templates. A single peak is characterized by a symmetrical linear incline and decline over a time span of length 9, with zeros on both sides and a peak midpoint in the range of [1.0, 1.2]. The sag is defined as the symmetrical shape of a single peak across the x-axis. A double peak is formed by concatenating two single-peak structures. We partition the time series into three segments of equal length. Within each segment, there is a 70% chance of nothing, while the single peak, sag, and double peak each have a 10% probability of being inserted at a random location.

### A.2. Real-World Augmented Datasets

### A.2.1. LLM CAPTION GENERATION

The overall LLM caption generation pipeline is illustrated in Figure 7. For datasets without textual information, we generate high-quality *morphological* caption for these time series by leveraging the capability of Large Language Model (LLM). In our implementation, we choose Gemini-2.5-flash (Comanici et al., 2025) as the LLM. LLM Caption Generation pipeline can be applied to both univariate and multivariate time series. Each variate data of the multivariate time series can also be processed by the pipeline separately if specifically required. To efficiently execute captioning operation, we pre-process time series data as follows:

- **Rounding:** We first round the numerical values of time series to 3 decimal places to reduce massive token cost.

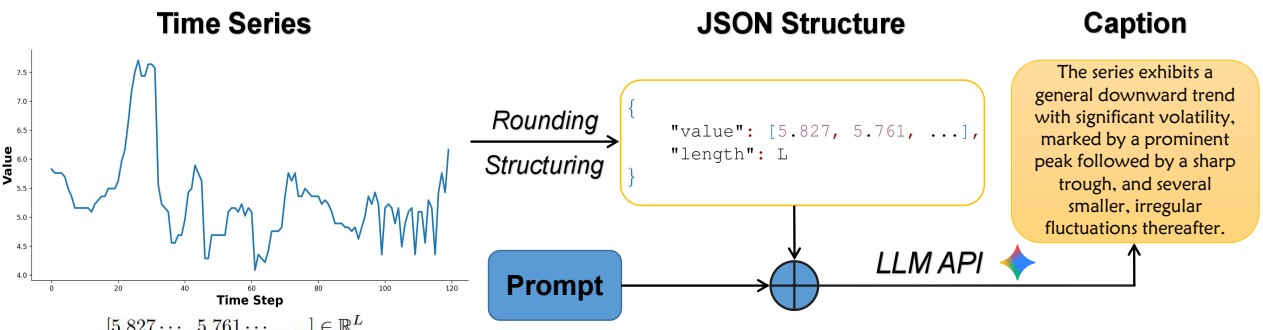

*Figure 7.* Overall Pipeline of LLM Caption Generation

• **Structuring:** We then convert the numpy number sequence into a list, compute its length, and encapsulate them into a JSON object which includes the sequence information and its length information. For processing multivariate time series, sequence information will be a nested list. Note that when dumping JSON for uploading to the LLM, the number list will be serialized into strings.

*Code Snippet 1.* LLM Captioning Generation Prompt for Univariate Time Series

```python
def make_prompt(
    include_context: bool,
    forbid_semantics: bool,
) -> str:
    context_line = (
        "If a channel_description is provided, include that context in your description."
        if include_context
        else "Do NOT mention any domain semantics or variable names."
    )
    if forbid_semantics:
        context_line = "Do NOT mention any domain semantics or variable names."
    prompt = f"""You are a time-series analyst. You will receive a single-channel sequence

    Task:
    - Write a concise intrinsic description focusing on trend, volatility, periodicity,
        and notable peaks/troughs.
    - Mention level shifts if present.
    - Keep it short (1-2 sentences).
    - {context_line}
    Output JSON schema:
    {{
      \"description\": \"<short description>\"
    }}
    Return only JSON (no extra text).
    """.strip()
    return prompt
```

As for prompt engineering, we carefully design the system instruction to prompt LLM to act as a 'Time-Series Analyst' and output structured JSON schema for returning caption. The instructions specifically instruct LLM to pay attention to the morphology of time series, including trend, periodicity, etc. It is worth noticing that mentioning any domain semantics or variable names of time series is not permitted and this can be assured through prompt engineering. An example prompt template for univariate time series can be referred to code snippet 1.

After combining the structured data and prompt into payload, we can obtain the morphological caption of time series by LLM API with rigorous configuration setting.

*Table 6.* **LLM annotation validation via 4-way matching.** Each task contains one text condition and four real time-series candidates. Overall accuracy is computed over the displayed datasets.

| Dataset | Correct / Total | Accuracy (%) |
|---|---|---|
| ETTm1 | 270 / 500 | 54.0 |
| AirQuality Beijing | 331 / 500 | 66.2 |
| Istanbul Traffic | 455 / 500 | 91.0 |
| TelecomTS-Segment | 333 / 500 | 66.6 |
| Overall | 1389 / 2000 | 69.45 |

### A.2.2. LLM ANNOTATION VALIDATION

To audit whether LLM-generated captions are visually aligned with their paired time series, we conduct a 4-way annotation matching validation. For each task, the evaluator is given one text condition and four real time-series candidates sampled from the same dataset: one correct paired series and three distractors. The evaluator must select the candidate that best matches the text condition. We use GPT-5.4[1] as the evaluator and sample 500 tasks per reported dataset. Table 6 shows that the matching accuracy substantially exceeds the 25% chance baseline, suggesting that the generated captions preserve identifiable temporal information.

### A.2.3. ATTRIBUTE VECTOR EXTRACTION

As aforementioned in Section 3.3, to obtain structured attribute vector $c^{\text{attr}}$, Attribute Vector Extraction comprises two procedures: Attribute Schema Discovery and Attribute Vector Value Assignment.

Attribute Schema Discovery Pipeline is developed to, if structured attribute information is not provided in the dataset, discover an appropriate and structured attribute set for all unstructured textual descriptions or captions in the dataset through leveraging the capability of Large Language Model(LLM). In our implementation, we choose Gemini-2.5-flash (Comanici et al., 2025) as the LLM.

The formal objective of the pipeline is to obtain a discrete attribute schema $\mathcal{S}$ mainly consisting of a set of attributes $\mathcal{A} = \{a_1, a_2, \ldots, a_m\}$, where each attribute $a_i$ is defined by a name, a semantic definition, and a set of discrete value options $\mathcal{V}_i$. Each discrete value option has a string name and a corresponding index. We denote dataset as $\mathcal{D}$, mini-batch size as $N$, stability threshold as $K$ and maximum iteration as $T$. An example schema of ETTm1 produced by the pipeline is presented in code snippet 2.

Because prompting LLM to look into the whole text dataset and extract our desired schema is nearly impossible, the discovery process is formulated as an iterative algorithm. Pipeline progressively refines the schema by exposing the model to random batches of data samples. The process continues until the schema converges and stabilizes. The pipeline of algorithm is elaborated as follows:

1. **Sampling:** A mini-batch of text samples $B_t \subset \mathcal{D}$ is drawn without replacement (where possible) to ensure diversity.

2. **Prompt Engineering:** A prompt $P_t$ is constructed containing:

   - **System Instruction:** Defines the task (designing a coarse discrete schema), constraints (e.g., 3–8 values per attribute), and the mandatory inclusion of an 'other' category for robustness.
   - **Current State:** The schema from the previous iteration, $\mathcal{S}_{t-1}$.
   - **Observations:** The current data batch $B_t$.

3. **Inference & Parsing:** The LLM generates a candidate update $\mathcal{S}'_t$. We enforce a strict JSON output schema to ensure syntactic validity. If schema parsing fails, the model is recursively prompted to repair the error JSON.

4. **Canonicalization:** The raw output is normalized (e.g., whitespace stripping) and deduplicated to form $\mathcal{S}_t$.

---

[1] We refer to GPT-5.4 Thinking as described in the OpenAI system card: https://deploymentsafety.openai.com/gpt-5-4-thinking.

*Code Snippet 2.* Example Schema of ETTm1 Dataset A.2.5

```
1  {
2    "scope": "text",
3    "attributes": [
4      {
5        "name": "level_shift_presence",
6        "definition": "Indicates if there are abrupt changes in the average level
             of the series.",
7        "values": [
8          "multiple_shifts", "no_shifts", "other",
9          "single_downward_shift", "single_upward_shift"
10         ]
11       },
12       ......
13     ]
14  }
```

5. **Stability Check:** We compute the hash value $H(\mathcal{S}_t)$. If $H(\mathcal{S}_t) = H(\mathcal{S}_{t-1})$ for $K$ consecutive rounds, the process terminates.

6. **Ending**: If stability check is passed within the maximum iteration limit, the algorithm immediately terminates and the final output schema is preserved. Otherwise the algorithm ends when iteration times reach the maximum limit $T$.

In our implementation, we empirically set $N = 100$, $K = 3$ and $T = 50$. Pseudocode of the algorithm can be referred to 1.

After obtaining the attribute set from the Attribute Schema Discovery Pipeline, we integrate dataset samples' textual descriptions with this attribute set through appropriate prompting and structuring, so as to subsequently feed them into LLM for value assignment to every sample's structured attribute vector.

---

**Algorithm 1** Attribute Schema Discovery Pipeline

---

1: **Input:** Dataset $\mathcal{D}$, Mini-Batch size $N$, Stability threshold $K$, Maximum Iteration $T$
2: **Output:** Optimized Schema $\mathcal{S}^*$
3: $\mathcal{S}_{prev} \leftarrow \emptyset, k_{stable} \leftarrow 0, t_{round} \leftarrow 0, UsedIndices \leftarrow \emptyset$
4: **while** $k_{stable} < K$ and $t_{round} < T$ **do**
5:     $t_{round} \leftarrow t_{round} + 1$
6:     $Indices \leftarrow \text{SAMPLEINDICES}(|\mathcal{D}|, N) \setminus UsedIndices$
7:     $B \leftarrow \{\mathcal{D}[i] \mid i \in Indices\}$
8:     $UsedIndices \leftarrow UsedIndices \cup Indices$
9:     $Prompt \leftarrow \text{BUILDPROMPT}(B, \mathcal{S}_{prev})$
10:     $\mathcal{S}_{raw} \leftarrow \text{LLM}(Prompt)$
11:     $\mathcal{S}_{curr} \leftarrow \text{CANONICALIZE}(\mathcal{S}_{raw})$
12:     **if** $\text{HASH}(\mathcal{S}_{curr}) == \text{HASH}(\mathcal{S}_{prev})$ **then**
13:         $k_{stable} \leftarrow k_{stable} + 1$
14:     **else**
15:         $k_{stable} \leftarrow 0$
16:     **end if**
17:     $\mathcal{S}_{prev} \leftarrow \mathcal{S}_{curr}$
18: **end while**
19: **return** $\mathcal{S}_{prev}$

---

### A.2.4. CLASS LABEL ACQUISITION

After attribute vector extraction, each sample in the dataset is assigned a structured attribute vector. The specific values within each vector define a *unique* attribute combination. By aggregating all such combinations *in the dataset*, we form a set of $N$ distinct entries. Therefore, each attribute vector can be accordingly mapped to an $N$-dimensional one-hot vector through indexing, denoted as its class label $c^{\text{label}}$.

### A.2.5. INDIVIDUAL DATASET DETAILS

**AirQuality Beijing** (Chen, 2017)  Pollutants readings are often included in the air quality reports and are very important for the environment and human society. Collected from 12 nationally-controlled monitoring stations and provided by the Beijing Municipal Environmental Monitoring Center, AirQuality Beijing dataset comprises hourly atmospheric pollutant records integrated with corresponding meteorological data. The time period is from March 1st, 2013 to February 28, 2017.

For each initial raw data sequence from one station, it contains 35,064 observations and is partitioned along the temporal axis into training, validation and test subsets using a ratio of 8:1:1. Then the dataset is obtained by using a sliding window with a sequence length of 24 and a stride of 24 to slice every station's subset data. Finally each sample in the dataset is a multivariate time series with a sequence length of 24 (representing 24-hour window in a day) and 6 variates which correspond to 6 critical air pollutants: PM2.5, PM10, $SO_2$, $NO_2$, CO, and $O_3$. The dataset contains $N = 17,532$ samples in total and the sizes of training, validation, and testing sets are $N_{\text{train}} = 14,025$, $N_{\text{valid}} = 1,753$, and $N_{\text{test}} = 1,754$, respectively. There are 5 attributes extracted from the schema produced by LLM Caption Generation A.2.1 and Attribute Schema Discovery Pipeline A.2.3 and are shown in Table 7.

| Attribute | Value Options | Definition |
|---|---|---|
| Particulate_Matter_Profile | Consistently Low, Low with a Significant Spike, Moderate and Fluctuating, High and Worsening, Consistently High, High with Significant Improvement | The overall 24-hour trend and severity of PM2.5 and PM10. |
| Ozone_Peak_Intensity | Suppressed or No Peak, Moderate Peak, Strong/Very High Peak | The strength of the afternoon ozone (O3) peak. |
| Inverse_Relationship_Strength | No Clear Relationship, Weak Relationship, Strong and Clear Relationship | The clarity of the inverse relationship between O3 and primary pollutants (NO2, CO). |
| Primary_Pollution_Event_Timing | No Specific Event, Morning Peak, Midday/Afternoon Peak, Evening/Nighttime Peak | The main time of day when primary pollutants (PM, CO, NO2) are highest. |

*Table 7.* Attribute Set for Air Quality Profile.

The case visualization of Airquality Beijing time series data with its corresponding condition in three modalities is shown in Figure 8. The statistic details of the number of tokens in the text data are given in Table 8.

| Set | Average Tokens | Median Tokens | Max Tokens | Std. Dev. |
|---|---|---|---|---|
| Training | 220.00 | 212.0 | 611 | 54.85 |
| Validation | 221.61 | 213.0 | 508 | 55.64 |
| Test | 218.08 | 209.0 | 472 | 54.33 |

*Table 8.* Summary of token number statistics for Airquality Beijing dataset.

**TelecomTS** (Feng et al., 2025)  In the context of monitoring complex systems, vast streams of time-series metrics produced by modern enterprises, also known as observability data, can be very important. Unlike conventional datasets that operate on minute-level aggregations, TelecomTS captures high-frequency network dynamics with a sampling interval of 100ms. The data was collected from a real-world 5G network testbed, incorporating Commercial Off-The-Shelf (COTS) user equipment (UE) under varying channel conditions.

The dataset comprises 32,000 samples generated from diverse realistic internet traffic scenarios. Following the ratio of 8:1:1, dataset is empirically split into training, validation and test set with size of 25600, 3200, 3200, respectively. Each sample in the dataset is a multivariate time series with a sequence length of 128 and 18 distinct Key Performance Indicator variates.

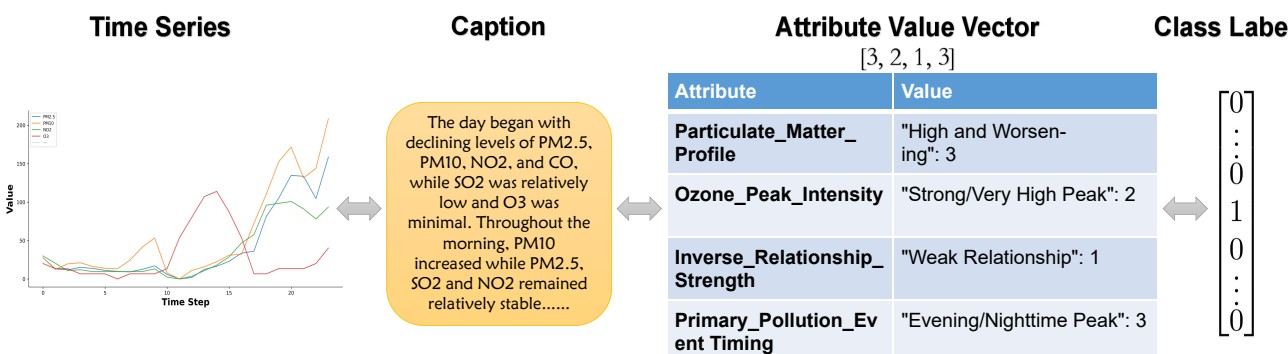

*Figure 8.* Case Visualization of Airquality Beijing Data

In our experiments, we use Reference Signal Received Power (RSRP) and Uplink Signal-to-Noise Ratio (UL_SNR) as a multivariate dataset. In our implementation, we predefine the attribute names as rsrp_seg{i}, ul_snr_seg{i}, $i \in \{1, 2, 3, 4\}$, and then use LLM Caption Generation A.2.1 and Attribute Schema Discovery Pipeline A.2.3 to produce discrete attribute value options. Eventually there are 8 attributes whose value options are identical, including "drop", "drop_recovery", "other", "periodic", "spiky", "stable", "step_change", "trend_down", "trend_up", "volatile". The case visualization of TelecomTS time series data with its corresponding condition in three modalities is shown in Figure 9.

For the Fine-Grained Control experiment in **RQ3**, we prepare an augmented TelecomTS dataset, denoted as TelecomTS-Segment. We partition the sequence into four segments and perform captioning for each segment. The TelecomTS-Segment statistic details of the number of tokens in the text data are given in Table 9.

| Set | Average Tokens | Median Tokens | Max Tokens | Std. Dev. |
|---|---|---|---|---|
| Training | 168.14 | 165.0 | 386 | 31.06 |
| Validation | 167.30 | 164.0 | 302 | 30.07 |
| Test | 168.67 | 165.0 | 323 | 31.27 |

*Table 9.* Summary of token number statistics for Telecomts-Segment dataset.

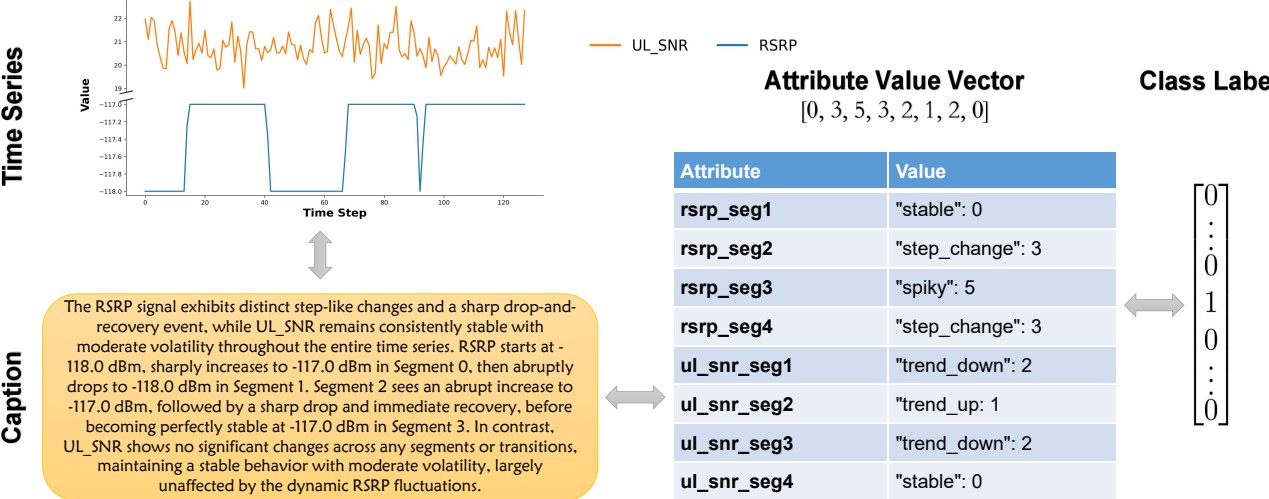

*Figure 9.* Case Visualization of TelecomTS Data

**ETTm1** The ETTm1 dataset is derived from the Electricity Transformer Dataset (ETDataset) (Zhou et al., 2021), which is collected via a real-world platform in partnership with the Beijing Guowang Fuda Science and Technology Development

Company. The time period is from July 2016 to July 2018. There are 7 variates in the raw data: three variates of useful load (HUFL, MUFL, LUFL), three variates of useless load (HULL, MULL, LULL), and Oil Temperature (OT).

The initial raw data sequence, containing 69,680 observations, is partitioned along the temporal axis into training, validation and test subsets using a ratio of 8:1:1. Then the dataset is obtained by using a sliding window with a sequence length of 120 and a stride of 30 to slice each subset. In addition, in our experiments, we decompose these multivariate time series into individual univariate sequences. Finally each sample in the dataset is a univariate time series with a sequence length of 120. The dataset contains $N = 16,275$ samples in total and the sizes of training, validation, and testing sets are $N_{\text{train}} = 13,013$, $N_{\text{valid}} = 1,631$, and $N_{\text{test}} = 1,631$, respectively. There are 5 attributes extracted from the schema produced by LLM Caption Generation A.2.1 and Attribute Schema Discovery Pipeline A.2.3 and are shown in Table 10.

| Attribute | Value Options | Definition |
|---|---|---|
| level_shift_presence | multiple_shifts, no_shifts, other, single_downward_shift, single_upward_shift | Indicates if there are abrupt changes in the average level of the series. |
| overall_trend | downward_trend, fluctuating_trend, mixed_trend, no_clear_trend, other, stable_level, upward_trend | Captures the long-term trajectory and persistent directional movement of the signal. |
| periodicity | absent, other, present, unclear | Assesses the existence of repetitive cycles or patterns occurring at fixed time intervals. |
| prominent_features | minor_fluctuations, other, peaks_and_troughs, peaks_only, troughs_only | Characterizes the most distinct visual landmarks or morphological events within the data. |
| volatility_level | decreasing_volatility, high_volatility, increasing_volatility, low_volatility, mixed_volatility, moderate_volatility, other | Evaluates the intensity and temporal evolution of variance and fluctuations in the series. |

*Table 10.* Attribute Set for ETTm1.

| Set | Average Tokens | Median Tokens | Max Tokens | Std. Dev. |
|---|---|---|---|---|
| Training | 43.37 | 42.0 | 86 | 7.54 |
| Validation | 43.84 | 43.0 | 81 | 8.03 |
| Test | 44.24 | 43.0 | 74 | 7.71 |

*Table 11.* Summary of token number statistics for ETTm1 dataset.

The case visualization of ETTm1 time series data with its corresponding condition in three modalities is shown in Figure 10. The statistic details of the number of tokens in the text data are given in Table 11.

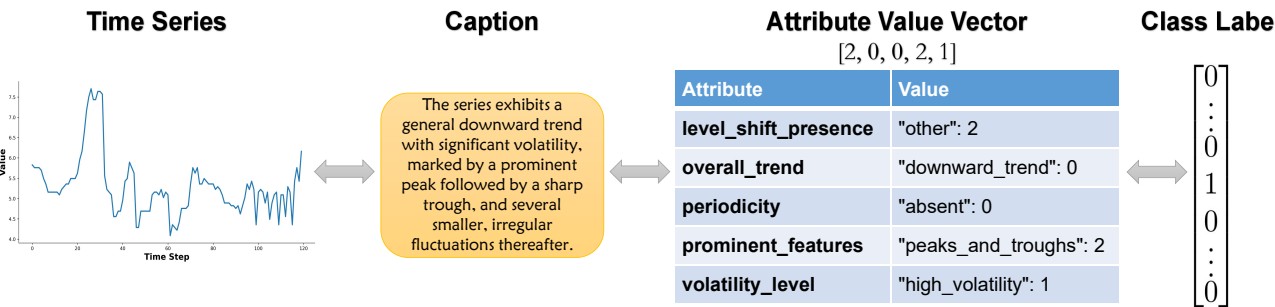

*Figure 10.* Case Visualization of ETTm1 Data

**Istanbul Traffic** (Leo, 2024)    Istanbul Traffic dataset records Istanbul city's traffic index at one-minute intervals. The time series data includes three key variates: a city-wide index (TI) and separate indices for the Asian (TI_An) and European (TI_Av) regions. The time period is from November 2022 to June 2024, with a sampling frequency of one minute and update frequency of a week.

The initial raw data sequence, containing 817,769 observations, is first taken a sample every ten minutes and then partitioned along the temporal axis into training, validation and test subsets using a ratio of 8:1:1. Then the Istanbul Traffic dataset is

obtained by using a sliding window with a sequence length of 144 and a stride of 24 to slice each subset. In addition, in our experiments, we decompose these multivariate time series into individual univariate sequences. Finally each sample in the dataset is a univariate time series with a sequence length of 144. The dataset contains $N = 31,971$ samples in total and the sizes of training, validation, and testing sets are $N_{train} = 25,596$, $N_{valid} = 3,186$, and $N_{test} = 3,189$, respectively. There are 6 attributes extracted from the schema produced by LLM Caption Generation A.2.1 and Attribute Schema Discovery Pipeline A.2.3 and are shown in Table 12.

| Attribute | Value Options | Definition |
|---|---|---|
| extreme_points | absent, other, present | Denotes the existence of significant local maxima (peaks) or minima (troughs) within the signal. |
| level_shifts | absent, other, present_distinct, present_gradual, step_wise | Specifies the occurrence and specific characteristics of sudden transitions in the series' baseline. |
| overall_trend | downward, mixed, other, stable, upward | Represents the predominant long-term trajectory of the data over the entire observation window. |
| periodicity | absent, other, present | Evaluates whether the sequence demonstrates rhythmic or recurring temporal structures. |
| volatility_change | decreasing, increasing, other, stable_volatility | Tracks the evolution of fluctuation intensity, indicating if the variance expands or contracts over time. |
| volatility_level | high, low, moderate, no_volatility, other | Quantifies the overall magnitude of oscillations and noise levels present in the time series. |

*Table 12.* Attribute Set for Istanbul Traffic.

The case visualization of Istanbul Traffic time series data with its corresponding condition in three modalities is shown in Figure 11. The statistic details of the number of tokens in the text data are given in Table 13.

| Set | Average Tokens | Median Tokens | Max Tokens | Std. Dev. |
|---|---|---|---|---|
| Training | 45.64 | 44.0 | 288 | 11.60 |
| Validation | 45.40 | 43.0 | 188 | 11.73 |
| Test | 44.41 | 43.0 | 122 | 11.22 |

*Table 13.* Summary of token number statistics for Istanbul Traffic dataset.

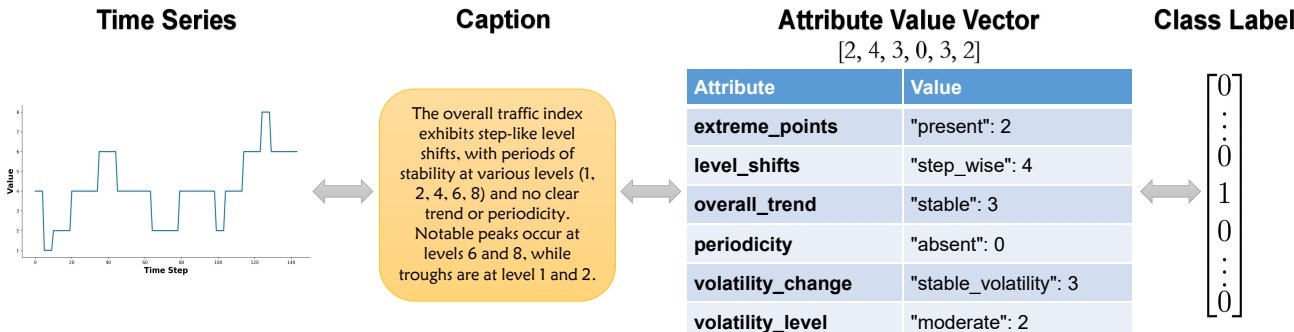

*Figure 11.* Case Visualization of Istanbul Traffic Data

### A.3. Real-World Datasets with Paired Conditions

As aforementioned, with respect to the experiment in **RQ2**, PTB-XL (Wagner et al., 2020) and Weather (Xu et al., 2024) datasets are processed to annotate every time series instance with both a morphological description and conceptual description.

A.3.1. PTB-XL

Electrocardiography (ECG) remains a fundamental diagnostic instrument for evaluating cardiac health. The integration of automated ECG interpretation systems can be rather important and beneficial. PTB-XL is a large dataset of 21,837 clinical 12-lead ECGs from 18,885 patients of 10 second length. After filtering according to our preprocessing pipeline, we obtain 21,799 ECG samples, which are split into training, validation and testing sets with the size of 17,418, 2,183 and 2,198, respectively. Each sample data contains a multivariate time series with the length of 1000 and variate number of 12.

Conceptual information is included in the original dataset for each sample and is used to form conceptual description. Following *Neurokit*2 (Makowski et al., 2021), to annotate samples with more accurate morphological description, instead of prompting LLM to do morphological captioning, we first leverage neuropsychological library *neurokit2* to obtain the physical features of ECG signals, categorize them into single-label attributes, and then generate textual description utilizing the attribute values and there corresponding predefined templates.

Instead of using Attribute Schema Discovery Pipeline A.2.3, the attribute sets for the morphological condition and conceptual condition are obtained from *Neurokit*2 library and patient diagnostic information respectively. The details of these two attribute sets are presented in Table 14 and Table 15.

| Attribute | Value Options | Definition |
|---|---|---|
| rhythm | SR, AFIB, AFLT, STACH, SBRAD, SARRH, PACE, SVARR, BIGU, TRIGU, SVTAC, PSVT, unknown | Primary rhythm code (highest confidence) |
| hr_cat | bradycardia, normal, tachycardia | HR $< 60$ / 60-100 / $> 100$ bpm |
| rr_regularity | regular, mild_irregular, irregular | RR CV $< 0.05$ / 0.05-0.12 / $> 0.12$ |
| qrs_cat | normal, borderline, wide | QRS $< 100$ / 100-120 / $> 120$ ms |
| qtc_cat | normal, borderline, prolonged | QTc $< 450$ / 450-480 / $> 480$ ms |
| st_anterior | normal, mild_elevation, high_elevation, mild_depression, high_depression | ST deviation in V1-V4 |
| st_lateral | normal, mild_elevation, high_elevation, mild_depression, high_depression | ST deviation in I, aVL, V5-V6 |
| st_inferior | normal, mild_elevation, high_elevation, mild_depression, high_depression | ST deviation in II, III, aVF |

*Table 14.* Morphological Condition Attribute Set for PTB-XL

| Attribute | Value Options | Definition |
|---|---|---|
| age_group | young, middle_aged, elderly | Age $< 40$ / 40-65 / $> 65$ |
| sex | male, female | From metadata |
| diagnosis | 43 diagnostic codes (e.g., NORM, IMI, ASMI, LVH, ...) | Primary diagnosis (highest confidence) |
| heart_axis | normal, LAD, RAD, ALAD, ARAD, unknown | From metadata |

*Table 15.* Conceptual Condition Attribute Set for PTB-XL

The case visualization of PTB-XL time series data with its corresponding condition in three modalities is shown in Figure 12. The statistic details of the number of tokens in the morphological and conceptual text data are given in Table 16 and 17 respectively.

A.3.2. WEATHER

Collected at the Max Planck Institute for Biogeochemistry's WS Beutenberg station in Jena, Germany, this comprehensive dataset includes an eight-year climatic observations from 2014 to 2022. There are 21 distinct meteorological parameters

| Set | Average Tokens | Median Tokens | Max Tokens | Std. Dev. |
|---|---|---|---|---|
| Training | 25.94 | 24.0 | 51 | 5.07 |
| Validation | 26.17 | 24.0 | 50 | 5.17 |
| Test | 26.16 | 24.0 | 50 | 4.98 |

*Table 16.* Summary of token number statistics for PTB-XL morphological text data.

| Set | Average Tokens | Median Tokens | Max Tokens | Std. Dev. |
|---|---|---|---|---|
| Training | 16.29 | 16.0 | 29 | 4.37 |
| Validation | 16.13 | 16.0 | 28 | 4.17 |
| Test | 16.19 | 16.0 | 27 | 4.19 |

*Table 17.* Summary of token number statistics for PTB-XL conceptual text data.

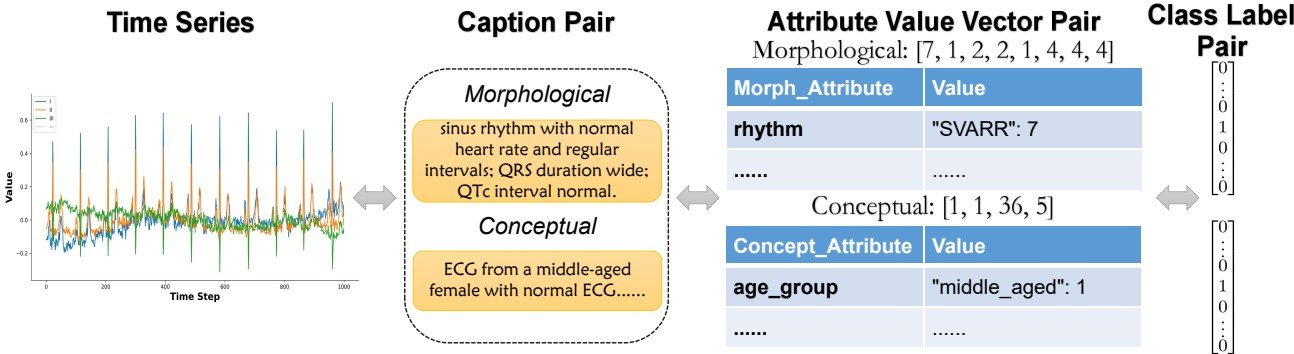

*Figure 12.* Case Visualization of PTB-XL Data

included in the dataset, ranging from atmospheric pressure (p, mbar) to carbon dioxide concentration (CO2, ppm). They are captured at frequent 10-minute intervals with per-second timestamp precision.

The raw data is sliced into 6-hour windows (36 time steps), strictly anchored by existing caption timestamps to ensure data validity. These time series snippets are partitioned in chronological order and in the ratio of 8:1:1, producing training, validation and testing sets with the size of 10489, 1311, 1312 respectively. In addition, given that time series with 21 variates will result in rather long captions produced by LLM, we extract 10 meteorological variates data out and treat them as the new multivariate data used for our experiment. These 10 variates include: temperature (T, degC), wind speed (wv, m/s), wind direction (wd, deg), atmospheric pressure (p, mbar), relative humidity (rh, %), rainfall (rain, mm), rain duration (raining, s), shortwave radiation (SWDR, W/m²), photosynthetically active radiation (PAR, µmol/m²/s) and maximum photosynthetically active radiation (max. PAR, µmol/m²/s).

The conceptual descriptions of the data are human expert weather forecasts obtained from public platforms. The morphological descriptions of the data are generated using Gemini-2.5-flash with appropriate structured payload and prompting. We utilize Attribute Schema Discovery Pipeline A.2.3 to obtain the attribute set for both morphological and conceptual conditions. For the morphological condition, so as to converge faster and produce more morphologically reasonable schema, we predefine each variate of time series as an attribute without specifying variate name or semantic meaning, and eventually obtain identical value options for each attribute, including "flat", "level_shift", "other", "periodic", "spiky", "trend_down" and "trend_up". For the conceptual condition, there are 7 attributes extracted from the produced schema shown in Table 18.

The case visualization of Weather time series data with its corresponding condition in three modalities is shown in Figure 13. The statistic details of the number of tokens in the morphological and conceptual text data are given in Table 19 and Table 20 respectively.

| Attribute | Value Options | Definition |
|---|---|---|
| humidity_level | dry, extremely_high, high, low, medium, moderately_dry, saturated, somewhat_humid, unknown, very_high | A qualitative representation of the ambient moisture concentration in the atmosphere. |
| pressure_level | average, high, low, unknown, very_high, very_low | Reflects the barometric status and atmospheric pressure fluctuations. |
| season | autumn, spring, summer, unknown, winter | Identifies the specific climatological period of the year for the given data. |
| sky_condition | broken_clouds, clear, cloudy, fog, other, partly_cloudy, passing_clouds, precipitation, scattered_clouds, sunny | Characterizes the visual appearance of the firmament and the density of cloud coverage. |
| temperature_level | chilly, cool, high, low, medium, unknown, warm | Provides a non-numeric evaluation of the thermal state of the environment. |
| time_of_day | afternoon, early_morning, evening, morning, night, unknown | Categorizes the observation into broad diurnal temporal segments. |
| wind_strength | calm, fresh, gentle, light, moderate, strong, unknown | Describes the magnitude and kinetic energy of atmospheric air flow. |

*Table 18.* Conceptual Condition Attribute Set for Weather.

| Set | Average Tokens | Median Tokens | Max Tokens | Std. Dev. |
|---|---|---|---|---|
| Training | 142.49 | 140.0 | 276 | 28.91 |
| Validation | 146.09 | 143.0 | 251 | 31.04 |
| Test | 144.45 | 142.0 | 269 | 31.14 |

*Table 19.* Summary of token number statistics for Weather morphological text data.

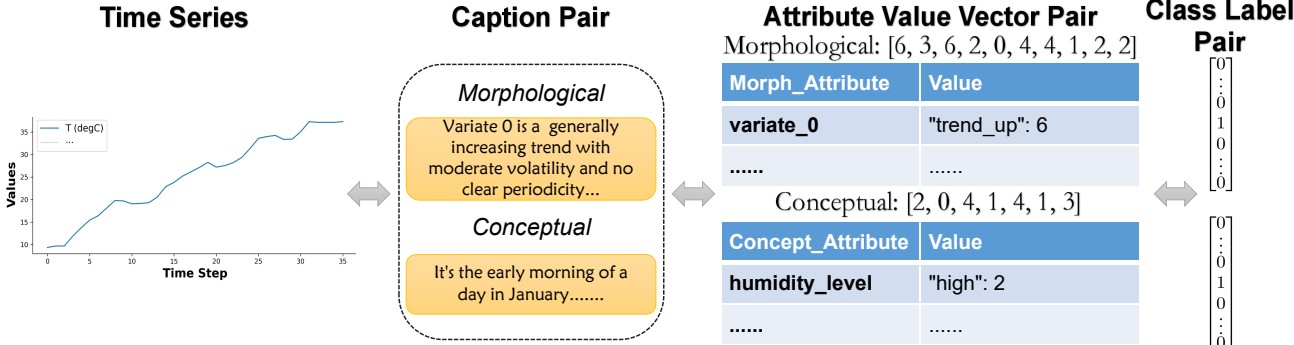

*Figure 13.* Case Visualization of Weather Data

## B. Model Implementation Details

In this section, we provide the implementation details for all evaluated models in ConTSG-Bench. We first describe the unified training configuration shared across all models, followed by model-specific adaptations organized by conditioning modality.

### B.1. General Training Configuration

All experiments are conducted on a single NVIDIA A40 GPU with 48GB memory using full-precision (FP32) training. All datasets are normalized using channel-wise z-score standardization, where each feature dimension is independently standardized to zero mean and unit variance based on training set statistics.

The maximum number of training epochs is set to 700, with early stopping triggered if the validation loss fails to improve for 50 consecutive epochs. For two-stage models (TimeVQVAE (Lee et al., 2023), T2S (Ge et al., 2025), Text2Motion (Guo et al., 2022), and DiffuSETS (Lai et al., 2025)), we allocate 200 epochs for the first stage and 500 epochs for the second stage, with early stopping applied independently to each stage. For adapted Diffusion-TS (Yuan & Qiao, 2024), we use a separate

| Set | Average Tokens | Median Tokens | Max Tokens | Std. Dev. |
|---|---|---|---|---|
| Training | 70.10 | 69.0 | 130 | 10.05 |
| Validation | 73.40 | 73.0 | 125 | 10.64 |
| Test | 72.87 | 73.0 | 112 | 10.58 |

*Table 20.* Summary of token number statistics for Weather conceptual text data.

two-stage procedure following the official implementation's classifier-guidance extension[2]: first train the unconditional diffusion model, then freeze it and train a noisy-sample classifier used for label-guided sampling. All models use the AdamW optimizer with a weight decay of $1 \times 10^{-4}$. We perform grid search over learning rate $\in \{10^{-3}, 10^{-4}\}$, batch size $\in \{32, 64, 128, 256\}$, and learning rate scheduler $\in \{\text{cosine}, \text{linear}, \text{none}\}$, selecting the best configuration based on validation loss.

For diffusion-based models, we adopt DDIM (Song et al., 2021) as the unified sampling framework, except for T2S (Ge et al., 2025) which employs rectified flow with Euler ODE integration. For text-conditioned models, we use Qwen3-Embedding-0.6B (Zhang et al., 2025) as the sentence encoder, except for VerbalTS (Gu et al., 2025) which trains its own text encoder from scratch.

### B.2. Label-Conditioned Models

**TTS-CGAN (Li et al., 2022).** TTS-CGAN employs a Transformer-based conditional GAN architecture (Goodfellow et al., 2020) for class-conditioned time series synthesis. We include it as a representative of GAN-based approaches, offering a fundamentally different training paradigm compared to diffusion-based methods. The model uses WGAN-GP (Gulrajani et al., 2017) training with auxiliary classification losses to encourage class-distinctive pattern generation. The core architecture, including the Transformer-based generator and patch-based discriminator, remains unchanged.

**TimeVQVAE (Lee et al., 2023).** TimeVQVAE leverages vector quantization (Van Den Oord et al., 2017) to compress time series into discrete latent representations, offering an alternative paradigm to continuous diffusion-based methods. The model operates in two stages: (1) a VQ-VAE encodes time series into discrete tokens via time-frequency decomposition with separate codebooks for low and high-frequency components; (2) a MaskGIT-style (Chang et al., 2022) bidirectional Transformer learns the prior distribution over quantized tokens. Our implementation is based on the official repository and preserves all core components. For adaptation to our benchmark, we map discrete attribute combinations onto class labels through Cartesian product encoding. The two-stage training is integrated into our multi-stage framework with automatic weight loading between phases.

**Diffusion-TS (Yuan & Qiao, 2024).** Diffusion-TS is an interpretable diffusion model for general time-series generation, using a Transformer backbone with trend-season decomposition and an $x_0$-prediction objective. The original paper focuses on unconditional generation and extends conditional generation to imputation and forecasting, where the condition is a set of observed historical or partial time-series values. This differs from the ConTSG-Bench setting, where the generator receives only an external condition, such as a class label, attribute vector, or text description, and must generate a complete time series from scratch. We therefore include Diffusion-TS as an *adapted* label-conditioned baseline rather than treating the original paper setting as directly label-conditioned. Our adaptation follows this official implementation extension: we first train the unconditional Diffusion-TS model, then train a separate classifier on noisy time series to predict the target class label. At inference time, the model receives only the discrete class label and uses the classifier gradient to guide the denoising trajectory; no observed historical time-series values are provided.

### B.3. Attribute-Conditioned Models

**TEdit (Jing et al., 2024).** TEdit was originally designed for the task of Time Series Editing (TSE), employing a multi-resolution diffusion architecture with attribute conditioning. We include it as a representative of multi-scale patch-based diffusion approaches. For adaptation to our benchmark, we reformulate TEdit from an editing model to a conditional generation model by removing the two-stage editing procedure (DDIM forward encoding followed by reverse decoding) and

---

[2]The official implementation is available at `https://github.com/Y-debug-sys/Diffusion-TS`.

instead performing standard diffusion-based generation from Gaussian noise. The bootstrap learning component is also removed as it is specific to the editing task. The core architecture remains unchanged, including the multi-resolution patch embedding, residual blocks with time-feature Transformer attention, and the heterogeneous attribute encoder.

**TimeWeaver (Narasimhan et al., 2024).** TimeWeaver is a diffusion-based model designed for generating time series conditioned on heterogeneous metadata. As the original implementation is not publicly available, we provide a faithful reimplementation based on the published paper, following the CSDI-based backbone architecture (Tashiro et al., 2021). For adaptation to our benchmark, we focus on discrete attribute conditioning while omitting support for continuous and time-varying metadata, as these modalities are not present in our evaluation datasets. The core architectural components, including interleaved temporal-feature attention and the metadata fusion mechanism, closely follow the original paper.

**WaveStitch (Shankar et al., 2025).** WaveStitch is designed for synthesizing tabular time series with hierarchical attributes. It employs a diffusion model backbone based on Structured State Spaces (S4) (Gu et al., 2022), which captures long-range temporal dependencies more efficiently than standard attention mechanisms. We include it as a representative of S4-based conditional diffusion architectures. The original implementation uses cyclic encoding that maps temporal attributes (e.g., hour-of-day) to sine-cosine pairs, assuming periodic semantics. For our benchmark, we replace cyclic encoding with learnable embeddings to handle general discrete attributes without assuming specific semantics. The embeddings are injected into each residual block through additive conditioning. The core S4-based diffusion backbone with skip connections and gating mechanism remains unchanged.

### B.4. Text-Conditioned Models

**BRIDGE (Li et al., 2025).** BRIDGE was originally proposed for text-controlled time series generation, serving as a representative of prototype-based diffusion architectures. The core architecture comprises a Domain-Unified Prototyper that extracts latent representations from example time series, and a 1D U-Net (Ronneberger et al., 2015) diffusion backbone with cross-attention between the denoising signal and the extracted prototypes. We adopt the official implementation from the TimeCraft repository and extend the input dimension from univariate to multivariate by modifying only the input channel dimension. At inference time, this model requires an additional example time series to guide generation through prototype extraction. To ensure fair comparison with other models, we randomly sample these example prompts from the training set rather than using oracle references. The core generative mechanism remains unchanged, including the prototype extraction, the cross-attention conditioning pathway, and the classifier-free guidance.

**T2S (Ge et al., 2025).** T2S employs a two-stage latent diffusion framework. In stage one, a convolutional AutoEncoder compresses time series into a fixed 30-position latent space via bilinear interpolation. In stage two, a Diffusion Transformer (Peebles & Xie, 2023) with Adaptive Layer Normalization (AdaLN) performs Rectified Flow (Liu et al., 2023) generation in the latent space. Although the original paper describes a VAE architecture, we follow the official codebase which implements a standard AutoEncoder without the variational component. The original implementation is restricted to univariate series. We extend the encoder and decoder to accept $C$ input channels, where the convolutional layers naturally aggregate cross-variate information.

**VerbalTS (Gu et al., 2025).** VerbalTS was originally proposed for text-conditioned multivariate time series generation. We include it as a representative of multi-scale diffusion approaches with hierarchical text conditioning mechanisms. Unlike other text-conditioned models that rely on pretrained sentence encoders, VerbalTS trains its text encoder from scratch jointly with the generation model, enabling end-to-end optimization of text-time series alignment. Our implementation follows the official repository and preserves all core components, including the multi-scale patch-based architecture and adaptive layer normalization conditioning.

**Text2Motion (Guo et al., 2022).** Text2Motion was originally proposed for generating human motion from natural language descriptions. We include this method as a representative of latent-space autoregressive VAE architectures, offering an alternative generative paradigm to diffusion-based approaches. For adaptation to our benchmark, we replace the original token-level word embeddings (which require part-of-speech annotations and language-specific preprocessing) with pre-computed sentence embeddings from Qwen3-Embedding-0.6B, projected through a learnable linear layer. This modification ensures consistency with other text-conditioned baselines while eliminating external NLP dependencies. The core generative mechanism, including the two-stage training protocol (autoencoder pretraining followed by conditional latent generator

training), remains unchanged.

**DiffuSETS (Lai et al., 2025).**     DiffuSETS was originally designed for 12-lead ECG generation conditioned on clinical text reports and patient-specific attributes. We include it as a representative of the latent diffusion paradigm, employing a two-stage architecture: a VAE (Kingma & Welling, 2019) compresses time series into latent space, followed by a U-Net (Ronneberger et al., 2015) denoiser with text conditioning via cross-attention. For adaptation to our benchmark, we generalize the input channel dimension to support arbitrary multivariate time series and remove patient-specific attributes (age, sex, heart rate), as these attributes are not consistently available across all benchmark datasets. The core latent diffusion mechanism remains unchanged.

# C. Evaluation Metrics Implementation Details

## C.1. Metric Implementation Details

In this section, we provide the mathematical formulations and implementation details for the evaluation metrics introduced in the main text. As aforementioned, we categorize these evaluation metrics into two groups: (1) generation fidelity and (2) condition adherence. Furthermore, depending on whether the metric leverages embeddings, we categorize these evaluation metrics into two groups: (1) statistical and (2) embedding-based. The taxonomy of evaluation metrics are listed in Table 21.

*Table 21.* Taxonomy of Evaluation Metrics for ConTSG

| Category | Statistical | Embedding-based |
|---|---|---|
| **Generation Fidelity** | • MDD (Marginal Distribution Difference)
• ACD (Auto-Correlation Difference)
• SD / KD (Skewness / Kurtosis Difference) | • FID (Fréchet Inception Distance)
• Precision & Recall |
| **Condition Adherence** | • DTW Score (Alignment Distance)
• CRPS (Distribution Calibration and Sharpness) | • CTTP Score (Embedding Dot Product)
• J-FTSD (Joint-Space FID)
• Joint Precision & Recall |

We denote the real data distribution consisting of $n$ pairs of time series and conditions as $D_r = \{(x_1^r, c_1), \ldots, (x_n^r, c_n)\}$, where $x_i^r \in \mathbb{R}^{L \times F}$ represents the real time series with length $L$ and $F$ features, and $c_i$ represents the corresponding condition. Similarly, we denote the dataset of generated time series produced by any arbitrary conditional generation model $G$ and corresponding conditions as $D_g = \{(x_1^g, c_1), \ldots, (x_n^g, c_n)\}$, where $x_i^g = G(c_i)$. We utilize time series encoder and textual description encoder from the trained CTTP model to project time series data and text data into the embedding space respectively. For embedding-based metrics, let $\phi_{\text{ts}}(\cdot)$ denote the time series encoder and $\phi_{\text{text}}(\cdot)$ denote the text encoder.

### C.1.1. GENERATION FIDELITY

These metrics evaluate the quality of generated distribution $x^g$ against the real distribution $x^r$ without considering condition adherence of the generated time series. The primary focus is to quantify the fidelity of generated time series samples and to assess whether the generator accurately captures the marginal distribution of the real time series data.

**Fréchet Inception Distance (Heusel et al., 2017)**     Utilizing the Wasserstein-2 distance between the Gaussian approximations of two embeddings, FID assesses the distance between the feature distributions of real and generated data. We compute the embeddings of time series data $z_i^d = \phi_{\text{ts}}(x_i^d)$ for $d \in \{r, g\}$ and approximate these two embedding distributions as Gaussian Distributions $\mathcal{N}(\mu_{z^r}, \Sigma_{z^r})$ and $\mathcal{N}(\mu_{z^g}, \Sigma_{z^g})$ where $\mu$ and $\Sigma$ are the empirical mean and covariance. Therefore FID metric is formally formulated as:

$$\text{FID}(D_r, D_g) = \|\mu_{z^r} - \mu_{z^g}\|_2^2 + \text{Tr}\left(\Sigma_{z^r} + \Sigma_{z^g} - 2(\Sigma_{z^r}\Sigma_{z^g})^{1/2}\right). \tag{5}$$

where $\mu_{z^d}$ and $\Sigma_{z^d}$ for $d \in \{g, r\}$ are calculated as:

$$\mu_{z^d} = \frac{1}{n}\sum_{i=1}^{n} z_i^d, \quad \Sigma_{z^d} = \frac{1}{n-1}\sum_{i=1}^{n}(z_i^d - \mu_{z^d})(z_i^d - \mu_{z^d})^\top. \tag{6}$$

**Precision & Recall (Kynkäänniemi et al., 2019)** While FID provides a statistic of the distance between feature distributions, Precision and Recall can evaluate the fidelity of generated samples and the diversity of generated distribution respectively. These metrics rely on constructing an explicit non-parametric representations of the manifolds of real and generated data in the feature space. Here we use $\Phi_r = \{\phi_{\text{ts}}(x) \mid x \in D_r\}$ and $\Phi_g = \{\phi_{\text{ts}}(x) \mid x \in D_g\}$ to denote the sets of embeddings for real and generated time series.

The manifold of $\Phi$ is approximated by forming a hypersphere for each sample embedding $\mathbf{v} \in \Phi$, where the radius of hypersphere is determined by the distance to its $k$-th nearest neighbor in $\Phi$. As such, a binary indicator function $f(\mathbf{q}, \Phi)$ can be defined to determine whether a query sample $\mathbf{q}$ in the embedding space lies within the manifold of $\Phi$:

$$f(\mathbf{q}, \mathbf{\Phi}) = \begin{cases} 1, & \text{if } \exists \phi' \in \mathbf{\Phi} \text{ s.t. } \|\mathbf{q} - \phi'\|_2 \leq \|\phi' - \text{NN}_k(\phi', \mathbf{\Phi})\|_2 \\ 0, & \text{otherwise,} \end{cases} \tag{7}$$

where $\text{NN}_k(\phi', \mathbf{\Phi})$ returns $k$-th nearest feature vector of $\phi'$ from set $\Phi$. Intuitively, $f(\mathbf{q}, \Phi)$ indicates if a sample in the embedding space falls within the $k$-NN hypersphere of at least one sample in the reference set $\mathbf{\Phi}$. We adopt $k = 5$ in our experiments. Now Precision and Recall metrics can be formally formulated leveraging manifold and function $f(\mathbf{q}, \Phi)$. Precision measures the fraction of generated samples that fall within the estimated manifold of the real data. A high precision indicates that the generated samples are realistic and resemble the training data. Precision is mathematically formulated as:

$$\text{Precision}(\Phi_r, \Phi_g) = \frac{1}{|\Phi_g|} \sum_{\mathbf{g} \in \Phi_g} f(\mathbf{g}, \Phi_r). \tag{8}$$

Recall measures the fraction of real samples that fall within the estimated manifold of the generated data. A high recall indicates that the generator covers diversity of the true distribution without mode collapse. Recall is mathematically formulated as:

$$\text{Recall}(\Phi_g, \Phi_r) = \frac{1}{|\Phi_r|} \sum_{\mathbf{r} \in \Phi_r} f(\mathbf{r}, \Phi_g). \tag{9}$$

**Distributional Statistics** Following TSGBench (Ang et al., 2023), to assess whether the generated time series captures fine-grained temporal properties, we utilize four key statistical measures including marginal distribution difference (MDD) (Ni et al., 2021), auto-correlation difference (ACD) (Lai et al., 2018), skewness difference (SD), and kurtosis difference (KD). For these four metrics, smaller values indicate more similar statistical distributions between the generated time series data and the real-world data.

Marginal Distribution Difference (MDD) measures how closely the distributions of the real and generated series align. The probability density functions of real and generated time series data are approximated using empirical histograms with $B$ bins for each dimension and time step. In our experiments, we discretize the continuous time series values into histograms with $B = 32$ bins whose boundaries are determined by the dynamic range in the training set. To ensure consistent evaluation, any values in the generated or test sets falling outside this pre-defined range are assigned to the first or last bin. Let $p_r(b)$ and $p_g(b)$ denote the probability mass in the $b$-th bin for real and generated data, respectively. MDD is therefore defined as the average absolute difference between two histograms across bins:

$$\text{MDD}(D_r, D_g) = \frac{1}{B} \sum_{b=1}^{B} |p_r(b) - p_g(b)|. \tag{10}$$

Auto-Correlation Difference (ACD) evaluates the preservation of temporal dependencies through computing the difference of autocorrelation of real and generated time series. Autocorrelation coefficient $\rho_k$ at lag $k$ for a single time series $x$ from dataset $D$, which measures the linear relationship between observations separated by $k$ time steps, is calculated as:

$$\rho_k(x) = \frac{\sum_{t=1}^{L-k}(x_t - \mu)(x_{t+k} - \mu)}{\sum_{t=1}^{L}(x_t - \mu)^2}. \tag{11}$$

where $\mu$ is the mean of this time series. Therefore the average lag $k$ autocorrelation profile for the entire dataset $D$ is denoted as $\bar{\rho}_k(D) = \frac{1}{|D|} \sum_{x \in D} \rho_k(x)$. As such, ACD is formally defined as the Euclidean distance between the mean

autocorrelation profile vectors of real and generated data denoted as $\bar{\boldsymbol{\rho}}^r$ and $\bar{\boldsymbol{\rho}}^g$:

$$\text{ACD}(D_r, D_g) = \|\bar{\boldsymbol{\rho}}^r - \bar{\boldsymbol{\rho}}^g\|_2 = \sqrt{\sum_{k=1}^{L-1} (\bar{\rho}_k(D_r) - \bar{\rho}_k(D_g))^2}. \tag{12}$$

Skewness characterizes difference in the asymmetry of the data distribution around its mean. Given the mean (standard deviation) of the real time series as $\mu_r(\sigma_r)$ and the generated time series as $\mu_g(\sigma_g)$, Skewness Difference (SD) is formulated as the difference of Skewness coefficients between the real and generated data distributions:

$$\text{SD}(D_r, D_g) = |\mathcal{S}(D_r) - \mathcal{S}(D_g)|, \quad \text{where } \underset{i \in \{r,g\}}{\mathcal{S}(D_i)} = \mathbb{E}\left[\left(\frac{D_i - \mu_i}{\sigma_i}\right)^3\right]. \tag{13}$$

Kurtosis measures the presence of outliers in a distribution, revealing extreme deviations from the mean. Given the mean (standard deviation) of the real time series as $\mu_r(\sigma_r)$ and the generated time series as $\mu_g(\sigma_g)$, Kurtosis Difference (KD) is formulated as the difference of Kurtosis coefficients between the real and generated data distributions:

$$\text{KD}(D_r, D_g) = |\mathcal{K}(D_r) - \mathcal{K}(D_g)|, \quad \text{where } \underset{i \in \{r,g\}}{\mathcal{K}(D_i)} = \mathbb{E}\left[\left(\frac{D_i - \mu_i}{\sigma_i}\right)^4\right]. \tag{14}$$

### C.1.2. CONDITION ADHERENCE

Since the aforementioned metrics exclusively focus on evaluating the generation fidelity and marginal distribution of time series features, they are insensitive to the condition adherence. They fail to penalize realistic samples that are mismatched with their corresponding conditions, making them insufficient for assessing conditional generation. Consequently, we introduce condition-adherence metrics to fairly evaluate whether the generated sample $x_i^g$ adhere to the specific condition $c_i$ and preserve the joint distribution properties.

**CTTP Score (Gu et al., 2025)**  Similar to CLIP Score (Radford et al., 2021), CTTP Score evaluates the sample-level alignment between the generated time series and its corresponding text description through the CTTP model, which aligns time series and text descriptions in a shared embedding space. Specifically, the CTTP model encodes the generated time series $x_i^g$ into $z_{x_i^g} = \psi_{\text{ts}}(x_i^g)$ and the text condition $c_i$ into $z_{c_i} = \psi_{\text{text}}(c_i)$. Following VerbalTS (Gu et al., 2025), the per-sample CTTP score is defined as the dot product between $z_{x_i^g}$ and $z_{c_i}$:

$$\text{CTTP}(x_i^g, c_i) = z_{x_i^g} \cdot z_{c_i}, \tag{15}$$

where $\cdot$ denotes the dot product. We report the average score over all generated samples:

$$\text{CTTP Score} = \frac{1}{n} \sum_{i=1}^{n} \text{CTTP}(x_i^g, c_i). \tag{16}$$

A higher CTTP score shows better semantic alignment between the time series and its corresponding textual description, indicating stronger condition adherence. Because this score is an embedding dot product rather than normalized cosine similarity, it is not constrained to the interval $[-1, 1]$. In our implementation, we compute the CTTP Score between time series and textual descriptions, rather than directly between time series and other condition modalities (e.g., numerical forecasting horizons or categorical labels). This design choice is motivated by our dataset construction process, where each sample is paired with a textual description that semantically encapsulates the conditioning information. As a result, the textual description can serve as a unified semantic proxy for evaluating condition adherence across different conditioning modalities within our benchmark.

**Joint Frechet Time Series Distance (J-FTSD) (Narasimhan et al., 2024)**  J-FTSD extends the FID to the joint feature space of time series and condition to reflect condition adherence. Let $\mathcal{X} \subseteq \mathbb{R}^{L \times F}$ be the domain of time series, $\mathcal{C}$ be the domain of conditions. Denote the dimension of embedding space of time series and text conditions as $k$. We define the **Joint Feature Space** $\mathcal{Z}$ as the image of the Cartesian product $\mathcal{X} \times \mathcal{C}$ under the joint projection $\Phi$:

$$\mathcal{Z} \triangleq \left\{ z \in \mathbb{R}^{2k} \mid z = \phi_{\text{ts}}(x) \oplus \phi_{\text{text}}(c), \quad \forall x \in \mathcal{X}, \forall c \in \mathcal{C} \right\}, \quad \Phi(x, c) = \phi_{\text{ts}}(x) \oplus \phi_{\text{text}}(c). \tag{17}$$

Here, $\oplus$ denotes concatenation, and $\Phi(x, c)$ maps the pair into the $2d$-dimensional vector space. Consequently we can construct joint space embeddings $z_i^d$ for $d \in \{g, r\}$ by concatenating the time series and condition embeddings:

$$z_i^d = \Phi(x_i^d, c_i) \quad \forall i : (x_i^d, c_i) \in D_d, \quad d \in \{g, r\}. \tag{18}$$

As such, J-FTSD is formally formulated as the Fréchet distance between these joint distributions on joint feature space. The metric function J-FTSD$(D_g, D_r)$ adopts the same formulation as formulae 5 and 6 but operate on the joint feature space 17.

**Joint Precision & Recall** Complementary to J-FTSD, we extend Precision & Recall metrics on the joint feature space of time series and condition. The formulae for these two metrics remain the same as equations 7, 8 and 9 but operate on the joint feature space 17.

**Dynamic Time Warping (DTW)** DTW measures the alignment distance between the generated sample $x_i^g$ and the ground truth reference $x_i^r$ associated with the same condition. Given two time series sequences $X = (x_1, \ldots, x_N)$ and $Y = (y_1, \ldots, y_M)$, DTW computes the optimal sequence alignment by minimizing the cumulative distance between aligned points. Under defined local distance measure $d(x_i, y_j)$, typically the Euclidean distance $\|x_i - y_j\|_2$, DTW distance is computed using dynamic programming to find the minimum cumulative cost matrix $D_p \in \mathbb{R}^{N \times M}$. Each element $D_p(i, j)$ represents the minimum distance between the subsequences $X_{1:i}$ and $Y_{1:j}$, governed by the following recurrence relation:

$$D_p(i, j) = d(x_i, y_j) + \min \begin{cases} D_p(i - 1, j) & \text{(insertion)} \\ D_p(i, j - 1) & \text{(deletion)} \\ D_p(i - 1, j - 1) & \text{(match)} \end{cases} \tag{19}$$

where boundary conditions are $D_p(0, 0) = 0$ and $D_p(i, 0) = D_p(0, j) = \infty$ for $i, j > 0$. The final DTW distance is given by $D_p(N, M)$ and thus we denote operator DTW$(\cdot, \cdot)$:

$$\text{DTW}(X, Y) = D_p(N, M), \quad \text{where } |X| = N, |Y| = M. \tag{20}$$

Utilizing operator DTW$(\cdot, \cdot)$, for each sample in the real-world dataset $D_r$ with condition $c_i$, we generate $K$ time series $\{x_{i,k}^g\}_{k=1}^K$, take the minimum as sample score so as to leverage best-of-$K$ strategy to account for the stochasticity of generation, and eventually average over samples:

$$\text{DTW Score} = \frac{1}{n} \sum_{i=1}^n \min_{k \in \{1 \ldots K\}} \text{DTW}(x_i^r, x_{i,k}^g), \quad \text{where } x_{i,k}^g = G(c_i). \tag{21}$$

**Continuous Ranked Probability Score (CRPS) (Ansari et al., 2024)** CRPS generalizes the Mean Absolute Error (MAE) to the probabilistic setting by measuring the distance between the empirical cumulative distribution of generated samples and a reference value, therefore can better assess both the calibration and sharpness of the generation distribution.

Let $F_{cd}$ denote the cumulative distribution function (CDF) of the generated probabilistic forecast, and $y$ be the ground truth observation value. The CRPS measures the integrated squared difference between the forecast CDF and the Heaviside step function of the observation, and can be mathematically formulated as:

$$\text{CRPS}(F_{cd}, y) = \int_{-\infty}^{\infty} (F_{cd}(z) - \mathbb{I}(z \geq y))^2 \, dz. \tag{22}$$

where $\mathbb{I}(\cdot)$ is the indicator function. Since generative models produce samples rather than CDF, this integral is approximated by leveraging the key property of CRPS showed by (Gneiting & Raftery, 2007) which is mathematically formulated as:

$$\text{CRPS}(F, y) = \mathbb{E}_{X \sim F}[|X - y|] - \frac{1}{2}\mathbb{E}_{X, X^* \sim F}[|X - X^*|], \tag{23}$$

where $X$ and $X^*$ are independent random variables drawn from distribution $F$ and $\mathbb{E}[\cdot]$. Leveraging the strategy of aggregating over all $K$ samples, let $\hat{y}_t = \{\hat{y}_t^{(1)}, \ldots, \hat{y}_t^{(K)}\}$ denote the set of $K$ generated samples at time step $t$ for one

instance, and $y_t$ be the corresponding real-world data value. Utilizing the approximation equation 23, the instance-level CRPS at time $t$ is computed as:

$$\widehat{\text{CRPS}}(t) = \frac{1}{K} \sum_{i=1}^{K} |\hat{y}_t^{(i)} - y_t| - \frac{1}{2K^2} \sum_{i=1}^{K} \sum_{j=1}^{K} |\hat{y}_t^{(i)} - \hat{y}_t^{(j)}|. \tag{24}$$

The final metric for one instance is computed as the average $\widehat{\text{CRPS}}$ across all time steps.

## C.2. CTTP Model Training

The Contrastive Text-Time Series Pretraining (CTTP) is used to bridge time series representation and condition representation. In the evaluation setup of our benchmark, we train CTTP model for *each* dataset so as to obtain time-series encoder $\phi_{ts}$ and text encoder $\phi_{text}$. These encoders after training can be utilized to produce corresponding embedding representation which is critical for embedding-based metrics introduced in Appendix C.1.

Conceptually similar to the CLIP model (Radford et al., 2021), CTTP training leverages contrastive learning technique to align time series data and associated textual conditions. Specifically, let $\mathbf{X} \in \mathbb{R}^{B \times K \times L}$ denote a batch of $B$ time-series samples and $\mathbf{C} \in \mathbb{N}^{B \times M}$ represent the associated tokenized textual description data. Following VerbalTS (Gu et al., 2025), we utilize PatchTST (Nie et al., 2023) as the temporal encoder $\psi_{ts}(\cdot)$ and Long-clip (Zhang et al., 2024a) as the tokenizer and text encoder $\psi_{text}(\cdot)$. As shown in Algorithm 2, these encoders project the raw data into a unified $d$-dimensional embedding space, yielding $\mathbf{Z_x} \in \mathbb{R}^{B \times d}$ and $\mathbf{Z_c} \in \mathbb{R}^{B \times d}$ respectively. Then we compute the similarity score matrix $\mathbf{S} \in \mathbb{R}^{B \times B}$ of $\mathbf{Z_x}$ and $\mathbf{Z_c}$. Subsequently, cross-entropy loss of similarity matrix $\mathbf{S}$ against ground truth label identity matrix $\mathbf{I} \in \mathbb{R}^{B \times B}$ can be calculated across two dimensions. The optimization target is to minimize a bidirectional cross-entropy loss that maximizes the similarity between true pairs while penalizing all mismatched pairs.

---

**Algorithm 2** CTTP
___
1: **Input:** Batch of aligned time-series data $\mathbf{X}$ and textual descriptions $\mathbf{C}$.
2: **Output:** Bidirectional cross-entropy loss $\mathcal{L}_{\text{total}}$.
3: **// Embeddings Extraction**
4: $\mathbf{Z_x} \leftarrow \phi_{ts}(\mathbf{X})$
5: $\mathbf{Z_c} \leftarrow \phi_{text}(\mathbf{C})$
6:
7: **// Similarity Computation**
8: $\mathbf{S} \leftarrow \text{Similarity}(\mathbf{Z_x}, \mathbf{Z_c})$
9:
10: **// Compute Cross-Entropy Loss**
11: $\mathcal{L}_{\text{ts\_align}} \leftarrow \text{CrossEntropy}(\mathbf{S}, \mathbf{I}, \dim = 1)$
12: $\mathcal{L}_{\text{txt\_align}} \leftarrow \text{CrossEntropy}(\mathbf{S}, \mathbf{I}, \dim = 0)$
13:
14: **// Compute Bidirectional Cross-Entropy Loss**
15: $\mathcal{L}_{\text{total}} \leftarrow \frac{1}{2}(\mathcal{L}_{\text{ts\_align}} + \mathcal{L}_{\text{txt\_align}})$
16: **return** $\mathcal{L}_{\text{total}}$

---

We train the CTTP model with batch size $B = 256$, learning rate $1 \times 10^{-4}$, early stopping patience of 50 epochs, and a maximum of 500 epochs. Table 22 reports the validation accuracy for each dataset. Note that under a batch size of 256, the random baseline accuracy is merely $1/256 \approx 0.39\%$. Even the lowest observed accuracy (16.09% on TelecomTS-Segment) exceeds this baseline by over $40\times$, indicating that CTTP successfully learns meaningful alignment between time series and textual conditions. The variation in accuracy across datasets reflects inherent differences in task difficulty, such as the discriminability of textual descriptions and the complexity of temporal patterns.

### C.2.1. PAIRWISE VALIDATION OF CTTP SCORES

Validation accuracy verifies that CTTP learns discriminative text–time-series alignment, but it does not by itself test whether CTTP score differences faithfully reflect relative generation quality. We therefore conduct a pairwise preference validation

*Table 22.* CTTP validation accuracy across different datasets. The values represent the maximum validation accuracy achieved during training.

| Dataset | Acc (%) | Dataset | Acc (%) |
|---|---|---|---|
| TelecomTS | 37.19 | ETTm1 | 16.59 |
| TelecomTS-Segment | 16.09 | Istanbul Traffic | 17.45 |
| PTB-XL (Morph.) | 28.31 | Synth-M | 98.35 |
| PTB-XL (Concept) | 33.85 | Synth-U | 92.52 |
| Weather (Morph.) | 20.52 | Air Quality | 18.37 |
| Weather (Concept) | 20.90 | | |

to examine whether CTTP score gaps are aligned with visual judgments of condition adherence.

We select three representative datasets, ETTm1, Istanbul Traffic, and TelecomTS-Segment, and evaluate five generation models with 20 generated samples per dataset. For each selected condition instance and each pair of models, we render the two generated time series as side-by-side plots together with the same conditioning text. An LLM-based visual evaluator, GPT-5.4, is asked to choose which generated time series better matches the conditioning text. We then compare this preference direction with the CTTP score direction, i.e., whether the model preferred by the evaluator also receives the higher CTTP score. Tied or invalid comparisons are excluded, resulting in 501 non-tie pairwise judgments.

Table 23 reports agreement after stratifying pairs by the absolute CTTP score gap. Agreement increases as the CTTP gap becomes larger. When $|\Delta\mathrm{CTTP}| \geq 1.0$, CTTP agrees with the LLM-based preference in 165 out of 273 comparisons, reaching 60.4% agreement with statistical significance ($p < 0.001$). This monotonic trend suggests that CTTP score differences carry meaningful preference-aligned signal: small score gaps are naturally ambiguous, while larger gaps more reliably indicate visible differences in condition adherence.

*Table 23.* **Pairwise validation of CTTP score differences.** We compare the CTTP score direction with an LLM-based visual preference judgment.

| CTTP score gap | Agree / Total | Agreement | $p$-value |
|---|---|---|---|
| $|\Delta\mathrm{CTTP}| \geq 0.5$ | 208 / 360 | 57.8% | 0.002 |
| $|\Delta\mathrm{CTTP}| \geq 1.0$ | 165 / 273 | 60.4% | 0.0003 |
| $|\Delta\mathrm{CTTP}| \geq 2.0$ | 106 / 175 | 60.6% | 0.003 |
| $|\Delta\mathrm{CTTP}| \geq 3.0$ | 76 / 120 | 63.3% | 0.002 |

## D. Additional Experimental Results

### D.1. Inference Efficiency

To assess deployment cost across model families, we profile inference on Synth-U using a single NVIDIA A40 GPU with 10 generated samples per condition. Table 24 reports the number of parameters, per-sample latency, and peak GPU memory usage for each model. Most models fit within 2GB peak VRAM under this setting, while Diffusion-TS requires higher memory because its guided sampling keeps the diffusion backbone and auxiliary classifier active. Latency differs substantially across architectures: TTSCGAN is fastest, whereas iterative diffusion-style generators such as Diffusion-TS and Bridge are slower. Model size alone does not determine runtime: for example, Text2Motion has 32.9M parameters but lower latency than several smaller models.

### D.2. Main Results (RQ1)

#### D.2.1. COMPLETE RESULTS

This section provides the complete per-dataset evaluation results for all benchmark models, as summarized in Section 4.1. Tables 25–34 report detailed metric scores across all ten dataset configurations, including both generation fidelity metrics (ACD, SD, KD, MDD, FID, Precision, Recall) and condition adherence metrics (J-FTSD, Joint Precision, Joint Recall, CTTP Score). Each entry reports the mean and standard deviation over three independent runs. Bold values indicate the

*Table 24.* **Inference efficiency on Synth-U.** All models are profiled on a single NVIDIA A40 GPU with 10 generated samples per condition.

| Model | #Params | Latency (ms/sample) | Peak VRAM (GB) |
|---|---|---|---|
| Bridge | 22.1M | 13.47 | 1.87 |
| DiffuSETS | 30.6M | 0.69 | 0.68 |
| Diffusion-TS | 1.1M | 55.92 | 4.50 |
| T2S | 971.9K | 3.90 | 1.08 |
| TEdit | 1.2M | 0.86 | 0.51 |
| Text2Motion | 32.9M | 0.22 | 0.20 |
| TimeVQVAE | 2.0M | 8.79 | 0.04 |
| TimeWeaver | 326.5K | 0.58 | 0.24 |
| TTSCGAN | 246.8K | 0.03 | 0.13 |
| VerbalTS | 52.5M | 5.43 | 0.26 |
| WaveStitch | 472.5K | 4.71 | 1.04 |

best-performing model for each metric on each dataset.

*Table 25.* Evaluation results for Synth-U dataset

| Model | ACD | SD | KD | MDD | FID | J-FTSD | Precision | Recall | Joint Precision | Joint Recall | CTTP Score |
|---|---|---|---|---|---|---|---|---|---|---|---|
| Bridge | $\mathbf{0.0314}_{\pm0.0116}$ | $0.0974_{\pm0.0775}$ | $0.6417_{\pm0.0562}$ | $0.0282_{\pm0.0001}$ | $50.1814_{\pm3.8816}$ | $55.1401_{\pm2.8398}$ | $0.5906_{\pm0.0411}$ | $0.0745_{\pm0.0104}$ | $0.8228_{\pm0.0404}$ | $0.4752_{\pm0.0240}$ | $22.9919_{\pm0.7649}$ |
| DiffuSETS | $0.0552_{\pm0.0113}$ | $0.1404_{\pm0.1382}$ | $0.8331_{\pm0.8645}$ | $0.0203_{\pm0.0050}$ | $41.5913_{\pm8.8965}$ | $46.5630_{\pm10.9944}$ | $0.5842_{\pm0.0770}$ | $\mathbf{0.2054}_{\pm0.0257}$ | $0.8427_{\pm0.1608}$ | $0.6600_{\pm0.1285}$ | $24.6628_{\pm4.6115}$ |
| Diffusion-TS | $0.0359_{\pm0.0153}$ | $0.0424_{\pm0.0331}$ | $1.3125_{\pm0.0151}$ | $0.0463_{\pm0.0017}$ | $99.2668_{\pm9.9346}$ | $109.1793_{\pm9.2668}$ | $\mathbf{0.7969}_{\pm0.0309}$ | $0.0017_{\pm0.0018}$ | $0.6274_{\pm0.0495}$ | $0.0734_{\pm0.0206}$ | $11.4218_{\pm1.9377}$ |
| T2S | $0.0489_{\pm0.0098}$ | $0.4960_{\pm0.4106}$ | $0.7833_{\pm0.4301}$ | $0.0358_{\pm0.0165}$ | $144.4827_{\pm27.3339}$ | $156.3209_{\pm27.2652}$ | $0.2330_{\pm0.1987}$ | $0.0000_{\pm0.0000}$ | $0.2862_{\pm0.0746}$ | $0.0140_{\pm0.0109}$ | $10.3251_{\pm4.3402}$ |
| TEdit | $0.0658_{\pm0.0015}$ | $0.1632_{\pm0.1069}$ | $0.2876_{\pm0.1147}$ | $0.0216_{\pm0.0056}$ | $49.1590_{\pm7.0552}$ | $59.9647_{\pm6.4816}$ | $0.5448_{\pm0.0081}$ | $0.1217_{\pm0.0344}$ | $0.7358_{\pm0.0081}$ | $0.3991_{\pm0.0576}$ | $20.3756_{\pm0.7434}$ |
| Text2Motion | $0.0821_{\pm0.0067}$ | $0.0715_{\pm0.0413}$ | $0.2046_{\pm0.0673}$ | $\mathbf{0.0150}_{\pm0.0029}$ | $58.5985_{\pm12.4265}$ | $66.6559_{\pm11.6903}$ | $0.5509_{\pm0.0702}$ | $0.0562_{\pm0.0302}$ | $0.7030_{\pm0.0355}$ | $0.3052_{\pm0.0937}$ | $17.6067_{\pm1.7010}$ |
| TimeVQVAE | $0.0800_{\pm0.0016}$ | $0.0457_{\pm0.0244}$ | $0.7758_{\pm0.0165}$ | $0.0245_{\pm0.0012}$ | $75.7424_{\pm2.1271}$ | $83.9404_{\pm2.0995}$ | $0.7228_{\pm0.0130}$ | $0.0170_{\pm0.0011}$ | $0.7329_{\pm0.0121}$ | $0.2192_{\pm0.0138}$ | $16.0165_{\pm0.2354}$ |
| TimeWeaver | $0.0735_{\pm0.0098}$ | $0.0469_{\pm0.0182}$ | $0.5210_{\pm0.1770}$ | $0.0207_{\pm0.0058}$ | $55.4405_{\pm14.8883}$ | $65.8629_{\pm14.1253}$ | $0.6252_{\pm0.0406}$ | $0.1102_{\pm0.0759}$ | $0.7436_{\pm0.0266}$ | $0.3437_{\pm0.1154}$ | $19.0760_{\pm1.8983}$ |
| TTSCGAN | $0.2849_{\pm0.0016}$ | $0.1043_{\pm0.0575}$ | $\mathbf{0.1352}_{\pm0.0728}$ | $0.0233_{\pm0.0049}$ | $119.9663_{\pm24.7832}$ | $132.8113_{\pm21.8516}$ | $0.1867_{\pm0.0720}$ | $0.0000_{\pm0.0000}$ | $0.1907_{\pm0.0009}$ | $0.0290_{\pm0.0273}$ | $9.3852_{\pm2.2275}$ |
| VerbalTS | $0.0601_{\pm0.0037}$ | $\mathbf{0.0317}_{\pm0.0252}$ | $0.3739_{\pm0.0601}$ | $0.0156_{\pm0.0020}$ | $\mathbf{37.9622}_{\pm1.8307}$ | $\mathbf{41.3352}_{\pm2.0161}$ | $0.6828_{\pm0.0204}$ | $0.2021_{\pm0.0208}$ | $\mathbf{0.9524}_{\pm0.0080}$ | $\mathbf{0.7239}_{\pm0.0285}$ | $\mathbf{27.2469}_{\pm0.5870}$ |
| WaveStitch | $0.0491_{\pm0.0262}$ | $0.1236_{\pm0.1098}$ | $0.3398_{\pm0.1632}$ | $0.0354_{\pm0.0135}$ | $54.6382_{\pm15.5595}$ | $66.8958_{\pm16.9945}$ | $0.4412_{\pm0.1738}$ | $0.1430_{\pm0.1044}$ | $0.6172_{\pm0.1921}$ | $0.3617_{\pm0.1186}$ | $19.1474_{\pm1.8355}$ |

*Table 26.* Evaluation results for Synth-M dataset

| Model | ACD | SD | KD | MDD | FID | J-FTSD | Precision | Recall | Joint Precision | Joint Recall | CTTP Score |
|---|---|---|---|---|---|---|---|---|---|---|---|
| Bridge | $0.0580_{\pm0.0239}$ | $0.1382_{\pm0.1703}$ | $0.4220_{\pm0.2471}$ | $0.0294_{\pm0.0003}$ | $46.1556_{\pm12.6821}$ | $53.5891_{\pm12.6868}$ | $0.6962_{\pm0.0760}$ | $0.1510_{\pm0.0483}$ | $0.4030_{\pm0.0832}$ | $0.2333_{\pm0.0861}$ | $18.8397_{\pm0.5783}$ |
| DiffuSETS | $\mathbf{0.0413}_{\pm0.0190}$ | $0.0847_{\pm0.0714}$ | $\mathbf{0.1596}_{\pm0.1405}$ | $0.0209_{\pm0.0030}$ | $35.6547_{\pm6.2623}$ | $42.0699_{\pm9.1293}$ | $0.6668_{\pm0.1098}$ | $0.2551_{\pm0.0755}$ | $0.5958_{\pm0.2316}$ | $0.4658_{\pm0.1855}$ | $21.2378_{\pm4.6499}$ |
| Diffusion-TS | $0.0417_{\pm0.0101}$ | $0.0672_{\pm0.0375}$ | $1.3035_{\pm0.0066}$ | $0.0436_{\pm0.0013}$ | $68.3258_{\pm5.0353}$ | $75.9039_{\pm4.4848}$ | $\mathbf{0.9001}_{\pm0.0309}$ | $0.0272_{\pm0.0150}$ | $0.4796_{\pm0.0143}$ | $0.1097_{\pm0.0108}$ | $17.0616_{\pm0.2600}$ |
| T2S | $0.0753_{\pm0.0360}$ | $0.3972_{\pm0.1165}$ | $1.4948_{\pm0.1724}$ | $0.0305_{\pm0.0047}$ | $113.6724_{\pm7.3416}$ | $123.0193_{\pm7.4631}$ | $0.4246_{\pm0.0932}$ | $0.0000_{\pm0.0000}$ | $0.1523_{\pm0.0379}$ | $0.0096_{\pm0.0025}$ | $1.6491_{\pm3.4922}$ |
| TEdit | $0.0556_{\pm0.0061}$ | $0.0314_{\pm0.0065}$ | $0.2552_{\pm0.1086}$ | $0.0155_{\pm0.0016}$ | $35.1944_{\pm0.9255}$ | $43.8138_{\pm1.1236}$ | $0.7398_{\pm0.0177}$ | $0.2298_{\pm0.0203}$ | $0.6117_{\pm0.0138}$ | $0.3923_{\pm0.0090}$ | $21.2387_{\pm0.3449}$ |
| Text2Motion | $0.0764_{\pm0.0132}$ | $0.0572_{\pm0.0455}$ | $0.2479_{\pm0.0465}$ | $0.0140_{\pm0.0007}$ | $60.4473_{\pm16.6599}$ | $65.1781_{\pm16.1543}$ | $0.8230_{\pm0.0588}$ | $0.0746_{\pm0.0694}$ | $0.6024_{\pm0.0818}$ | $0.2304_{\pm0.1378}$ | $19.9581_{\pm1.7206}$ |
| TimeVQVAE | $0.0733_{\pm0.0028}$ | $\mathbf{0.0282}_{\pm0.0148}$ | $0.6307_{\pm0.1129}$ | $0.0211_{\pm0.0006}$ | $60.6209_{\pm0.4303}$ | $66.8617_{\pm0.3628}$ | $0.8671_{\pm0.0062}$ | $0.0531_{\pm0.0037}$ | $0.5932_{\pm0.0054}$ | $0.2115_{\pm0.0043}$ | $19.8635_{\pm0.1381}$ |
| TimeWeaver | $0.0580_{\pm0.0070}$ | $0.0508_{\pm0.0214}$ | $0.1758_{\pm0.1539}$ | $0.0155_{\pm0.0010}$ | $31.1737_{\pm2.9653}$ | $40.3713_{\pm3.1536}$ | $0.7210_{\pm0.0280}$ | $0.2762_{\pm0.0169}$ | $0.5937_{\pm0.0572}$ | $0.4322_{\pm0.0477}$ | $20.8463_{\pm0.9775}$ |
| TTSCGAN | $0.2652_{\pm0.0004}$ | $0.0990_{\pm0.0456}$ | $0.6451_{\pm0.0323}$ | $0.0397_{\pm0.0013}$ | $99.8948_{\pm16.5567}$ | $111.5366_{\pm15.1510}$ | $0.0789_{\pm0.0287}$ | $0.0001_{\pm0.0001}$ | $0.0726_{\pm0.0003}$ | $0.0166_{\pm0.0080}$ | $10.1633_{\pm0.3424}$ |
| VerbalTS | $0.0514_{\pm0.0023}$ | $0.0405_{\pm0.0323}$ | $0.1925_{\pm0.0603}$ | $0.0153_{\pm0.0002}$ | $33.5643_{\pm5.2371}$ | $38.0327_{\pm5.2630}$ | $0.7711_{\pm0.0128}$ | $0.2871_{\pm0.0439}$ | $\mathbf{0.7493}_{\pm0.0315}$ | $0.5033_{\pm0.0727}$ | $\mathbf{24.6611}_{\pm0.8267}$ |
| WaveStitch | $0.0539_{\pm0.0225}$ | $0.2148_{\pm0.1342}$ | $0.3029_{\pm0.2332}$ | $\mathbf{0.0137}_{\pm0.0042}$ | $\mathbf{22.2000}_{\pm7.4868}$ | $\mathbf{36.2440}_{\pm6.5269}$ | $0.5129_{\pm0.0297}$ | $\mathbf{0.5322}_{\pm0.1085}$ | $0.2262_{\pm0.0381}$ | $\mathbf{0.5402}_{\pm0.1755}$ | $11.9796_{\pm2.1070}$ |

*Table 27.* Evaluation results for AirQuality Beijing dataset

| Model | ACD | SD | KD | MDD | FID | J-FTSD | Precision | Recall | Joint Precision | Joint Recall | CTTP Score |
|---|---|---|---|---|---|---|---|---|---|---|---|
| Bridge | **0.0337**$_{\pm0.0069}$ | 0.2094$_{\pm0.0560}$ | 0.7371$_{\pm0.0695}$ | 0.0290$_{\pm0.0012}$ | 159.6945$_{\pm20.3313}$ | 160.6358$_{\pm20.4623}$ | 0.8957$_{\pm0.0040}$ | 0.0458$_{\pm0.0079}$ | 0.8081$_{\pm0.0362}$ | 0.2602$_{\pm0.0690}$ | 5.2102$_{\pm0.7037}$ |
| DiffuSETS | 0.0467$_{\pm0.0051}$ | 0.1189$_{\pm0.0173}$ | **0.3604**$_{\pm0.1177}$ | 0.0249$_{\pm0.0016}$ | 121.7182$_{\pm10.4377}$ | 122.5601$_{\pm10.4545}$ | 0.9183$_{\pm0.0111}$ | 0.0466$_{\pm0.0323}$ | 0.8985$_{\pm0.0229}$ | 0.4681$_{\pm0.0790}$ | 6.1099$_{\pm0.2019}$ |
| Diffusion-TS | 0.0843$_{\pm0.0106}$ | 0.1777$_{\pm0.0848}$ | 0.7073$_{\pm0.1028}$ | 0.0309$_{\pm0.0040}$ | **68.9139**$_{\pm15.8802}$ | **69.9562**$_{\pm15.8739}$ | 0.6881$_{\pm0.0399}$ | 0.1342$_{\pm0.0542}$ | 0.6771$_{\pm0.0294}$ | 0.3827$_{\pm0.0807}$ | 2.9660$_{\pm0.3423}$ |
| T2S | 0.1079$_{\pm0.0264}$ | 0.3482$_{\pm0.1443}$ | 1.5353$_{\pm0.2622}$ | 0.0382$_{\pm0.0045}$ | 223.7563$_{\pm25.8692}$ | 225.3871$_{\pm25.8658}$ | 0.0029$_{\pm0.0040}$ | **0.3919**$_{\pm0.2264}$ | 0.1564$_{\pm0.0922}$ | 0.0281$_{\pm0.0088}$ | −1.2634$_{\pm1.6210}$ |
| TEdit | 0.0369$_{\pm0.0082}$ | **0.0975**$_{\pm0.0171}$ | 0.5143$_{\pm0.0824}$ | **0.0213**$_{\pm0.0008}$ | 109.7493$_{\pm5.9814}$ | 110.7139$_{\pm5.9993}$ | 0.8953$_{\pm0.0242}$ | 0.1456$_{\pm0.0287}$ | 0.8407$_{\pm0.0087}$ | 0.5160$_{\pm0.0375}$ | 4.8901$_{\pm0.5026}$ |
| Text2Motion | 0.0518$_{\pm0.0023}$ | 0.1444$_{\pm0.0659}$ | 0.5000$_{\pm0.0190}$ | 0.0246$_{\pm0.0003}$ | 83.4466$_{\pm5.3609}$ | 84.2614$_{\pm5.3351}$ | 0.9126$_{\pm0.0092}$ | 0.1368$_{\pm0.0201}$ | **0.9610**$_{\pm0.0023}$ | **0.7296**$_{\pm0.0086}$ | **6.9590**$_{\pm0.1399}$ |
| TimeVQVAE | — | — | — | 0.0469$_{\pm0.0199}$ | | | | | | | |
| TimeWeaver | 0.0400$_{\pm0.0090}$ | 0.3657$_{\pm0.3719}$ | 1.1163$_{\pm0.9383}$ | 0.0223$_{\pm0.0011}$ | 108.5736$_{\pm17.3495}$ | 109.5707$_{\pm17.4145}$ | 0.8407$_{\pm0.0568}$ | 0.1144$_{\pm0.0389}$ | 0.8267$_{\pm0.0478}$ | 0.4719$_{\pm0.0876}$ | 4.8673$_{\pm0.3098}$ |
| TTSCGAN | 0.1453$_{\pm0.0008}$ | 0.3554$_{\pm0.0263}$ | 1.1388$_{\pm0.0845}$ | 0.0408$_{\pm0.0021}$ | 260.6681$_{\pm8.8347}$ | 261.9308$_{\pm8.7990}$ | **0.9905**$_{\pm0.0097}$ | 0.0002$_{\pm0.0003}$ | 0.6790$_{\pm0.0311}$ | 0.0272$_{\pm0.0046}$ | 4.5250$_{\pm0.2955}$ |
| VerbalTS | 0.0433$_{\pm0.0067}$ | 0.1661$_{\pm0.0616}$ | 0.6455$_{\pm0.0758}$ | 0.0238$_{\pm0.0021}$ | 110.4491$_{\pm15.4693}$ | 111.3438$_{\pm15.5030}$ | 0.8862$_{\pm0.0116}$ | 0.1045$_{\pm0.0216}$ | 0.8719$_{\pm0.0354}$ | 0.4574$_{\pm0.1072}$ | 5.9685$_{\pm0.3691}$ |
| WaveStitch | 0.0497$_{\pm0.0059}$ | 0.4480$_{\pm0.0737}$ | 3.6987$_{\pm2.3803}$ | 0.0256$_{\pm0.0020}$ | 145.6924$_{\pm7.8526}$ | 146.9278$_{\pm7.8691}$ | 0.8476$_{\pm0.0803}$ | 0.0829$_{\pm0.0568}$ | 0.7385$_{\pm0.0465}$ | 0.2315$_{\pm0.0299}$ | 4.0933$_{\pm0.2179}$ |

*Table 28.* Evaluation results for ETTm1 dataset

| Model | ACD | SD | KD | MDD | FID | J-FTSD | Precision | Recall | Joint Precision | Joint Recall | CTTP Score |
|---|---|---|---|---|---|---|---|---|---|---|---|
| Bridge | 0.1069$_{\pm0.0210}$ | **0.4925**$_{\pm0.0322}$ | 3.3521$_{\pm0.0656}$ | 0.0272$_{\pm0.0021}$ | 42.0076$_{\pm9.4701}$ | 42.6005$_{\pm9.5725}$ | 0.6698$_{\pm0.0357}$ | 0.4616$_{\pm0.0276}$ | 0.6089$_{\pm0.0069}$ | 0.4793$_{\pm0.0305}$ | 1.8156$_{\pm0.3851}$ |
| DiffuSETS | 0.1420$_{\pm0.0860}$ | 8.5331$_{\pm11.3647}$ | 258.7216$_{\pm428.3528}$ | 0.0424$_{\pm0.0130}$ | 56.4434$_{\pm37.3998}$ | 56.9916$_{\pm37.5302}$ | 0.6126$_{\pm0.4295}$ | 0.3859$_{\pm0.2482}$ | 0.5409$_{\pm0.1071}$ | 0.4806$_{\pm0.2584}$ | 1.9019$_{\pm1.1863}$ |
| Diffusion-TS | 0.0703$_{\pm0.0262}$ | 1.6305$_{\pm0.1972}$ | 6.7182$_{\pm0.1073}$ | 0.0434$_{\pm0.0020}$ | 59.2449$_{\pm21.3858}$ | 59.9125$_{\pm21.3858}$ | 0.5852$_{\pm0.0500}$ | 0.3323$_{\pm0.2293}$ | 0.4727$_{\pm0.0147}$ | 0.4379$_{\pm0.0350}$ | **3.6984**$_{\pm0.2672}$ |
| T2S | 0.1150$_{\pm0.1321}$ | 1.1269$_{\pm1.5301}$ | 3.1332$_{\pm1.8773}$ | 0.0540$_{\pm0.0060}$ | 95.0026$_{\pm49.3734}$ | 95.6860$_{\pm49.4863}$ | 0.1115$_{\pm0.0891}$ | 0.3323$_{\pm0.2293}$ | 0.2886$_{\pm0.1804}$ | 0.2387$_{\pm0.1566}$ | 3.4514$_{\pm0.9474}$ |
| TEdit | 0.1692$_{\pm0.0495}$ | 0.7493$_{\pm0.1354}$ | 3.8786$_{\pm0.3975}$ | **0.0233**$_{\pm0.0046}$ | **23.1472**$_{\pm1.4568}$ | **23.8535**$_{\pm1.4339}$ | 0.6089$_{\pm0.0985}$ | 0.5883$_{\pm0.0525}$ | 0.4828$_{\pm0.0635}$ | 0.5748$_{\pm0.0152}$ | 1.9972$_{\pm0.0725}$ |
| Text2Motion | 0.1114$_{\pm0.0418}$ | 0.8648$_{\pm1.1884}$ | 8.8360$_{\pm13.6306}$ | 0.0243$_{\pm0.0050}$ | 48.5253$_{\pm14.8678}$ | 48.9968$_{\pm14.9018}$ | 0.6105$_{\pm0.1036}$ | 0.2676$_{\pm0.1474}$ | **0.6579**$_{\pm0.0434}$ | 0.4560$_{\pm0.0629}$ | 1.9365$_{\pm0.4914}$ |
| TimeVQVAE | 0.0530$_{\pm0.0183}$ | 1.9516$_{\pm0.1991}$ | 9.3044$_{\pm5.1511}$ | 0.0358$_{\pm0.0025}$ | 28.2658$_{\pm6.1412}$ | 28.7677$_{\pm6.1357}$ | **0.7515**$_{\pm0.0630}$ | 0.3961$_{\pm0.0843}$ | 0.6195$_{\pm0.0233}$ | **0.6089**$_{\pm0.0345}$ | 2.1357$_{\pm0.1940}$ |
| TimeWeaver | 0.0840$_{\pm0.0421}$ | 1.0313$_{\pm0.2021}$ | 4.1960$_{\pm0.9162}$ | 0.0284$_{\pm0.0096}$ | 48.9813$_{\pm10.8794}$ | 49.6649$_{\pm10.8627}$ | 0.6918$_{\pm0.0600}$ | **0.6074**$_{\pm0.0174}$ | 0.4462$_{\pm0.0399}$ | 0.5810$_{\pm0.0774}$ | 2.6474$_{\pm0.6668}$ |
| TTSCGAN | 0.3025$_{\pm0.0004}$ | 1.4856$_{\pm0.1301}$ | 4.8669$_{\pm0.0738}$ | 0.0397$_{\pm0.0128}$ | 109.9494$_{\pm39.1794}$ | 110.6666$_{\pm39.2287}$ | 0.5916$_{\pm0.2918}$ | 0.0310$_{\pm0.0434}$ | 0.5872$_{\pm0.0650}$ | 0.2304$_{\pm0.1585}$ | 1.5059$_{\pm0.4984}$ |
| VerbalTS | 0.1580$_{\pm0.0851}$ | 1.7842$_{\pm1.1089}$ | 4.5994$_{\pm2.9595}$ | 0.0312$_{\pm0.0158}$ | 48.1654$_{\pm36.0804}$ | 48.7078$_{\pm36.0243}$ | 0.5001$_{\pm0.1365}$ | 0.4321$_{\pm0.2522}$ | 0.5983$_{\pm0.1195}$ | 0.4766$_{\pm0.1713}$ | 2.0389$_{\pm0.6608}$ |
| WaveStitch | **0.0522**$_{\pm0.0263}$ | 0.8096$_{\pm0.6837}$ | **2.1095**$_{\pm1.8257}$ | 0.0346$_{\pm0.0037}$ | 57.2002$_{\pm27.9974}$ | 57.7153$_{\pm27.9582}$ | 0.6116$_{\pm0.1092}$ | 0.4460$_{\pm0.1493}$ | 0.5980$_{\pm0.0778}$ | 0.4833$_{\pm0.1294}$ | 1.9481$_{\pm0.8432}$ |

*Table 29.* Evaluation results for Istanbul Traffic dataset

| Model | ACD | SD | KD | MDD | FID | J-FTSD | Precision | Recall | Joint Precision | Joint Recall | CTTP Score |
|---|---|---|---|---|---|---|---|---|---|---|---|
| Bridge | 0.0294$_{\pm0.0019}$ | 0.4812$_{\pm0.2556}$ | 0.1500$_{\pm0.1943}$ | 0.0363$_{\pm0.0051}$ | 14.5301$_{\pm2.1501}$ | 15.0547$_{\pm2.1321}$ | **0.9347**$_{\pm0.0086}$ | 0.2896$_{\pm0.0140}$ | 0.6885$_{\pm0.0511}$ | 0.5944$_{\pm0.0409}$ | 4.7030$_{\pm0.0194}$ |
| DiffuSETS | 0.1470$_{\pm0.0883}$ | 2.2346$_{\pm1.5908}$ | 12.9962$_{\pm11.1741}$ | 0.0408$_{\pm0.0191}$ | 28.6764$_{\pm21.1976}$ | 29.3117$_{\pm21.3957}$ | 0.5917$_{\pm0.3045}$ | 0.3923$_{\pm0.3925}$ | 0.6023$_{\pm0.1723}$ | 0.6127$_{\pm0.1566}$ | 4.7812$_{\pm0.0942}$ |
| Diffusion-TS | 0.1143$_{\pm0.0073}$ | 0.5206$_{\pm0.0358}$ | 0.5546$_{\pm0.0259}$ | 0.0457$_{\pm0.0007}$ | 7.6542$_{\pm0.1156}$ | 8.4541$_{\pm0.0858}$ | 0.8831$_{\pm0.0430}$ | 0.3628$_{\pm0.0476}$ | 0.4939$_{\pm0.0054}$ | 0.5358$_{\pm0.0149}$ | 4.7917$_{\pm0.0246}$ |
| T2S | 0.2451$_{\pm0.0331}$ | 0.3192$_{\pm0.3463}$ | 1.5178$_{\pm0.0697}$ | 0.0281$_{\pm0.0060}$ | 142.4714$_{\pm16.1469}$ | 143.6973$_{\pm16.1239}$ | 0.0000$_{\pm0.0000}$ | 0.1653$_{\pm0.0300}$ | 0.0052$_{\pm0.0042}$ | 0.0708$_{\pm0.0754}$ | 4.7114$_{\pm0.0205}$ |
| TEdit | 0.0310$_{\pm0.0013}$ | 0.2893$_{\pm0.1829}$ | 0.2992$_{\pm0.2275}$ | 0.0176$_{\pm0.0020}$ | 7.4086$_{\pm1.4342}$ | 8.1024$_{\pm1.4414}$ | 0.8277$_{\pm0.0097}$ | 0.8405$_{\pm0.0470}$ | 0.6223$_{\pm0.0059}$ | 0.7407$_{\pm0.0267}$ | 4.7114$_{\pm0.0205}$ |
| Text2Motion | 0.0573$_{\pm0.0095}$ | 0.1450$_{\pm0.0801}$ | **0.0779**$_{\pm0.0754}$ | **0.0102**$_{\pm0.0015}$ | **2.7859**$_{\pm1.9849}$ | **3.1654**$_{\pm1.9895}$ | 0.8548$_{\pm0.0071}$ | 0.7271$_{\pm0.0547}$ | **0.8559**$_{\pm0.0074}$ | 0.8218$_{\pm0.0083}$ | 4.8171$_{\pm0.0368}$ |
| TimeVQVAE | 0.2015$_{\pm0.2004}$ | 0.3223$_{\pm0.1256}$ | 2.0618$_{\pm2.2354}$ | 0.0401$_{\pm0.0049}$ | 33.9157$_{\pm29.6866}$ | 34.7945$_{\pm29.6908}$ | 0.4548$_{\pm0.5252}$ | 0.1102$_{\pm0.1273}$ | 0.3241$_{\pm0.3565}$ | 0.3084$_{\pm0.2920}$ | 4.7204$_{\pm0.2820}$ |
| TimeWeaver | 0.0267$_{\pm0.0040}$ | 0.2055$_{\pm0.1461}$ | 0.5147$_{\pm0.1591}$ | 0.0200$_{\pm0.0013}$ | 8.5903$_{\pm1.4171}$ | 9.3096$_{\pm1.3745}$ | 0.6785$_{\pm0.0899}$ | 0.8306$_{\pm0.0199}$ | 0.6244$_{\pm0.0079}$ | 0.7515$_{\pm0.0314}$ | 4.7321$_{\pm0.0164}$ |
| TTSCGAN | 0.3399$_{\pm0.0019}$ | **0.1098**$_{\pm0.1353}$ | 0.3162$_{\pm0.0363}$ | 0.0205$_{\pm0.0065}$ | 10.5484$_{\pm1.9197}$ | 11.4818$_{\pm1.7699}$ | 0.2352$_{\pm0.0364}$ | 0.1501$_{\pm0.0575}$ | 0.5300$_{\pm0.0348}$ | 0.6533$_{\pm0.0315}$ | 4.7343$_{\pm0.0327}$ |
| VerbalTS | 0.0236$_{\pm0.0064}$ | 0.1323$_{\pm0.1749}$ | 0.0921$_{\pm0.1554}$ | 0.0114$_{\pm0.0013}$ | 4.2518$_{\pm1.3981}$ | 4.6698$_{\pm1.4376}$ | 0.8212$_{\pm0.0148}$ | **0.8730**$_{\pm0.0154}$ | 0.8438$_{\pm0.0212}$ | **0.8502**$_{\pm0.0111}$ | 4.8572$_{\pm0.0309}$ |
| WaveStitch | **0.0215**$_{\pm0.0143}$ | 0.4960$_{\pm0.5091}$ | 0.5819$_{\pm0.8230}$ | 0.0179$_{\pm0.0085}$ | 10.9867$_{\pm7.3042}$ | 11.7453$_{\pm7.4962}$ | 0.8430$_{\pm0.0097}$ | 0.8246$_{\pm0.0213}$ | 0.5964$_{\pm0.0711}$ | 0.7315$_{\pm0.0028}$ | 4.8229$_{\pm0.1616}$ |

*Table 30.* Evaluation results for TelecomTS dataset

| Model | ACD | SD | KD | MDD | FID | J-FTSD | Precision | Recall | Joint Precision | Joint Recall | CTTP Score |
|---|---|---|---|---|---|---|---|---|---|---|---|
| Bridge | 0.0487$_{\pm0.0105}$ | 0.9491$_{\pm0.4001}$ | 9.1130$_{\pm2.3490}$ | 0.0328$_{\pm0.0092}$ | 91.1473$_{\pm30.6606}$ | 97.2862$_{\pm34.8033}$ | 0.6494$_{\pm0.2329}$ | 0.1973$_{\pm0.1341}$ | 0.1758$_{\pm0.0312}$ | 0.2073$_{\pm0.1534}$ | 1.6884$_{\pm1.3374}$ |
| DiffuSETS | 0.0510$_{\pm0.0225}$ | 0.7260$_{\pm0.4397}$ | 5.6203$_{\pm2.3342}$ | 0.0237$_{\pm0.0061}$ | 65.9887$_{\pm43.5030}$ | 71.4901$_{\pm48.4934}$ | 0.5795$_{\pm0.0342}$ | 0.2395$_{\pm0.1685}$ | 0.2928$_{\pm0.0737}$ | 0.3366$_{\pm0.1251}$ | 4.5457$_{\pm3.3877}$ |
| Diffusion-TS | **0.0335**$_{\pm0.0103}$ | 0.7321$_{\pm0.1888}$ | 8.2908$_{\pm0.1904}$ | 0.0274$_{\pm0.0087}$ | 55.8755$_{\pm21.7167}$ | 60.9086$_{\pm21.4270}$ | 0.6040$_{\pm0.1601}$ | 0.4732$_{\pm0.0347}$ | 0.1662$_{\pm0.0098}$ | 0.4699$_{\pm0.0293}$ | 1.9019$_{\pm1.1446}$ |
| T2S | 0.1037$_{\pm0.0461}$ | **0.5110**$_{\pm0.3631}$ | 7.4168$_{\pm1.4742}$ | 0.0352$_{\pm0.0099}$ | 155.1567$_{\pm53.1827}$ | 161.8594$_{\pm51.7074}$ | 0.6012$_{\pm0.1847}$ | 0.0556$_{\pm0.0582}$ | 0.1321$_{\pm0.0769}$ | 0.0390$_{\pm0.0366}$ | 0.1068$_{\pm1.1882}$ |
| TEdit | 0.0661$_{\pm0.0202}$ | 0.5190$_{\pm0.2698}$ | 8.3735$_{\pm1.4039}$ | 0.0240$_{\pm0.0027}$ | 58.6366$_{\pm7.6197}$ | 62.8779$_{\pm7.5856}$ | 0.5381$_{\pm0.0291}$ | 0.4052$_{\pm0.1134}$ | 0.4103$_{\pm0.0534}$ | | 2.1875$_{\pm0.3157}$ |
| Text2Motion | 0.0791$_{\pm0.0194}$ | 0.9901$_{\pm0.5188}$ | 6.9028$_{\pm7.4454}$ | **0.0134**$_{\pm0.0013}$ | 53.5293$_{\pm49.7279}$ | 57.1443$_{\pm55.5941}$ | 0.6083$_{\pm0.0382}$ | 0.3191$_{\pm0.0470}$ | **0.6771**$_{\pm0.0478}$ | **0.5239**$_{\pm0.0683}$ | **9.3332**$_{\pm5.8628}$ |
| TimeVQVAE | 0.0945$_{\pm0.0048}$ | 0.6221$_{\pm0.5406}$ | 9.2415$_{\pm5.1114}$ | 0.0356$_{\pm0.0044}$ | 123.6608$_{\pm29.9672}$ | 127.4633$_{\pm29.4431}$ | 0.4742$_{\pm0.0994}$ | 0.1486$_{\pm0.0431}$ | 0.1206$_{\pm0.0329}$ | 0.1557$_{\pm0.0355}$ | 1.1549$_{\pm0.2186}$ |
| TimeWeaver | 0.0405$_{\pm0.0041}$ | 0.5127$_{\pm0.4358}$ | 7.8862$_{\pm1.0464}$ | 0.0270$_{\pm0.0033}$ | 61.9709$_{\pm8.2364}$ | 66.1418$_{\pm8.0489}$ | 0.5307$_{\pm0.0174}$ | 0.3902$_{\pm0.0212}$ | 0.1752$_{\pm0.0253}$ | 0.3628$_{\pm0.0304}$ | 2.2295$_{\pm0.7243}$ |
| TTSCGAN | 0.0968$_{\pm0.0016}$ | 0.6770$_{\pm0.1124}$ | 7.1605$_{\pm0.0680}$ | 0.0336$_{\pm0.0049}$ | 162.4984$_{\pm16.7258}$ | 165.5502$_{\pm16.3078}$ | 0.4214$_{\pm0.1161}$ | 0.0081$_{\pm0.0087}$ | 0.1009$_{\pm0.0369}$ | 0.0343$_{\pm0.0203}$ | 1.0022$_{\pm0.3437}$ |
| VerbalTS | 0.0681$_{\pm0.0123}$ | 0.7456$_{\pm0.4566}$ | **3.0511**$_{\pm2.1876}$ | 0.0244$_{\pm0.0123}$ | 70.7905$_{\pm49.6275}$ | 76.3063$_{\pm54.6152}$ | 0.4344$_{\pm0.1827}$ | 0.4723$_{\pm0.1165}$ | 0.2819$_{\pm0.1858}$ | 0.4065$_{\pm0.1094}$ | 4.8817$_{\pm4.8977}$ |
| WaveStitch | 0.0359$_{\pm0.0131}$ | 9.1142$_{\pm7.8505}$ | 520.8309$_{\pm673.3080}$ | 0.0254$_{\pm0.0027}$ | **52.9260**$_{\pm3.8831}$ | **57.0388**$_{\pm3.8700}$ | 0.5677$_{\pm0.0573}$ | **0.5003**$_{\pm0.0939}$ | 0.1884$_{\pm0.0125}$ | 0.4248$_{\pm0.0151}$ | 2.9696$_{\pm0.4048}$ |

*Table 31.* Evaluation results for Weather Conceptual dataset

| Model | ACD | SD | KD | MDD | FID | J-FTSD | Precision | Recall | Joint Precision | Joint Recall | CTTP Score |
|---|---|---|---|---|---|---|---|---|---|---|---|
| Bridge | 0.0343$_{\pm0.0026}$ | 19.5672$_{\pm0.0892}$ | 2320.6266$_{\pm0.5614}$ | 0.0205$_{\pm0.0002}$ | 45.0646$_{\pm12.7574}$ | 47.7810$_{\pm13.0501}$ | 0.7533$_{\pm0.0367}$ | 0.0292$_{\pm0.0035}$ | 0.6301$_{\pm0.0803}$ | 0.2497$_{\pm0.0523}$ | 27.3101$_{\pm2.2800}$ |
| DiffuSETS | 0.1379$_{\pm0.0601}$ | 19.1210$_{\pm0.2734}$ | 2317.8549$_{\pm1.5389}$ | 0.0395$_{\pm0.0004}$ | 149.2328$_{\pm16.4051}$ | 156.1811$_{\pm15.6110}$ | 0.5943$_{\pm0.5162}$ | 0.0025$_{\pm0.0027}$ | 0.1057$_{\pm0.0414}$ | 0.0368$_{\pm0.0335}$ | 16.0544$_{\pm1.6389}$ |
| Diffusion-TS | 0.0687$_{\pm0.0097}$ | 25.4508$_{\pm3.6633}$ | 2483.9676$_{\pm148.7709}$ | 0.0215$_{\pm0.0006}$ | 86.3180$_{\pm6.2815}$ | 95.2605$_{\pm5.1351}$ | 0.8902$_{\pm0.0218}$ | 0.1679$_{\pm0.0571}$ | 0.2248$_{\pm0.0050}$ | 0.1474$_{\pm0.0608}$ | 16.8997$_{\pm0.2261}$ |
| T2S | 0.0932$_{\pm0.0099}$ | 18.8126$_{\pm0.2219}$ | 2315.7835$_{\pm3.1762}$ | 0.0303$_{\pm0.0120}$ | 110.0314$_{\pm70.9398}$ | 114.0980$_{\pm71.4178}$ | 0.1852$_{\pm0.2814}$ | 0.2594$_{\pm0.2369}$ | 0.2071$_{\pm0.2433}$ | 0.2154$_{\pm0.2516}$ | 20.2179$_{\pm6.5903}$ |
| TEdit | 0.0341$_{\pm0.0079}$ | 23.4401$_{\pm3.5566}$ | 2362.1122$_{\pm232.1081}$ | 0.0087$_{\pm0.0006}$ | 12.0106$_{\pm0.3898}$ | 14.0126$_{\pm0.3361}$ | 0.8595$_{\pm0.0103}$ | 0.4530$_{\pm0.0183}$ | 0.8321$_{\pm0.0394}$ | 0.7543$_{\pm0.0223}$ | 30.2036$_{\pm0.6333}$ |
| Text2Motion | 0.0500$_{\pm0.0067}$ | 18.1200$_{\pm0.0912}$ | 2298.9082$_{\pm3.6489}$ | 0.0077$_{\pm0.0002}$ | **5.5376**$_{\pm0.3020}$ | **6.8965**$_{\pm0.3269}$ | **0.9215**$_{\pm0.0068}$ | **0.5861**$_{\pm0.0076}$ | **0.9726**$_{\pm0.0013}$ | **0.9174**$_{\pm0.0037}$ | **32.8174**$_{\pm0.5007}$ |
| TimeVQVAE | 0.0707$_{\pm0.0160}$ | 19.0337$_{\pm1.1169}$ | 2319.5249$_{\pm4.1065}$ | 0.0263$_{\pm0.0006}$ | 47.5908$_{\pm4.9055}$ | 53.5766$_{\pm5.1236}$ | 0.3864$_{\pm0.0614}$ | 0.0030$_{\pm0.0033}$ | 0.2642$_{\pm0.0303}$ | 0.1413$_{\pm0.0256}$ | 22.9355$_{\pm0.3264}$ |
| TimeWeaver | 0.0417$_{\pm0.0111}$ | 30.7571$_{\pm5.0285}$ | 2618.7799$_{\pm427.4573}$ | 0.0092$_{\pm0.0009}$ | 14.5267$_{\pm2.7804}$ | 16.7852$_{\pm2.8460}$ | 0.8554$_{\pm0.0144}$ | 0.3404$_{\pm0.0782}$ | 0.7838$_{\pm0.0635}$ | 0.6519$_{\pm0.1023}$ | 29.3724$_{\pm1.0899}$ |
| TTSCGAN | 0.1674$_{\pm0.0000}$ | 19.5566$_{\pm0.0473}$ | 2319.8547$_{\pm0.0190}$ | 0.0278$_{\pm0.0014}$ | 177.7704$_{\pm8.0560}$ | 87.3545$_{\pm6.9016}$ | 0.0988$_{\pm0.0421}$ | 0.0023$_{\pm0.0048}$ | 0.1047$_{\pm0.0207}$ | 0.0302$_{\pm0.0133}$ | 19.4969$_{\pm0.6018}$ |
| VerbalTS | **0.0312**$_{\pm0.0063}$ | **16.8770**$_{\pm3.3247}$ | **2119.4812**$_{\pm115.5857}$ | **0.0069**$_{\pm0.0005}$ | 6.7655$_{\pm0.4107}$ | 8.2458$_{\pm0.4857}$ | 0.9062$_{\pm0.0115}$ | 0.5810$_{\pm0.0376}$ | 0.9365$_{\pm0.0206}$ | 0.8643$_{\pm0.0387}$ | 32.0586$_{\pm0.3861}$ |
| WaveStitch | 0.0360$_{\pm0.0078}$ | 19.3397$_{\pm3.0419}$ | 2381.9672$_{\pm223.8178}$ | 0.0100$_{\pm0.0015}$ | 23.5472$_{\pm10.8784}$ | 25.8877$_{\pm11.4028}$ | 0.8214$_{\pm0.0593}$ | 0.3806$_{\pm0.1746}$ | 0.6773$_{\pm0.1797}$ | 0.5343$_{\pm0.2120}$ | 28.1586$_{\pm2.5472}$ |

*Table 32.* Evaluation results for Weather Morphological dataset

| Model | ACD | SD | KD | MDD | FID | J-FTSD | Precision | Recall | Joint Precision | Joint Recall | CTTP Score |
|---|---|---|---|---|---|---|---|---|---|---|---|
| Bridge | 0.0434$_{\pm0.0056}$ | 20.2137$_{\pm1.0936}$ | 2332.3187$_{\pm23.7888}$ | 0.0247$_{\pm0.0019}$ | 64.8944$_{\pm13.2709}$ | 66.2769$_{\pm13.2006}$ | 0.6966$_{\pm0.0382}$ | 0.0536$_{\pm0.0298}$ | 0.5948$_{\pm0.0314}$ | 0.3425$_{\pm0.0798}$ | 9.3509$_{\pm0.6411}$ |
| DiffuSETS | 0.1140$_{\pm0.0194}$ | 19.3972$_{\pm0.0958}$ | 2316.9628$_{\pm0.8818}$ | 0.0351$_{\pm0.0073}$ | 157.8052$_{\pm30.8667}$ | 159.6305$_{\pm30.7734}$ | 0.6047$_{\pm0.0700}$ | 0.0359$_{\pm0.1084}$ | 0.0432$_{\pm0.0213}$ | | 6.6785$_{\pm0.4084}$ |
| Diffusion-TS | 0.0721$_{\pm0.0042}$ | 21.5415$_{\pm1.8096}$ | 2381.3810$_{\pm67.7593}$ | 0.0182$_{\pm0.0022}$ | 71.7762$_{\pm19.3621}$ | 74.0661$_{\pm19.2738}$ | 0.8374$_{\pm0.0534}$ | 0.2436$_{\pm0.0413}$ | 0.5053$_{\pm0.0415}$ | 0.2988$_{\pm0.0852}$ | 6.5788$_{\pm0.0631}$ |
| T2S | 0.1051$_{\pm0.0062}$ | 18.9384$_{\pm1.1181}$ | 2307.1773$_{\pm15.1844}$ | 0.0379$_{\pm0.0060}$ | 192.1624$_{\pm16.7269}$ | 194.2186$_{\pm16.1008}$ | 0.0262$_{\pm0.0197}$ | 0.1500$_{\pm0.0008}$ | 0.1502$_{\pm0.0008}$ | 0.0246$_{\pm0.0032}$ | 5.9304$_{\pm0.2041}$ |
| TEdit | 0.0678$_{\pm0.0148}$ | 22.1650$_{\pm2.8657}$ | 2419.3456$_{\pm169.2376}$ | 0.0218$_{\pm0.0031}$ | 65.5004$_{\pm3.5415}$ | 67.7105$_{\pm3.5083}$ | 0.6801$_{\pm0.1054}$ | 0.2309$_{\pm0.0457}$ | 0.4573$_{\pm0.0650}$ | 0.3496$_{\pm0.0231}$ | 7.0731$_{\pm0.0808}$ |
| Text2Motion | 0.0537$_{\pm0.0011}$ | **16.9457**$_{\pm6.8890}$ | 2202.2627$_{\pm81.5057}$ | **0.0116**$_{\pm0.0008}$ | 28.5415$_{\pm3.2102}$ | 29.2164$_{\pm3.3280}$ | **0.8674**$_{\pm0.0180}$ | 0.2289$_{\pm0.0476}$ | **0.8948**$_{\pm0.0168}$ | **0.7569**$_{\pm0.0237}$ | **12.2402**$_{\pm0.2928}$ |
| TimeVQVAE | 0.0743$_{\pm0.0107}$ | 19.5004$_{\pm0.0346}$ | 2320.3193$_{\pm5.6500}$ | 0.0276$_{\pm0.0012}$ | 56.0410$_{\pm1.7473}$ | 58.1840$_{\pm1.7424}$ | 0.4939$_{\pm0.1206}$ | 0.0224$_{\pm0.0178}$ | 0.4527$_{\pm0.0118}$ | 0.2782$_{\pm0.0165}$ | 7.1866$_{\pm0.1688}$ |
| TimeWeaver | 0.0577$_{\pm0.0045}$ | 30.4996$_{\pm9.8795}$ | 3074.8412$_{\pm1242.8652}$ | 0.0209$_{\pm0.0048}$ | 59.2733$_{\pm7.1994}$ | 61.5426$_{\pm7.1911}$ | 0.6905$_{\pm0.0756}$ | 0.1913$_{\pm0.0212}$ | 0.4632$_{\pm0.0466}$ | 0.3435$_{\pm0.0263}$ | 6.9464$_{\pm0.2249}$ |
| TTSCGAN | 0.1674$_{\pm0.0003}$ | 19.5790$_{\pm0.0341}$ | 2317.6421$_{\pm0.0104}$ | 0.0287$_{\pm0.0021}$ | 90.0769$_{\pm2.5817}$ | 92.3755$_{\pm2.5714}$ | 0.3608$_{\pm0.0361}$ | 0.0114$_{\pm0.0077}$ | 0.3905$_{\pm0.0179}$ | 0.1065$_{\pm0.0104}$ | 7.5842$_{\pm0.1109}$ |
| VerbalTS | **0.0380**$_{\pm0.0095}$ | 19.9470$_{\pm5.1280}$ | **2189.2177**$_{\pm79.7855}$ | 0.0131$_{\pm0.0015}$ | **27.5585**$_{\pm0.6906}$ | **28.4955**$_{\pm0.7503}$ | 0.8168$_{\pm0.0375}$ | **0.3803**$_{\pm0.0366}$ | 0.7990$_{\pm0.0427}$ | 0.6867$_{\pm0.0249}$ | 11.0544$_{\pm0.3955}$ |
| WaveStitch | 0.0847$_{\pm0.0280}$ | 23.5656$_{\pm3.5672}$ | 2753.7917$_{\pm388.6124}$ | 0.0230$_{\pm0.0010}$ | 73.5734$_{\pm26.5046}$ | 75.6773$_{\pm26.4374}$ | 0.6954$_{\pm0.1230}$ | 0.1049$_{\pm0.0660}$ | 0.4726$_{\pm0.0331}$ | 0.2945$_{\pm0.1092}$ | 7.0805$_{\pm0.4286}$ |

*Table 33.* Evaluation results for PTB-XL Conceptual dataset

| Model | ACD | SD | KD | MDD | FID | J-FTSD | Precision | Recall | Joint Precision | Joint Recall | CTTP Score |
|---|---|---|---|---|---|---|---|---|---|---|---|
| Bridge | $0.0623_{\pm0.0109}$ | $2.8098_{\pm0.0209}$ | $184.4093_{\pm0.2316}$ | $0.0149_{\pm0.0004}$ | $324.8132_{\pm8.8372}$ | $336.5941_{\pm8.4960}$ | $0.9994_{\pm0.0011}$ | $0.0000_{\pm0.0000}$ | $0.5886_{\pm0.0157}$ | $0.0002_{\pm0.0003}$ | $219.3064_{\pm2.9385}$ |
| DiffuSETS | $0.1051_{\pm0.0163}$ | $5.8130_{\pm2.0835}$ | $212.3178_{\pm45.9144}$ | $0.0187_{\pm0.0109}$ | $247.9444_{\pm75.7033}$ | $277.3275_{\pm56.9501}$ | $0.7419_{\pm0.3434}$ | $0.0027_{\pm0.0023}$ | $0.4659_{\pm0.2270}$ | $0.0224_{\pm0.0357}$ | $234.0506_{\pm9.7990}$ |
| Diffusion-TS | $0.1913_{\pm0.0328}$ | $2.8754_{\pm0.0237}$ | $185.3141_{\pm0.4289}$ | $0.0106_{\pm0.0002}$ | $19.5408_{\pm6.7047}$ | $19.9687_{\pm6.7596}$ | $0.4773_{\pm0.1048}$ | $0.0507_{\pm0.0092}$ | $0.5293_{\pm0.1068}$ | $0.1454_{\pm0.0163}$ | $5.2916_{\pm0.2100}$ |
| T2S | $0.2110_{\pm0.0467}$ | $3.2119_{\pm0.2035}$ | $177.6389_{\pm4.7735}$ | $0.0296_{\pm0.0113}$ | $264.7998_{\pm77.9925}$ | $298.9812_{\pm71.9978}$ | $0.5946_{\pm0.1745}$ | $0.0005_{\pm0.0008}$ | $0.4369_{\pm0.1171}$ | $0.0030_{\pm0.0041}$ | $251.9576_{\pm41.4880}$ |
| TEdit | $0.0861_{\pm0.0166}$ | $2.7860_{\pm0.0251}$ | $182.9535_{\pm0.4488}$ | $0.0123_{\pm0.0031}$ | $250.0269_{\pm17.9467}$ | $283.1505_{\pm12.2875}$ | $0.7050_{\pm0.1150}$ | $0.0000_{\pm0.0000}$ | $0.5127_{\pm0.0309}$ | $0.0015_{\pm0.0027}$ | $247.6307_{\pm9.3556}$ |
| Text2Motion | $0.3232_{\pm0.0300}$ | $3.6515_{\pm0.3254}$ | $176.4981_{\pm6.7947}$ | $0.0184_{\pm0.0003}$ | $375.3458_{\pm11.3807}$ | $384.0842_{\pm10.1483}$ | $1.0000_{\pm0.0000}$ | $0.0000_{\pm0.0000}$ | $0.4729_{\pm0.0241}$ | $0.0000_{\pm0.0000}$ | $202.0106_{\pm3.7835}$ |
| TimeVQVAE | $0.1330_{\pm0.0069}$ | $3.0693_{\pm0.0762}$ | $178.4336_{\pm5.5833}$ | $0.0169_{\pm0.0001}$ | $336.9363_{\pm5.3872}$ | $348.3181_{\pm4.9416}$ | $1.0000_{\pm0.0000}$ | $0.0000_{\pm0.0000}$ | $0.5535_{\pm0.0279}$ | $0.0000_{\pm0.0000}$ | $213.6566_{\pm2.1786}$ |
| TimeWeaver | $0.0957_{\pm0.0134}$ | $2.7221_{\pm0.0887}$ | $182.5281_{\pm1.0799}$ | $0.0166_{\pm0.0024}$ | $226.2710_{\pm35.8371}$ | $269.3634_{\pm34.6358}$ | $0.6708_{\pm0.0833}$ | $0.0003_{\pm0.0003}$ | $0.3922_{\pm0.0087}$ | $0.0070_{\pm0.0046}$ | $247.5866_{\pm7.5712}$ |
| TTSCGAN | $0.1191_{\pm0.0052}$ | $2.8939_{\pm0.0169}$ | $185.1279_{\pm0.0856}$ | $0.0099_{\pm0.0010}$ | $253.2181_{\pm44.4526}$ | $283.2832_{\pm37.8909}$ | $0.8003_{\pm0.1479}$ | $0.0000_{\pm0.0000}$ | $0.5071_{\pm0.0272}$ | $0.0002_{\pm0.0005}$ | $237.0574_{\pm19.2253}$ |
| VerbalTS | $0.0877_{\pm0.0035}$ | $2.7724_{\pm0.0393}$ | $182.9009_{\pm0.4347}$ | $0.0147_{\pm0.0024}$ | $291.2634_{\pm22.8198}$ | $312.2934_{\pm16.3870}$ | $0.7965_{\pm0.2009}$ | $0.0000_{\pm0.0000}$ | $0.5801_{\pm0.0384}$ | $0.0000_{\pm0.0000}$ | $231.1296_{\pm10.9904}$ |
| WaveStitch | $0.2455_{\pm0.0231}$ | $2.9168_{\pm0.1337}$ | $183.0252_{\pm1.1058}$ | $0.0454_{\pm0.0030}$ | $159.5548_{\pm18.2590}$ | $200.7510_{\pm16.2223}$ | $0.3872_{\pm0.1124}$ | $0.0010_{\pm0.0020}$ | $0.3751_{\pm0.0228}$ | $0.0106_{\pm0.0123}$ | $301.7921_{\pm35.3379}$ |

*Table 34.* Evaluation results for PTB-XL Morphological dataset

| Model | ACD | SD | KD | MDD | FID | J-FTSD | Precision | Recall | Joint Precision | Joint Recall | CTTP Score |
|---|---|---|---|---|---|---|---|---|---|---|---|
| Bridge | $0.0561_{\pm0.0069}$ | $2.8160_{\pm0.0264}$ | $184.2775_{\pm0.0876}$ | $0.0158_{\pm0.0004}$ | $500.8181_{\pm26.0678}$ | $513.6664_{\pm22.7653}$ | $0.5895_{\pm0.3693}$ | $0.0000_{\pm0.0000}$ | $0.3423_{\pm0.0101}$ | $0.0011_{\pm0.0003}$ | $140.6902_{\pm12.3004}$ |
| DiffuSETS | $0.1072_{\pm0.0214}$ | $2.6434_{\pm0.4425}$ | $157.9633_{\pm33.0249}$ | $0.0134_{\pm0.0048}$ | $313.8154_{\pm93.6001}$ | $346.6596_{\pm89.2739}$ | $0.4795_{\pm0.2270}$ | $0.0073_{\pm0.0043}$ | $0.4268_{\pm0.0776}$ | $0.0173_{\pm0.0132}$ | $211.1689_{\pm39.4551}$ |
| Diffusion-TS | $0.1684_{\pm0.0336}$ | $2.8478_{\pm0.0687}$ | $183.9858_{\pm2.2999}$ | $0.0132_{\pm0.0045}$ | $399.7197_{\pm70.8724}$ | $437.2591_{\pm62.4090}$ | $0.4992_{\pm0.0582}$ | $0.0223_{\pm0.0189}$ | $0.4721_{\pm0.0582}$ | $0.0708_{\pm0.0324}$ | $9.2693_{\pm1.3258}$ |
| T2S | $0.1938_{\pm0.0339}$ | $3.5991_{\pm0.9877}$ | $176.7059_{\pm6.9755}$ | $0.0295_{\pm0.0149}$ | $399.7197_{\pm70.8724}$ | $437.2591_{\pm62.4090}$ | $0.9020_{\pm0.0396}$ | $0.0021_{\pm0.0021}$ | $0.4022_{\pm0.0374}$ | $0.0026_{\pm0.0026}$ | $150.1159_{\pm18.8086}$ |
| TEdit | $0.0896_{\pm0.0067}$ | $2.8194_{\pm0.0438}$ | $182.7912_{\pm0.4054}$ | $0.0107_{\pm0.0013}$ | $292.2583_{\pm34.9973}$ | $292.2583_{\pm34.9973}$ | $0.8967_{\pm0.0563}$ | $0.0009_{\pm0.0010}$ | $0.5207_{\pm0.0606}$ | $0.0047_{\pm0.0024}$ | $115.7349_{\pm0.4308}$ |
| Text2Motion | $0.2502_{\pm0.0827}$ | $4.7425_{\pm0.1136}$ | $165.2220_{\pm3.7201}$ | $0.0192_{\pm0.0001}$ | $558.6359_{\pm0.9435}$ | $564.2688_{\pm0.3375}$ | $0.9998_{\pm0.0003}$ | $0.0002_{\pm0.0003}$ | $0.3270_{\pm0.0009}$ | $0.0003_{\pm0.0005}$ | $115.7349_{\pm0.4308}$ |
| TimeVQVAE | $0.1271_{\pm0.0131}$ | $3.8563_{\pm0.7084}$ | $204.6594_{\pm31.9805}$ | $0.0167_{\pm0.0004}$ | $541.0194_{\pm10.8172}$ | $554.5409_{\pm9.8160}$ | $0.9472_{\pm0.0359}$ | $0.2059_{\pm0.1926}$ | $0.2991_{\pm0.0128}$ | $0.0231_{\pm0.0257}$ | $120.2263_{\pm3.5252}$ |
| TimeWeaver | $0.1583_{\pm0.0744}$ | $2.9044_{\pm0.1796}$ | $182.1393_{\pm1.1913}$ | $0.0276_{\pm0.0136}$ | $414.3049_{\pm42.5995}$ | $446.2753_{\pm44.4224}$ | $0.6742_{\pm0.0786}$ | $0.0003_{\pm0.0005}$ | $0.4721_{\pm0.0582}$ | $0.0049_{\pm0.0053}$ | $153.4790_{\pm20.0120}$ |
| TTSCGAN | $0.1137_{\pm0.0114}$ | $2.8947_{\pm0.0903}$ | $185.0735_{\pm0.1622}$ | $0.0095_{\pm0.0007}$ | $285.7187_{\pm26.8165}$ | $324.8612_{\pm32.3804}$ | $0.8935_{\pm0.0274}$ | $0.0010_{\pm0.0018}$ | $0.5270_{\pm0.0173}$ | $0.0049_{\pm0.0054}$ | $240.9490_{\pm37.7297}$ |
| VerbalTS | $0.0806_{\pm0.0009}$ | $2.7478_{\pm0.0734}$ | $182.2294_{\pm0.8419}$ | $0.0147_{\pm0.0018}$ | $410.6861_{\pm20.8180}$ | $440.4774_{\pm22.6577}$ | $0.4398_{\pm0.4801}$ | $0.0035_{\pm0.0025}$ | $0.3800_{\pm0.1228}$ | $0.0015_{\pm0.0009}$ | $183.1183_{\pm26.2431}$ |
| WaveStitch | $0.1745_{\pm0.0565}$ | $2.9124_{\pm0.1740}$ | $182.5091_{\pm0.5141}$ | $0.0415_{\pm0.0090}$ | $397.0965_{\pm128.0726}$ | $433.9989_{\pm125.4409}$ | $0.5732_{\pm0.1572}$ | $0.0025_{\pm0.0022}$ | $0.4787_{\pm0.0527}$ | $0.0146_{\pm0.0120}$ | $140.0256_{\pm65.4110}$ |

### D.2.2. GENERATION QUALITY VISUALIZATION AND CASE STUDIES

To complement the quantitative metrics reported in Section 4.1, we provide qualitative visualizations of generated time series across all benchmark datasets. Figures 14–23 present side-by-side comparisons of all evaluated models, where each subplot displays the ground truth (blue) against the generated distribution characterized by the median (red line) and quantile bands (25%–75% dark, 10%–90% light) computed from 10 independent samples. These visualizations offer intuitive insights into each model's ability to capture temporal dynamics, variance calibration, and condition adherence that may not be fully reflected by aggregate metrics.

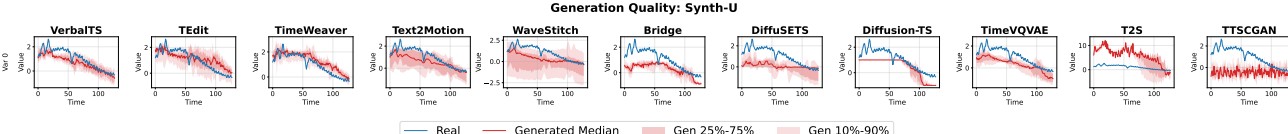

*Figure 14.* Generation Quality Comparison: Synth-U. Each subplot shows one model's performance on one variable. Blue line: ground truth; Red line: generated median; Red bands: 25%-75% (dark) and 10%-90% (light) quantile ranges of 10 generated samples.

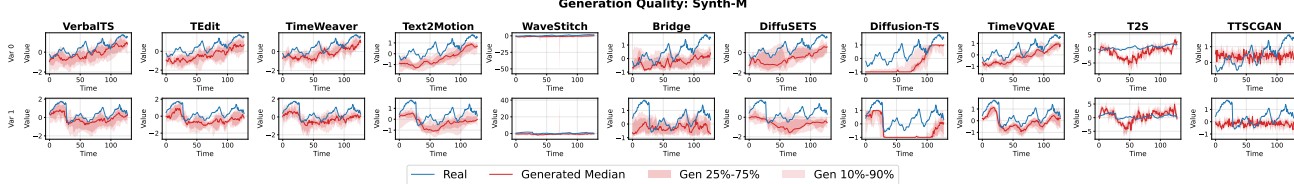

*Figure 15.* Generation Quality Comparison: Synth-M. Each subplot shows one model's performance on one variable. Blue line: ground truth; Red line: generated median; Red bands: 25%-75% (dark) and 10%-90% (light) quantile ranges of 10 generated samples.

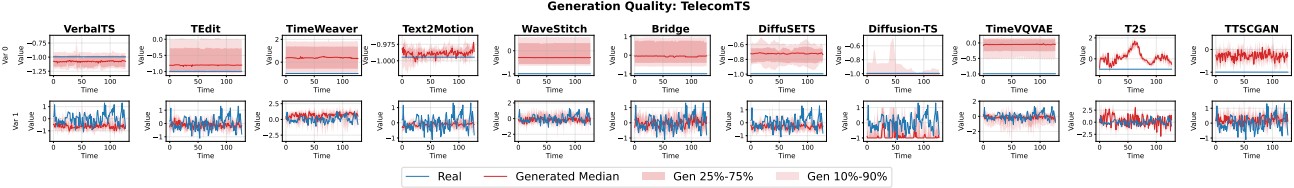

*Figure 16.* Generation Quality Comparison: AirQuality Beijing. Each subplot shows one model's performance on one variable. Blue line: ground truth; Red line: generated median; Red bands: 25%-75% (dark) and 10%-90% (light) quantile ranges of 10 generated samples.

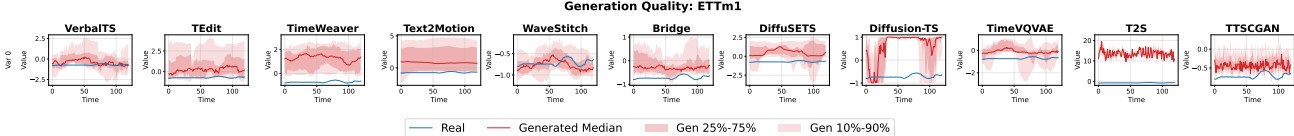

*Figure 17.* Generation Quality Comparison: TelecomTS. Each subplot shows one model's performance on one variable. Blue line: ground truth; Red line: generated median; Red bands: 25%-75% (dark) and 10%-90% (light) quantile ranges of 10 generated samples.

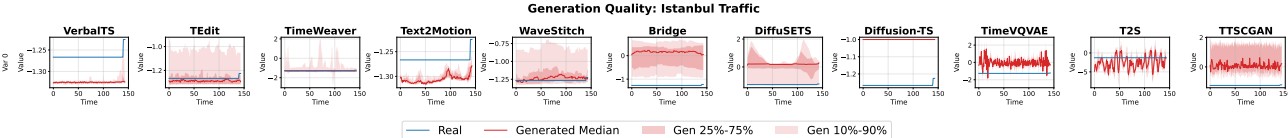

*Figure 18.* Generation Quality Comparison: ETTm1. Each subplot shows one model's performance on one variable. Blue line: ground truth; Red line: generated median; Red bands: 25%-75% (dark) and 10%-90% (light) quantile ranges of 10 generated samples.

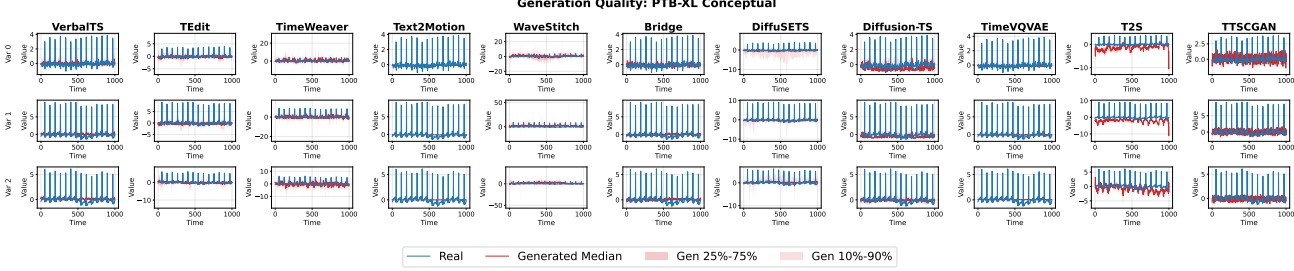

*Figure 19.* Generation Quality Comparison: Istanbul Traffic. Each subplot shows one model's performance on one variable. Blue line: ground truth; Red line: generated median; Red bands: 25%-75% (dark) and 10%-90% (light) quantile ranges of 10 generated samples.

*Figure 20.* Generation Quality Comparison: PTB-XL Conceptual. Each subplot shows one model's performance on one variable. Blue line: ground truth; Red line: generated median; Red bands: 25%-75% (dark) and 10%-90% (light) quantile ranges of 10 generated samples.

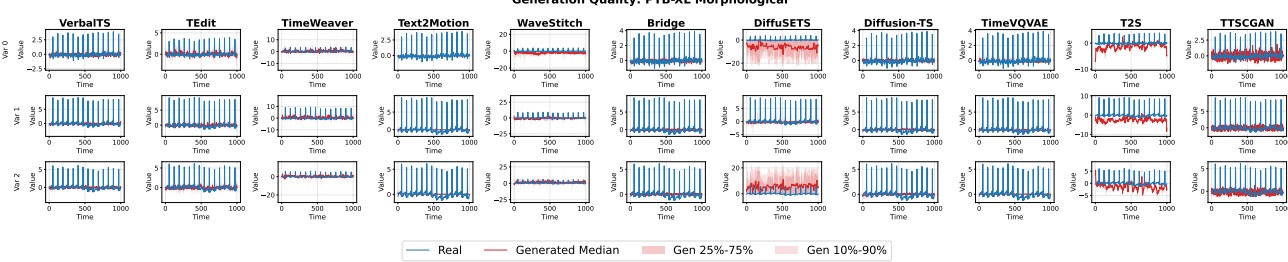

*Figure 21.* Generation Quality Comparison: PTB-XL Morphological. Each subplot shows one model's performance on one variable. Blue line: ground truth; Red line: generated median; Red bands: 25%-75% (dark) and 10%-90% (light) quantile ranges of 10 generated samples.

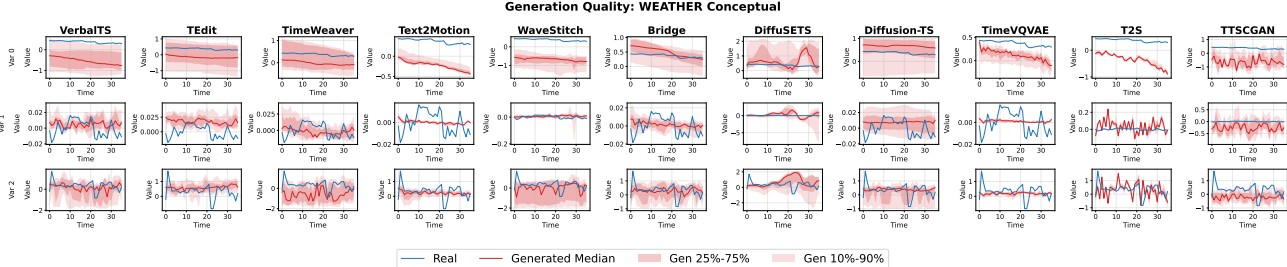

*Figure 22.* Generation Quality Comparison: WEATHER Conceptual. Each subplot shows one model's performance on one variable. Blue line: ground truth; Red line: generated median; Red bands: 25%-75% (dark) and 10%-90% (light) quantile ranges of 10 generated samples.

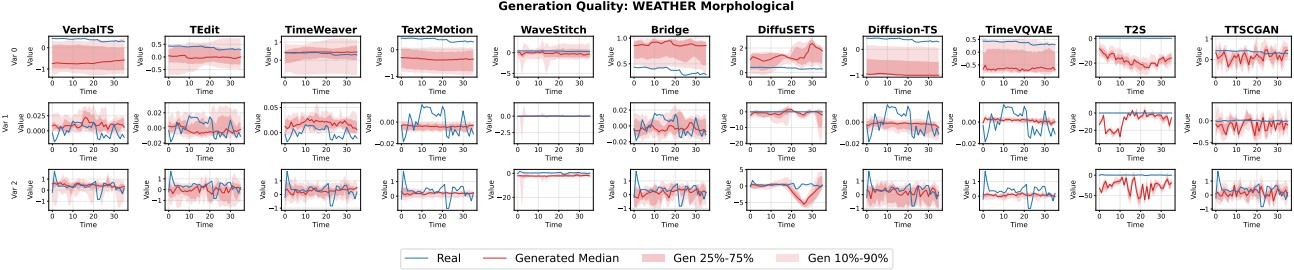

*Figure 23.* Generation Quality Comparison: WEATHER Morphological. Each subplot shows one model's performance on one variable. Blue line: ground truth; Red line: generated median; Red bands: 25%-75% (dark) and 10%-90% (light) quantile ranges of 10 generated samples.

### D.3. Morphological vs. Conceptual Conditions (RQ2)

This section extends the analysis in Section 4.2 by visualizing the rank stability of models across morphological and conceptual conditions. Figure 24 reveals substantial rank shifts when switching between condition types: many models lie far from the diagonal, indicating that their relative performance is highly sensitive to the semantic abstraction level of the conditioning signal. For instance, some models excel under morphological descriptions that explicitly specify temporal patterns, yet underperform when conditioned on conceptual descriptions that require implicit domain knowledge mapping. This divergence underscores that semantic abstraction level constitutes a critical axis for benchmarking conditional generation—evaluations conducted under only one condition type may yield conclusions that do not generalize to the other. The dataset-dependent patterns (compare PTB-XL vs. Weather panels) suggest that the optimal condition type depends on domain characteristics, motivating future work on condition-adaptive architectures.

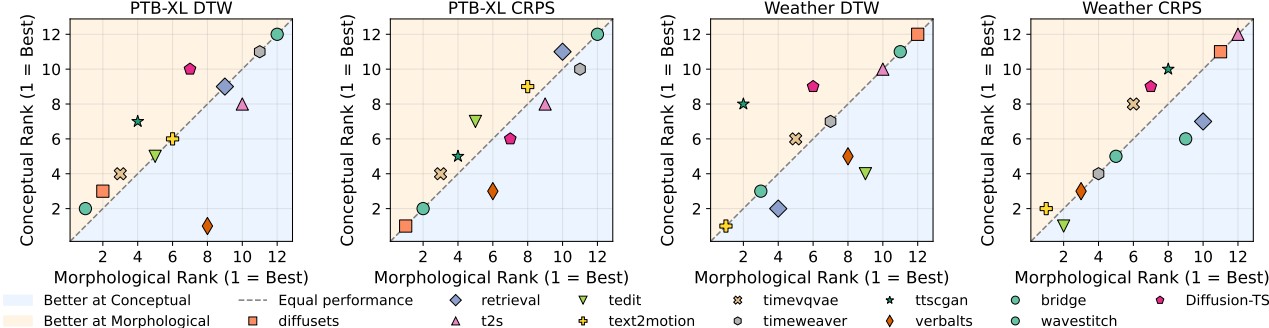

*Figure 24.* **Morphological vs. conceptual conditioning: rank stability.** Each point is a model with ranks computed separately under morphological (x-axis) and conceptual (y-axis) conditions (lower is better). Points near the diagonal indicate stable rankings across condition types, while off-diagonal points indicate sensitivity to condition semantics.

### D.4. Fine-Grained Control (RQ3)

This section reports implementation details and failure-mode analysis for the fine-grained control experiments.

#### D.4.1. SYNTH-U CLASSIFIER DETAILS

To verify whether generated segments contain the specified local patterns, we train a lightweight 1D-CNN classifier on ground-truth Synth-U segments. The classifier consists of three convolutional layers followed by global average pooling and two independent linear heads predicting the number of peaks (3 classes) and sags (2 classes) as defined in Section A.1.3. We train with AdamW (lr=$10^{-3}$, weight decay=$10^{-4}$) for 30 epochs and select the checkpoint with the highest validation joint accuracy. Joint accuracy requires *all three* segment-level predictions (peaks and sags) to be correct simultaneously, providing a strict measure of fine-grained control success.

#### D.4.2. SYNTH-U PER-SEGMENT ACCURACY

Table 35 reports per-segment joint accuracy on Synth-U. VerbalTS and DiffuSETS remain above 60% on every segment, whereas T2S stays near 20.8% across all positions, indicating a consistent failure to capture segment-level structure. The ground-truth classifier reaches approximately 99.9% accuracy on all segments, confirming that the evaluator can recover the scripted segment labels.

#### D.4.3. TEMPORAL ORDER ANALYSIS ON TELECOMTS-SEGMENT

Figure 25 visualizes the temporal order confusion matrices. Each row corresponds to a generated segment (TS $i$), and each column to a within-series caption (Text $j$); an ideal model should exhibit strong diagonal dominance. Strikingly, all generative models produce near-uniform distributions (entries $\approx 0.25$), indicating that they fail to associate segment-level descriptions with their correct temporal positions. In contrast, the ground-truth retrieval (rightmost panel) shows clear diagonal structure, confirming that the alignment between segments and texts is learnable in principle; however, current generators fail to capture it.

*Table 35.* **Per-segment joint accuracy on Synth-U.** Values are percentages; generated-model rows report mean±std over seeds, and Retrieval and Ground truth are single-run point estimates.

| Model | Beginning | Middle | End | Avg. |
|---|---|---|---|---|
| VerbalTS | 79.50±2.91 | 69.11±1.88 | 68.33±5.34 | 72.31±3.30 |
| DiffuSETS | 75.41±11.14 | 68.93±11.63 | 66.29±10.70 | 70.21±11.04 |
| Retrieval | 62.28 | 59.52 | 53.77 | 58.53 |
| Bridge | 58.65±14.58 | 51.12±17.14 | 45.83±10.40 | 51.86±13.49 |
| Text2Motion | 46.57±1.74 | 38.95±1.69 | 38.14±1.54 | 41.22±1.66 |
| T2S | 20.84±0.79 | 20.60±0.21 | 20.82±0.75 | 20.75±0.27 |
| Ground truth | 99.90 | 99.90 | 99.85 | 99.88 |

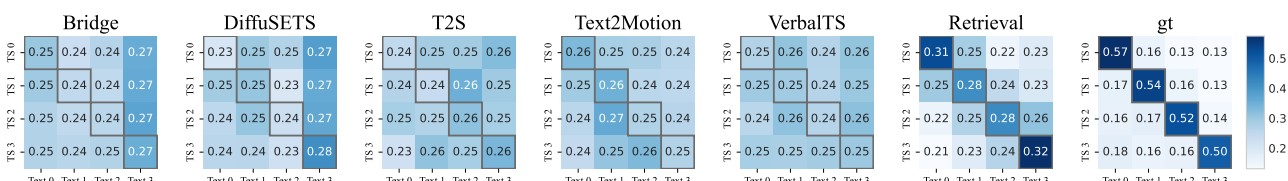

*Figure 25.* **Temporal order confusion matrices on TelecomTS-Segment.** Rows denote generated segments (TS $i$) and columns denote within-series captions (Text $j$); diagonal dominance indicates correct segment–text alignment.

## D.5. Practical Utility (RQ5)

As discussed in Section 4.5, we evaluate the practical utility of generated time series by measuring how well they can substitute for real data in downstream classification tasks. This standardized classification-based protocol serves as an initial reproducible utility test rather than a complete assessment of all possible downstream applications, such as anomaly detection augmentation, forecasting model stress testing, or synthetic clinical trial data generation. Following the drop rate metric defined in Equation 2, we train classifiers on fully generated data and compare their performance against classifiers trained on real data. Table 37 presents the detailed Drop Rate for each model across ten dataset configurations, complementing the aggregated visualization in Figure 6.

To assess the robustness of the practical-utility evaluation, we repeat the train-on-synthetic, test-on-real protocol across ten dataset configurations using three standard deep time-series classification backbones (Foumani et al., 2024): a 1D convolutional classifier, a bidirectional LSTM classifier, and a Transformer encoder classifier. For each dataset, generator, and classifier architecture, we average macro-accuracy over three classifier seeds and compute Drop Rate. Because the main RQ5 result compares Drop Rate values rather than only generator ranks, we summarize architecture sensitivity by the cross-architecture coefficient of variation (CV) and range of Drop Rate for each generator, then average these quantities within each dataset. Table 36 shows that the Drop Rate estimates are numerically stable across classifier architectures: averaged over all dataset configurations, the mean CV is 6.2%, the mean cross-architecture Drop Rate range is 0.107, and 85% of model–dataset cases have a Drop Rate range no larger than 0.20.

*Table 36.* RQ5 sensitivity to downstream classifier architecture across ten dataset configurations. We report the mean coefficient of variation (CV) and mean range of Drop Rate across CNN, BiLSTM, and Transformer classifiers for each dataset configuration. Lower values indicate lower sensitivity to classifier architecture.

| Dataset | Mean CV (%) | Mean DR Range | DR Range $\leq 0.20$ |
|---|---|---|---|
| AirQuality Beijing | 6.5 | 0.112 | 100% |
| ETTm1 | 3.5 | 0.076 | 100% |
| Istanbul Traffic | 7.3 | 0.126 | 90% |
| PTB-XL Concept | 7.3 | 0.139 | 70% |
| PTB-XL Morphology | 1.3 | 0.029 | 100% |
| Synth-M | 8.1 | 0.131 | 80% |
| Synth-U | 7.2 | 0.111 | 80% |
| TelecomTS-Segment | 1.8 | 0.040 | 100% |
| Weather Concept | 8.7 | 0.121 | 80% |
| Weather Morphology | 10.2 | 0.189 | 50% |
| Average | 6.2 | 0.107 | 85% |

*Table 37.* Drop Rate by Model and Dataset Configuration (lower is better)

| Dataset | TEdit | TimeWeaver | DiffuSETS | TimeVQVAE | VerbalTS | WaveStitch | Diffusion-TS | Text2Motion | Bridge | T2S | TTSCGAN |
|---|---|---|---|---|---|---|---|---|---|---|---|
| AirQuality Beijing | 0.301 | 0.336 | 0.574 | 0.509 | 0.631 | 0.505 | 0.659 | 0.696 | 0.787 | 1.106 | 0.863 |
| ETTm1 | 0.290 | 0.382 | 0.365 | 0.560 | 0.279 | 0.507 | 1.604 | 1.086 | 0.598 | 0.426 | 0.573 |
| Istanbul Traffic | 1.232 | 0.705 | 0.918 | 1.210 | 1.020 | 1.046 | 0.652 | 0.741 | 0.953 | 1.290 | 0.747 |
| PTB-XL (Conceptual) | 0.672 | 0.722 | 0.532 | 0.841 | 0.582 | 0.827 | 0.700 | 1.383 | 0.568 | 1.003 | 1.083 |
| PTB-XL (Morphological) | 0.320 | 0.624 | 0.202 | 0.607 | 0.499 | 0.620 | 0.358 | 0.585 | 0.719 | 0.645 | 0.546 |
| Synth-M | 0.093 | 0.096 | 0.538 | 0.420 | 0.098 | 0.571 | 0.626 | 0.537 | 0.730 | 0.905 | 0.989 |
| Synth-U | 0.165 | 0.205 | 0.598 | 0.389 | 0.035 | 0.286 | 0.597 | 0.640 | 0.296 | 1.000 | 0.967 |
| TelecomTS | 0.015 | 0.018 | 0.022 | 0.307 | 0.012 | 0.005 | 0.060 | 0.078 | 0.094 | 0.104 | 0.157 |
| Weather (Conceptual) | 0.850 | 0.879 | 0.993 | 0.898 | 0.704 | 0.787 | 0.790 | 0.840 | 0.863 | 0.799 | 0.942 |
| Weather (Morphological) | 0.668 | 0.436 | 0.433 | 0.517 | 1.303 | 1.073 | 0.955 | 1.311 | 0.994 | 0.555 | 1.036 |
| **Mean** | **0.461** | **0.440** | **0.517** | **0.626** | **0.516** | **0.623** | **0.700** | **0.790** | **0.660** | **0.783** | **0.790** |

