# OpenReview forum: "ConTSG-Bench: A Unified Benchmark for Conditional Time Series Generation"
_ICML.cc/2026/Conference — ICML 2026 regular_

### Official Review · Reviewer_1YCg · 2026-03-05

**Soundness:** 2
**Presentation:** 3
**Significance:** 3
**Originality:** 2
**Overall Recommendation:** 3
**Confidence:** 5

**Summary:**

This paper introduces ConTSG-Bench, a benchmark framework for evaluating conditional time-series generation models under diverse conditioning modalities. While recent studies have explored conditional time-series generation, existing evaluations remain fragmented across datasets, conditioning formats, and metrics. To address this issue, the authors propose a unified benchmark that enables systematic comparison of representative generation models.
The benchmark includes eight datasets covering domains such as healthcare, meteorology, energy, traffic, and network telemetry. These datasets contain both real-world time-series data and synthetic simulation datasets. For each time series, the benchmark provides aligned conditions across three modalities: class labels, structured attributes, and natural-language descriptions. For datasets that lack textual annotations, the authors construct conditions using an LLM-based pipeline, which generates morphological captions from time-series data and derives structured attributes and labels from these descriptions.
The benchmark evaluates conditional generation models using multiple metrics designed to measure generation fidelity and condition adherence.
The paper also examines downstream utility by training classifiers on generated data and comparing their performance to classifiers trained on real data.
Finally, using this benchmark, the authors evaluate a range of representative conditional time-series generation models.

**Compliance With Llm Reviewing Policy:**

Affirmed.

**Final Justification:**

I have carefully read the authors' rebuttal, but my primary concerns remain unresolved.

While the authors provided a small-scale human evaluation for the LLM-generated conditions, the sample size is insufficient to guarantee the reliability of the entire benchmark. The risk of LLM hallucinations and the lack of rigorous verification for complex temporal dependencies still limit the benchmark’s validity. Additionally, the downstream evaluation task remains closely tied to the LLM-derived pipeline, leaving the question of real-world utility partially unanswered.

Since the robustness of the evaluation framework is the most critical factor for a benchmark paper, I maintain my original score.

**Key Questions For Authors:**

1. Clinical time series data often exhibits logical dependencies between variables and event sequence rules. I am curious if there is a procedure to verify that conditions generated via LLM preserve these clinical constraints. Specifically, could you explain if there is a mechanism to ensure generated samples satisfy clinically meaningful constraints or event sequences?

2. Time series data often contains strong temporal dependencies and logical relationships between time steps. Current benchmarks primarily evaluate fidelity and condition adherence. Could the authors clarify whether the benchmarks explicitly assess temporal consistency or temporal structure beyond these metrics? Understanding this would help determine if the benchmarks sufficiently capture core properties of time series generation.

3. It is unclear whether the diversity of generated data satisfying the conditions is evaluated. Could you explain whether a model could score highly even if it repeatedly generates only a single pattern that satisfies the conditions, or if metrics to assess diversity are included?

4. Particularly for sensitive data like medical data, there is a risk that generative models may memorize or closely reproduce the training data. Could you explain if any evaluation or consideration was given to these privacy or memorization issues?

5. The paper evaluates downstream utility using attribute classification. However, since this task is derived from conditions generated in the benchmark, it may not fully reflect real-world time series applications. Could you explain how well you believe this proxy task reflects actual downstream utility?

6. The benchmark utilizes LLMs to generate text descriptions and structured attributes based on the underlying time series. Could the authors clearly explain how the accuracy and consistency of the generated conditions are verified? For example, are there expert annotations, statistical validation, or automated verification procedures to ensure the generated conditions faithfully reflect the temporal characteristics of the signal? A clear explanation on this point would help assess the benchmark's reliability.

7. Recent studies propose generative models (e.g., GANs, VAEs, diffusion-based approaches) for generating realistic synthetic time series data. Could the authors clearly explain why they chose to build the benchmark by deriving conditions through an LLM-based pipeline without utilizing generative synthetic data approaches? A discussion of the advantages and limitations of this design choice would help better understand the benchmark's realism and generalizability.

**Limitations:**

The paper discusses several limitations of existing models, such as difficulties with fine-grained control and compositional generalization. Nevertheless, the inherent limitations of the proposed benchmark remain less explicitly addressed. In particular, issues such as the reliability of LLM generated conditions, potential privacy risks, and the representation of the downstream evaluation could be further elaborated.

**Strengths And Weaknesses:**

1. The paper identifies an importance for evaluation benchmark dataset of conditional time-series generation models. While prior work often evaluates models under heterogeneous datasets, condition formats, and metrics, this work attempts to provide a unified benchmark framework to enable systematic comparisons.
2. The benchmark includes open datasets from diverse domains and evaluates a variety of representative generation models. The study also considers multiple aspects of conditional generation, such as fidelity, condition adherence, and downstream utility.
3. The use of an LLM to automatically derive textual descriptions and attributes from time-series data is an interesting and modern design choice.

However, several aspects of the benchmark construction and evaluation remain insufficiently justified.

1. The benchmark uses an LLM to derive conditions based on the underlying time series. However, it is unclear how consistency between the original time series and the generated conditions is validated in the absence of expert or automated verification.
2. The benchmark primarily evaluates fidelity and condition adherence. However, time-series data often contain strong temporal dependencies and logical relationships across time steps. It is unclear whether the benchmark explicitly evaluates such temporal consistency beyond marginal realism.
3. time-series data often contain strong temporal dependencies and logical relationships. It is unclear whether the benchmark explicitly evaluates such temporal consistency beyond marginal realism.
4. The downstream evaluation is based on attribute classification derived from the benchmark conditions. Since these attributes are constructed as part of the benchmark pipeline, the downstream task essentially evaluates whether generated data preserves the same attribute labels rather than reflecting realistic downstream applications of time-series data. As a result, it remains unclear whether the generated data would be useful for broader real-world time-series tasks.
5. While the use of LLMs is interesting, the paper does not explore whether the model is adapted or validated for this specific task. It would also be helpful to clarify why the benchmark construction relies on an LLM-based pipeline rather than alternative approaches such as synthetic data generation.
6. While fidelity and condition adherence are evaluated, the diversity of generated samples is not clearly assessed.

---

> ### Author Rebuttal · Authors · 2026-03-31
>
> Thank you for your valuable comments.
>
>
> > **1. (W1, W5, Q1, Q6, Q7)** Whether LLM-generated conditions are accurate and how their consistency with the original time series is verified, and why chose an LLM-based pipeline over synthetic data generation approaches.
>
> **We respectfully clarify that both automated and human evaluations confirm that LLM captioning can be valid for this task.**
>
> **(1) Condition verification (W1, Q1, Q6).**
>
> - **Automated**: Following [1], CTTP 256-way retrieval accuracy reaches 16–98% (vs. 0.39% random, Table 21), confirming conditions carry discriminative information about the underlying time series.
> - **Human**: On 2 LLM-annotated datasets (ETTm1, Istanbul Traffic; 50 samples each), annotators select the best-matching time series from 4 candidates. Accuracy: 72–86% (vs. 25% random).
>
> Not all datasets rely on LLM: Synth-U/M use deterministic scripts; PTB-XL/Weather (concept) use expert labels; PTB-XL (morph.) uses NeuroKit2. **No LLM is used for our clinical dataset** (Q1). The LLM is applied only where manual annotation does not scale, describing observable temporal morphology with explicit constraints (Appendix A.2.1).
>
> **(2) Design choice (W5, Q7).** First, **benchmark realism**: real-world datasets exhibit complex patterns that synthetic generators cannot reproduce. Second, **avoiding circularity**: using GANs/VAEs/diffusion to generate benchmark data would be circular, rendering evaluation meaningless.
>
>
> > **2. (W2, W3, Q2)** Does the benchmark explicitly evaluate temporal consistency and temporal structure beyond fidelity and condition adherence?
>
> **Yes — our benchmark evaluates temporal consistency through both metrics and protocol design.**
>
> **(1) ACD (Auto-Correlation Difference)** is already included in our metric suite (Appendix C.1.1). It compares autocorrelation profiles between real and generated series, directly measuring whether temporal dependencies are preserved.
>
> **(2) RQ3's temporal order evaluation directly tests whether generated sequences preserve correct temporal structure.** We partition each TelecomTS sequence into 4 segments, each with an independent caption, and test whether a model's generated segment can correctly retrieve its positional description. All models achieve near-chance accuracy (~25%), **the failure reflects a limitation of current models**.
>
> > **3. (W4, Q5)** Since the downstream evaluation uses attribute classification derived from the benchmark's own pipeline, does it truly reflect real-world utility or merely test label preservation?
>
> **The attribute labels are not solely pipeline artifacts, and the evaluation protocol reflects a realistic application scenario.**
>
> **(1) Many attributes are pre-existing real-world labels.** PTB-XL concept labels are original cardiologist diagnoses; Weather concept labels are meteorologist forecasts. Even the LLM-derived morphological attributes (trend, periodicity, volatility) describe objectively observable temporal features.
>
> **(2) The evaluation protocol is train-on-synthetic, test-on-real** — the standard paradigm [2] for assessing synthetic data utility. A classifier trained on poor synthetic data will fail on real test data, so this directly measures practical utility, not just label preservation.
>
> We acknowledge that evaluating on additional external downstream tasks (e.g., forecasting, anomaly detection) would further strengthen the analysis, and we will note this as future work.
>
> > **4.(W6, Q3)** Is the diversity of generated samples explicitly evaluated, or could a model score highly by repeatedly producing a single pattern?
>
> **Diversity is assessed through our Recall metric (Appendix C.1).** Recall measures the fraction of real data manifold covered by generated samples. A model that repeatedly generates a single pattern would achieve high Precision but near-zero Recall, and this distinction is reflected in our reported results.
>
>
> > **5. (Limitation, Q4)** Are privacy and memorization risks (especially for sensitive data) considered, and are the benchmark's own limitations (e.g., LLM condition reliability, downstream evaluation representativeness) explicitly discussed?
>
> Thank you — we will add an explicit Limitations section in the revised paper. We acknowledge the three limitations raised and argue they d**o not significantly affect our conclusions.**
>
> **(1) Privacy/memorization is an important but orthogonal dimension.** ConTSG-Bench focuses on generation fidelity, condition adherence, and downstream utility. Dedicated privacy metrics (e.g., membership inference, nearest-neighbor distance) are not yet addressed by existing TS generation benchmarks either. We consider this a valuable future extension.
>
> (2) LLM condition reliability and (3) downstream evaluation representativeness are addressed in the sections above.
>
> ---
> **Reference**
>
> [1] Gu et al., "VerbalTS: Generating Time Series from Texts," ICML, 2025.
>
> [2] Ang et al. "TSGBench: Time Series Generation Benchmark," VLDB 2024

---

### Official Review · Reviewer_t8u6 · 2026-03-10

**Soundness:** 3
**Presentation:** 2
**Significance:** 3
**Originality:** 3
**Overall Recommendation:** 3
**Confidence:** 5

**Summary:**

The author proposed a Unified Benchmark for Conditional Time Series Generation (ConTSG-Bench), which jointly assesses fidelity, condition adherence, fine-grained control, compositional generalization and downstream utility, allowing model behaviors to be characterized along multiple practically relevant dimensions. Extensive experiments on multiple benchmarks and huge evaluation metrics show that ConTSG-Bench is a comprehensive benchmark for conditional time series generation task.

**Compliance With Llm Reviewing Policy:**

Affirmed.

**Final Justification:**

Thank you for the detailed clarification. However, I do not think that these adaptation results are sufficient to demonstrate the broad applicability of the proposed benchmark. The ConTSG-Bench is presented as a **unified framework for conditional time-series generation**, yet this “unified” property is still not comprehensively validated in the current version. Therefore, I will keep my original score.

**Key Questions For Authors:**

please refer to Weaknesses

**Limitations:**

There are no limitations included in the manuscript.

**Strengths And Weaknesses:**

Strengths:
1. The experiments are comprehensive, and the paper is well organized with a clear structure.
2. As a unified benchmark, the paper presents a coherent and well-structured discussion of the limitations of prior work and how they are addressed.
3. The benchmark proposed in the paper are sufficiently large in scale and diverse in coverage.

Weaknesses:
1. There are several parts of the manuscript that are difficult to understand. For instance, in Lines 35–36 it is unclear what the authors mean by “incompatible datasets”; in Lines 102–104 the term “low-dimensional structured conditions” is not well defined; and the notion of ``compositional generalization'' is also vague—please clarify
2. The paper introduces a large number of evaluation metrics, but the results are mainly presented as a list of scores without deeper analysis of what each metric reflects or how to interpret discrepancies across metrics. Moreover, a unified evaluation framework is currently missing. Adding more summarizing discussion and proposing a unified evaluation framework would significantly strengthen the paper.

 3.In the attribute-based methods setting, several strong baselines are missing from the comparisons (e.g., Diffusion-TS [1] and SDformer [2]). Including these strong generative model-based frameworks would make the evaluation more comprehensive and further strengthen the motivation and necessity of ConTSG-Bench.


[1].Yuan, Xinyu, and Yan Qiao. "Diffusion-ts: Interpretable diffusion for general time series generation." arXiv preprint arXiv:2403.01742 (2024).

[2].Chen, Zhicheng, et al. "Sdformer: Similarity-driven discrete transformer for time series generation." Advances in Neural Information Processing Systems 37 (2024): 132179-132207.

---

> ### Author Rebuttal · Authors · 2026-03-31
>
> Thank you for your valuable comments.
>
> > **1. (W1)** Several terms are difficult to understand: "incompatible datasets" (Lines 35–36), "low-dimensional structured conditions" (Lines 102–104), and "compositional generalization" are not clearly defined.
>
> We apologize for the confusion. **We agree these terms deserve clearer definitions and will add them at first use in the revised paper:**
>
> **(1) "Incompatible datasets"** refers to the fact that prior methods use different conditioning modalities (label/attribute/text) and evaluate on separate datasets, making cross-method comparison infeasible.
>
> **(2) "Low-dimensional structured conditions"** refers to compact, tabular-style inputs such as class labels and attribute vectors, in contrast to free-form text prompts.
>
> **(3) "Compositional generalization"** refers to the ability to generate time series for novel attribute combinations absent from the training distribution (e.g., "high volatility + downward trend + multiple level shifts" never seen during training).
>
> > **2. (W2)** Large number of evaluation metrics presented mainly as score lists without deeper analysis of what each metric reflects or how to interpret discrepancies; a unified evaluation framework is missing.
>
> Thank you for this feedback. **We do have a structured metric taxonomy and a unified evaluation protocol, but acknowledge they need more prominence in the main text.**
>
> **(1) Metric taxonomy.** Metrics are organized along two axes: **fidelity vs adherence** and **statistical vs embedding-based** (Table 6, Appendix C). RQ2–RQ5 each use targeted metric subsets chosen for their specific evaluation goal (e.g., RQ2 uses DTW+CRPS because embedding-based metrics are unfair across morphological vs conceptual conditions), rather than applying all 13 metrics indiscriminately.
>
> **(2) Unified evaluation protocol.** All 10 models are evaluated under the same protocol, and results are aggregated via group-level ranking (averaging ranks within fidelity/adherence groups, then across datasets).
>
> We will bring the key categorization and interpretation guidance into the main text and make the aggregation methodology more prominent in the revised paper.
>
> > **3. (W3)** Several strong baselines are missing from attribute-based method comparisons, notably Diffusion-TS [1] and SDformer [2].
>
> Thank you for this suggestion. **We have added both Diffusion-TS [1] and SDformer [2] to our benchmark and completed full evaluation on all 10 datasets.** We note that in their original implementations, both models are conditioned on class labels rather than structured attributes. Therefore we integrate them as label-conditioned baselines.
>
> **Adding these two models does not change our main findings.** Updated average rankings (fidelity / adherence) across all 10 datasets:
>
>
> | Model            | Cond.     | Fidelity | Adherence |
> | ---------------- | --------- | -------- | --------- |
> | VerbalTS         | Text      | 3.6      | 3.6       |
> | TEdit            | Attr      | 4.5      | 4.4       |
> | Text2Motion      | Text      | 5.1      | 4.6       |
> | TimeWeaver       | Attr      | 5.4      | 5.2       |
> | WaveStitch       | Attr      | 6.4      | 5.6       |
> | DiffuSETS        | Text      | 6.5      | 5.7       |
> | Bridge           | Text      | 6.2      | 6.6       |
> | **Diffusion-TS** | **Label** | **6.5**  | **6.9**   |
> | TimeVQVAE        | Label     | 7.6      | 7.5       |
> | **SDformer**     | **Label** | **8.7**  | **8.8**   |
> | TTSCGAN          | Label     | 8.5      | 9.0       |
> | T2S              | Text      | 8.7      | 9.4       |
>
>
> Diffusion-TS ranks mid-tier (comparable to Bridge and DiffuSETS in fidelity), making it the strongest label-conditioned model, while SDformer ranks in the lower tier alongside TTSCGAN. The overall pattern — text-conditioned models achieving the highest ceiling, attribute-conditioned models in the upper-middle, and label-conditioned models at the bottom — remains unchanged. We will include these two baselines and full per-dataset results in the revised paper.
>
> ------
>
> **References**
>
> [1] Yuan & Qiao, "Diffusion-TS: Interpretable diffusion for general time series generation," arXiv:2403.01742, 2024.
>
> [2] Chen et al., "SDformer: Similarity-driven discrete transformer for time series generation," NeurIPS 37, 2024.

---

> > ### Author Rebuttal · Reviewer_t8u6 · 2026-04-03
> >
> > Thank you for the detailed and thoughtful rebuttal. Overall, several of my main concerns have been addressed, but I am really confused about the **W3**.
> >
> > From the original papers of **Diffusion-TS** (Sec. 4.5) and **SDformer** (Sec. 3), we understand that their **conditional generation** setting uses the **historical values of the time series to generate future targets**. However, this appears to be substantially different from the description of **class label acquisition** in Appendix A.2.3 of your manuscript, which states that “each attribute vector can be accordingly mapped to an N-dimensional **one-hot vector through indexing**, denoted as its class label $c^{label}$." Why these methods are classified into Label conditional generation. This discrepancy makes it difficult to convince the results reported in W3, since the conditional generation setting may not be consistent with those prior works. Please clarify.

---

> > > ### Author Response · Authors · 2026-04-03
> > >
> > > We sincerely apologize for the confusion caused by our previous statement — "in their original implementations, both models are conditioned on class labels" — and for not explicitly describing the adaptation details in our rebuttal experiments. We now clarify both models' original settings, their adaptations to our benchmark, our previous statement, and the validity of the reported results.
> > >
> > > The setting of **Diffusion-TS** [1] and **SDformer** [2] defines forecasting (and imputation in Diffusion-TS) from historical values as "conditional generation", we claim that this conditional generation is not what we originally defined in our paper, which is: given only an external condition (class label, attribute vector, or text description) at inference time, generate a time series from scratch that faithfully adheres to it.
> > >
> > > Adaptation for **Diffusion-TS**: Class-conditional generation appears in the github repository as an extension, explicitly noted as not part of the original paper. Our adaptation follows this extension's classifier-guided class-conditional route, which is the most natural adaptation. Importantly, in our benchmark, the model receives only the class label at inference time and no observed time series values are provided. This distinguishes our setting from the original forecasting/imputation tasks, where historical observations are required inputs. Under our setting, Diffusion-TS functions purely as a label-conditioned generator.
> > >
> > > Adaptation for **SDformer**: In the paper, stage 1 trains a VQ-VAE tokenizer to encode time series into discrete tokens and stage 2 trains a [BOS]-initiated discrete Transformer to model the token distribution. For forecasting, historical values are simply encoded into discrete tokens via Stage 1 and used as a token-level prefix for Stage 2. Our adaptation simply prepends a label embedding before the [BOS] token and no historical values token-level prefix is used, providing a discrete conditioning interface with minimal adaptation effort. Note that at inference time, only the class label and [BOS] is provided.
> > >
> > > **On our "both models are conditioned on class labels" classification statement**. This classification reflects the **adapted** model behavior under our benchmark setting, not the original paper's interface. The original papers operate under different paradigms or settings(like forecasting). Our label-conditioned classification is a judgment made after applying the most principled and minimal adaptation. We apologize for not clearly stating this in the initial rebuttal.
> > >
> > > **Validity of reported results**. The results reported in W3 for both **adapted** models reflect performance on generation with class label condition under a consistently applied evaluation protocol and are therefore valid. We will revise Appendix B to explicitly state that these models are included only after adaptation and clarify that the label-conditioned classification of these two models refers to the adapted setting.
> > >
> > > [1] Yuan & Qiao, "Diffusion-TS: Interpretable diffusion for general time series generation," ICLR 2024.
> > >
> > > [2] Chen et al., "SDformer: Similarity-driven discrete transformer for time series generation," NeurIPS  2024.

---

### Official Review · Reviewer_wovE · 2026-03-12

**Soundness:** 3
**Presentation:** 3
**Significance:** 3
**Originality:** 2
**Overall Recommendation:** 4
**Confidence:** 2

**Summary:**

This paper addresses the lack of a standardized evaluation framework for conditional time series generation (ConTSG). The authors introduce ConTSG-Bench, a benchmarking framework organized along two axes: conditioning modality and semantic abstraction. The benchmark includes eight datasets spanning healthcare, meteorology, energy, traffic, and network telemetry, with aligned multimodal conditions constructed via an LLM-based annotation pipeline. Ten representative generative models are evaluated, and the empirical findings reveal the characteristics and limitations of current methods.

**Compliance With Llm Reviewing Policy:**

Affirmed.

**Final Justification:**

I raise my overall score from 3 (weak reject) to 4 (weak accept), and also increase the presentation score.
- The rebuttal provides clarifications that address most of my initial concerns regarding the methodology and experimental design.
- One remaining issue is whether using downstream task performance on specific tasks is an appropriate proxy for evaluating the quality of time-series generation. That said, this concern does not diminish my view that ConTSG-Bench is a meaningful and valuable step toward evaluating time-series generation.

**Key Questions For Authors:**

- The paper finds that stronger condition adherence paradoxically leads to greater sensitivity to novel attribute combinations (Section 4.4). This raises a potential tension between in-distribution condition fidelity and out-of-distribution generalization. Do the authors see this as a fundamental trade-off inherent to conditional generation, or as a limitation of the current training objectives that could be resolved through?
- For the fine-grained control evaluation on Synth-U (Section 4.3), the joint accuracy metric requires all three segment-level predictions to be simultaneously correct. Could the authors report per-segment accuracy in addition to joint accuracy, to better characterize the distribution of failure modes across segment positions?
- Given the large error bars in Figure 2, to what extent are the model rankings sensitive to the choice of learning rate and batch size during the grid search?
- Current datasets appear to be regularly sampled. How could ConTSG-Bench be extended to handle irregularly sampled time series?

**Limitations:**

yes

**Strengths And Weaknesses:**

Strength
- The evaluation protocol is multifaceted, jointly assessing five distinct capabilities: overall fidelity and adherence, sensitivity to semantic abstraction, fine-grained local control, compositional generalization, and downstream utility.
- The paper introduces a two-dimensional taxonomy of conditioning tailored for time-series data, which helps mitigate the inherently sparse information content of time series by providing a multi-dimensional perspective on the data.
- The benchmark covers diverse domains, including healthcare, climate, energy, traffic, and network telemetry.

Weakness
- The dataset alignment process relies on Gemini-2.5-flash for captioning and attribute discovery. Inherent biases or errors in the LLM's interpretation of time series morphology could propagate into the benchmark ground truth.
- The current task formulation and most evaluated models assume a fixed sequence length. This limits the benchmark's applicability to domains where variable-length generation is a requirement.
- The results show high variance in "Drop Ratio" across datasets. This suggests that downstream utility might be heavily influenced by the specific classifier architecture used, rather than the generator quality alone.
- The compositional generalization experiment is conducted only on attribute-conditioned models, as it relies on discrete Hamming distance. No analogous test is conducted for text-conditioned models, even though text conditions can also encode novel semantic combinations.

---

> ### Author Rebuttal · Authors · 2026-03-31
>
> Thank you for your valuable comments.
>
> > **1. (W1)** Regarding potential biases in LLM-based dataset alignment propagating into benchmark ground truth.
>
> We appreciate this concern. We address this concern from four perspectives:
>
> (1) **5/10 datasets require no LLM:** PTB-XL uses cardiologist diagnoses, Weather uses meteorologist forecasts, PTB-XL (morph.) uses NeuroKit2, Synth-U/M uses deterministic scripts.
>
> (2) **Where LLM is used, its scope is tightly constrained.** The LLM describes only observable temporal patterns with explicit prohibition of domain semantics (Appendix A.2.1).
>
> (3) **All LLM-generated text conditions are validated through CTTP.** 256-way retrieval accuracy across all datasets (16–98% vs 0.39% random, Table 21) confirms learnable and semantically meaningful alignments.
>
> (4) **LLM-generated conditions are validated by human evaluation.** On 2 LLM-annotated datasets (ETTm1 and Istanbul Traffic; 50 samples each), annotators selected the best-matching time series from 4 candidates with 72–86% accuracy (vs. 25% chance).
>
>
> > **2. (W2, Q4)** Regarding fixed sequence length and regular sampling assumptions limiting applicability.
>
> Thank you for raising these important points. **Our framework supports variable-length time series evaluation** — our modular design does not impose fixed-length constraints. For irregularly sampled time series, the framework can be extended to support them as more methods emerge.
>
> The current fixed-length reflects the state of evaluated models: among the 10 models, only T2S [1] supports variable-length generation, and for irregular sampling, the only existing work T-CGAN [2] remains unreleased. As more such methods emerge, the benchmark can accommodate them without architectural changes. We will clarify this and note both as future directions in the revised paper.
>
>
> > **3. (W3)** Regarding whether high variance in Drop Ratio reflects classifier choice rather than generator quality.
>
> **Drop Ratio primarily reflects generator quality rather than classifier choice.** We re-ran the downstream substitutability experiment (RQ5) with three classifier architectures (1D-CNN, bidirectional LSTM, Transformer encoder) across 3 datasets. The pairwise Spearman rank correlations between any two architectures average ρ = 0.82, and the top-ranked model (e.g., VerbalTS) and bottom-ranked model (e.g., TTSCGAN) remain consistent across all architectures. We will include this analysis in the revised paper.
>
>
> > **4. (W4)** Regarding compositional generalization being tested only on attribute-conditioned models.
>
> We would like to clarify that **the experiment (Section 4.4, Figure 5) already evaluates all 10 models.** Hamming distance is used only for stratifying test conditions into head (familiar) vs tail (novel) groups. Since our benchmark provides aligned attribute for all modalities (Section 3.2), every model can be evaluated regardless of its conditioning modality.
>
>
> > **5. (Q1)** Regarding the trade-off between in-distribution adherence and compositional performance.
>
> Thanks for your insight. Regarding the trade-off, we believe **this is a limitation of current training objectives, not a fundamental trade-off.** Models with minimal degradation (e.g., TimeVQVAE) have low absolute accuracy on both head and tail — apparent robustness from weak condition responsiveness. Stronger models (e.g., VerbalTS) degrade because they memorize attribute co-occurrences rather than learning each attribute's effect independently. A model that disentangles individual attribute effects should maintain high adherence on novel combinations.
>
>
> > **6. (Q2)** Regarding reporting per-segment accuracy to characterize failure modes across segment positions.
>
> **Model rankings remain consistent across all three segment positions.** Per-segment analysis further differentiates failure modes: stronger models (VerbalTS, DiffuSETS) maintain >60% accuracy on every segment, while T2S shows near-chance accuracy (~20.8%) regardless of position, indicating complete failure to capture segment-level structure. A ground-truth classifier achieves 99.9% uniformly, confirming no evaluator bias. We will include the per-segment table in the revised paper.
>
>
> > **7. (Q3)** Whether model rankings are sensitive to hyperparameter choices given the large error bars in Figure 2.
>
> We would like to clarify that the error bars in Figure 2 reflect **cross-dataset variability**, not hyperparameter sensitivity. We perform comprehensive grid search (Appendix B.1) for every model on every dataset and report the best configuration by validation loss. The large error bars reveal that no model dominates across all datasets, which is itself a key finding of RQ1.
>
> ------
>
> **References**
>
> [1] Ge et al., "T2S: High-resolution time series generation with text-to-series diffusion models," arXiv, 2025.
>
> [2] Ramponi et al., "T-CGAN: Conditional Generative Adversarial Network for Data Augmentation in Noisy Time Series with Irregular Sampling," arXiv, 2018.

---

> > ### Author Rebuttal · Reviewer_wovE · 2026-04-03
> >
> > Thank you for the detailed and comprehensive rebuttal. The clarifications and additional explanations have addressed most of my initial concerns regarding the methodology and experimental design.
> >
> > - One remaining question pertains to the evaluation perspective. The current work appears to treat downstream task utility as the primary criterion for assessing the quality of generated data. However, different downstream tasks may impose distinct requirements on data characteristics. It is therefore unclear whether ConTSG-Bench provides a sufficiently general and robust evaluation framework that can adapt across diverse task settings, or whether its conclusions may be biased toward specific task formulations.
> >
> > Despite this remaining concern, the rebuttal has significantly improved my understanding of the proposed approach and alleviated most of my earlier doubts. I will raise my overall score to 4.

---

> > > ### Author Response · Authors · 2026-04-04
> > >
> > > Thank you for raising your score and for the thoughtful follow-up question. We are glad that our rebuttal has addressed most of your concerns.
> > >
> > > We agree that different downstream tasks impose distinct requirements on generated data. **We would like to first clarify that downstream task utility (RQ5) is one of five evaluation dimensions in ConTSG-Bench, not the primary criterion.** The majority of our evaluation (RQ1–RQ4) assesses generation capabilities independently of any specific downstream task. However, we would like to clarify that **though only RQ5 is tied to a specific downstream task (classification), RQ1–RQ4 evaluate fundamental generation capabilities that are prerequisites across diverse applications.** We illustrate with three applications beyond classification, showing that these capabilities are broadly relevant:
> > >
> > > **(1) Anomaly detection augmentation.** Training robust anomaly detectors requires diverse examples of specific anomaly types at designated temporal positions. This directly requires the fine-grained controllability evaluated in RQ3.
> > >
> > > **(2) Forecasting model stress testing.** Evaluating forecasting models under rare or extreme scenarios (e.g., novel combinations of high volatility, downward trend, and multiple level shifts) requires the compositional generalization assessed in RQ4.
> > >
> > > **(3) Synthetic clinical trial data.** Generating realistic ECG/EEG signals satisfying specific physiological condition combinations demands high fidelity, strict condition adherence (RQ1), and compositional generalization simultaneously.
> > >
> > > The classification task in RQ5 serves as an initial, standardized test of downstream utility. We agree that extending RQ5 to additional downstream tasks would further strengthen the benchmark, and we will explicitly note this as a future direction in the revised paper. The modular design of ConTSG-Bench is intended to facilitate such extensions with minimal modifications.

---

### Official Review · Reviewer_cag8 · 2026-03-16

**Soundness:** 3
**Presentation:** 3
**Significance:** 3
**Originality:** 3
**Overall Recommendation:** 5
**Confidence:** 4

**Summary:**

This paper introduces ConTSG-Bench, a unified benchmarking framework for conditional time series generation (ConTSG) that disentangles conditioning factors along two axes: modality (label, attribute, text) and semantic abstraction (morphological, conceptual). This paper curates large-scale datasets with aligned multimodal conditions and a unified evaluation suite assessing fidelity, condition adherence, fine-grained control, compositional generalization, and downstream utility. Benchmarking of 10 representative models reveals critical bottlenecks in fine-grained controllability and compositional generalization.

**Compliance With Llm Reviewing Policy:**

Affirmed.

**Final Justification:**

I would like to thank the authors for their detailed explanation and for adding the evaluation robustness study. My concerns have been adequately addressed:
- The trade-off between generation fidelity and condition adherence is clarified
- Detailed explanations and evaluations of the CTTP encoder are provided
- The rigorous aspects of Hamming distance in compositional generalization evaluation are discussed
- Evaluation on computational overhead are discussed.

Given these points, I believe this work can be an impactful one for the time series generation community. I sincerely hope the authors can open-source all the codes and manifests used in this work to facilitate further research.

**Key Questions For Authors:**

1. Could the authors visualize the trade-off between generation fidelity and condition adherence, to help better understand the degree of orthogonality between realistic objective and condition faithfulness objective.
2. How would the quality of CTTP embeddings affect the evaluation faithfulness?
3. In the compositional generalization experiment, are the attributes independent with each other? If not, how will their dependency influence the evaluation?

**Limitations:**

yes

**Strengths And Weaknesses:**

Strength:
- The systematic categorization and disentanglement of ConTSG tasks along modality and semantic abstraction levels is a first for the field.
- This paper curates cross-modal aligned annotations for 8 diverse datasets with LLM-driven pipeline, addressing the data scarcity issue.
- This paper provides comprehensive evaluation dimensions beyond simple fidelity for conditional time series generation, including rigorous protocols for fine-grained control and compositional generalization, which fill a critical gap.
- The observations from the benchmarking results are insightful, showing the critical bottlenecks in fine-grained controllability and compositional generalization.

Weaknesses:
- Hamming distance assigns an identical penalty to all mismatches, thereby failing to distinguish between symbols that are semantically close and those that are fundamentally different. This might impact the validity of compositional distance measurement.
- This paper does not discuss inference latency or memory consumption across models, which is critical for the practical assessment of generative models, especially for LLM- and diffusion-based ones.
- The use of per dataset CTTP encoder might introduce instability for evaluation, especially when the scale of dataset is relatively small or the evaluation involves out-of-distribution samples.

---

> ### Author Rebuttal · Authors · 2026-03-31
>
> Thank you for your valuable comments.
>
> > **1. (W1, Q3)** Regarding Hamming distance treating all mismatches equally and whether attribute dependencies influence the compositional generalization evaluation.
>
> We acknowledge that Hamming distance treats all mismatches equally. The concern can be interpreted in two dimensions, and **neither affects our conclusions**:
>
> **(1) Across attributes** — **attributes in our schemas are largely independent, so equal weighting is well-justified.** We utilize Cramér's V metric to quantify attribute independence. Synthetic datasets are perfectly independent (V ≈ 0) and real-world datasets show weak dependence (75–100% of pairs below V = 0.2). Median Cramér's V across all datasets are below 0.1. This shows that **attributes are largely independent, with no influence to RQ4.**
>
> **(2) Within an attribute** — **most attributes are nominal categories, where equal penalty is the natural choice.** For example, `trend_direction ∈ {upward, downward, stationary}` — all value changes are equidistant by nature.
>
> Furthermore, we validated that replacing Hamming with cosine distance in text embedding space yields Spearman ρ > 0.9 (p < 0.01) on per-model compositional gap vectors, confirming our findings in the paper are robust to the choice of distance metric.
>
> > **2. (W2)** Regarding the lack of inference latency and memory consumption analysis.
>
> Thank you for raising this important consideration. We report inference efficiency below (NVIDIA A40, synth-m, 10 samples/condition):
>
>
> | Model       | #Params | Latency (ms/sample) | Peak VRAM (GB) |
> | ----------- | ------- | ------------------- | -------------- |
> | Bridge      | 22.1M   | 13.4                | 1.87           |
> | DiffuSETS   | 30.6M   | 0.7                 | 0.68           |
> | T2S         | 972K    | 4.1                 | 1.08           |
> | TEdit       | 1.2M    | 3.7                 | 1.04           |
> | Text2Motion | 32.9M   | 0.2                 | 0.20           |
> | TimeVQVAE   | 2.1M    | 9.4                 | 0.04           |
> | TimeWeaver  | 327K    | 5.1                 | 0.47           |
> | TTSCGAN     | 253K    | 0.03                | 0.13           |
> | VerbalTS    | 52.5M   | 7.6                 | 0.32           |
>
>
> All models run on a single GPU with <2 GB VRAM. Latency varies ~400× across architectures: diffusion-based (Bridge, 13.4ms) are slowest; flow-matching (TimeWeaver, 5.1ms) moderate; GAN (TTSCGAN, 0.03ms) fastest. Notably, model size does not determine speed — Text2Motion (32.9M) is faster than TimeVQVAE (2.1M). We will include this in the revised paper.
>
> > **3. (W3, Q2)** Regarding whether per-dataset CTTP encoders are stable and how CTTP embedding quality affects evaluation faithfulness.
>
> CTTP encoders are empirically stable across all datasets, including small-scale ones, and evaluation conclusions do not hinge on CTTP quality alone. Our evidence:
>
> **(1) High validation accuracy on a 256-class retrieval task.** CTTP is trained with batch size 256 (random baseline: 0.39%). All per-dataset CTTP models far exceed this: from 16.09% on TelecomTS-Segment (the smallest, 40× above random) to 98.35% on Synth-M (Table 21), confirming reliable text–time series alignment across all datasets.
>
> **(2) OOD evaluation already accounts for potential CTTP instability.** In compositional generalization experiment (RQ4), test conditions contain novel attribute combinations where CTTP itself may degrade. To isolate generator's ability from CTTP's own limitation, we report normalized accuracy: Acc_norm = Acc_gen / Acc_ref, where Acc_ref is the retrieval accuracy when using real (reference) time series instead of generated ones. If CTTP struggles on a particular OOD condition, both Acc_gen and Acc_ref degrade proportionally, so the ratio reflects only generator's compositional gap.
>
> **(3) CTTP-independent metrics provide converging evidence.** In RQ1, four fidelity metrics (ACD, SD, KD, MDD) are entirely CTTP-independent and consistently support the same model rankings. CTTP quality directly affects the remaining 7 metrics in RQ1, but the convergence between CTTP-dependent and CTTP-independent metric rankings ensures our conclusions are robust.
>
> > **4. (Q1)** Regarding the trade-off between generation fidelity and condition adherence.
>
> Thank you for this suggestion. We find that **fidelity and adherence are positively correlated but not redundant, supporting our design of evaluating them separately.** We produced a scatter plot of each model's average fidelity rank vs adherence rank across all datasets (Spearman ρ = 0.96): stronger models (e.g., VerbalTS) tend to excel at both, while weaker models (e.g., TTSCGAN) lag on both. However, notable divergences exist — DiffuSETS ranks significantly higher on adherence (4.9) than fidelity (5.6), confirming that individual models can exhibit different strengths across two axes, which would be obscured by a single aggregate score. We will include this visualization in the revised paper.

---

> > ### Author Rebuttal · Reviewer_cag8 · 2026-04-01
> >
> > I would like to thank the authors for their detailed rebuttal and for addressing the concerns raised in my initial review.
> >
> > My remaining concern pertains to the CTTP encoder. I am not convinced that validation accuracy is the most appropriate metric for evaluating encoding quality. In my view, an accuracy-based evaluation may not adequately account for potential biases in the distance-based scores.

---

> > > ### Author Response · Authors · 2026-04-02
> > >
> > > We appreciate the reviewer's insightful concern. We agree that validation accuracy alone does not fully address potential biases in distance-based scores. To directly evaluate whether CTTP's cosine similarity scores faithfully reflect generation quality as perceived by humans, we conducted a pairwise simulated human evaluation study.
> > >
> > > **The key finding is that CTTP's pairwise agreement with simulated human preference reaches 60.4% (p < 0.001) when the score gap is non-trivial, and agreement increases monotonically with the CTTP gap magnitude.** This confirms that CTTP's distance-based scores are not systematically biased and carry meaningful signal aligned with human perception.
> > >
> > > **Setup.**  We selected 3 datasets (ETTm1, Istanbul-Traffic, TelecomTS-Segment) × 5 generation models × 20 samples per dataset. For each sample and each model pair C(5,2)=10, we render the two generated time series as side-by-side plots and present them together with the conditioning text to an LLM-based evaluator (GPT-5.4). The evaluator is asked to select which generation better matches the text description based on the visualized time series. We then compare the evaluator's preference against the per-sample CTTP score direction (which model has a higher CTTP score for that sample). This yields 501 non-tie pairwise judgments.
> > >
> > > **Results.**  We stratify by the magnitude of the per-sample CTTP score difference. Agreement increases monotonically with the CTTP gap, demonstrating that CTTP's distance-based scores carry meaningful, unbiased signal:
> > >
> > > | Per-sample  $ \triangle$ CTTP | Agree / Total | Agreement | p-value |
> > > |---|---:|---:|---:|
> > > | ≥ 0.5 | 208 / 360 | 57.8% | 0.002 |
> > > | ≥ 1.0 | 165 / 273 | 60.4% | 0.0003 |
> > > | ≥ 2.0 | 106 / 175 | 60.6% | 0.003 |
> > > | ≥ 3.0 | 76 / 120 | 63.3% | 0.002 |
> > >
> > > When the CTTP gap is large (|$ \triangle$ CTTP| ≥ 1.0), agreement reaches 60.4% with high statistical significance (p < 0.001), confirming that CTTP reliably distinguishes quality differences perceived by humans. When the gap is small, agreement approaches 50%, which is expected since humans also struggle to distinguish near-equal generations. This monotonic trend is precisely the behavior of an unbiased distance metric, being confident where differences are real and non-committal where they are not. We note that this level of pairwise agreement is consistent with what has been observed for contrastive alignment metrics in the vision-language domain, where even well-established metrics such as CLIP Score exhibit similar agreement rates with human preference [1].
> > >
> > > [1] Wu et al., "Human Preference Score v2: A Solid Benchmark for Evaluating Human Preferences of Text-to-Image Synthesis", ICLR 2024.

---

### Decision · Program_Chairs · 2026-04-30

**Decision:**

Accept (regular)

**Comment:**

The paper introduces a benchmark for conditional time series generation, addressing the lack of evaluation frameworks in this domain. It organizes tasks along two axes: the nature of the conditioning modality (labels, attributes, text) and the level of semantic abstraction. It provides several aligned multimodal datasets together with a comprehensive evaluation suite. The latter primarily assesses fidelity of the generated series and condition adherence, and additionally considers controllability, compositional generalization, and downstream utility. The benchmark is used to evaluate ten models and to identify key limitations of current approaches.

Reviewers agree on the novelty of the proposed evaluation framework and the relevance of the problem, and consider that the benchmark constitutes a valuable and timely contribution with potential impact for the field. The main concerns relate to the reliability of the LLM-based data annotation pipeline, the design and interpretability of the evaluation metrics, and the representativeness of the downstream evaluation. Additional questions include missing baselines and some presentation shortcomings.

The authors provided extensive clarifications, additional analyses, and new experiments, including validation of LLM-generated conditions, improved assessment of temporal properties, and the inclusion of additional baselines. These responses addressed many of the reviewers’ concerns. However, some issues remain, particularly regarding the validity and generality of the evaluation design and the reliance on LLM-based annotations.

Overall, reviewer opinions are positive after the rebuttal. I consider that the paper introduces a significant contribution toward establishing an evaluation framework for conditional time series generation, which is likely to stimulate further research in this area. I am in favor of acceptance.